# Newly trained navigation and verbal memory skills in humans elicit changes in task-related networks but not brain structure

**Li Zheng[1,2], Zachary Boogaart[3], Andrew McAvan[1], Joshua Garren[1], Stephanie G Doner[1], Bradley J Wilkes[4], Will Groves[5], Ece Yuksel[5], Lucia Cherep[1], Arne Ekstrom[1,2]\*, Steven M Weisberg[5,6]\***

[1]Department of Psychology, University of Arizona, Tucson, United States; [2]Evelyn McKnight Brain Institute, University of Arizona, Tucson, United States; [3]The University of Miami Miller School of Medicine, University of Miami, Coral Gables, United States; [4]Department of Applied Physiology & Kinesiology, University of Florida, Gainesville, United States; [5]Department of Psychology, University of Florida, Gainesville, United States; [6]Evelyn McKnight Brain Institute, University of Florida, Gainesville, United States

**\*For correspondence:**
adekstrom@arizona.edu (AE);
smweis@gmail.com (SMW)

**Competing interest:** The authors declare that no competing interests exist.

## eLife Assessment

This work presents a **useful** investigation of functional and structural brain changes following navigation and verbal memory training. The analyses of whole-brain volumetric changes are **convincing** and support the study's main conclusion regarding the lack of a volumetric whole-brain plasticity effects. Some analyses are **compelling** in demonstrating the presence of longitudinal behavioural effects, the presence of functional activation changes, and the lack of hippocampal volume changes.

**Abstract** Training cognitive skills, such as remembering a list of words or navigating a new city, has important implications for everyday life. Yet, understanding what brain changes in humans underlies the acquisition of complex cognitive skills remains unresolved. Here, we developed and validated intensive multiweek interventions in which participants were randomly assigned training in either navigation or verbal memory. Healthy young participants (N=75) underwent structural and functional imaging prior to and following the training. Based on pre-registered and exploratory analyses, we did not find any evidence for changes to gross hippocampal or hippocampal subfield volume, cortical brain volume, or microstructural properties of major white matter tracts due to the training. In contrast, network-based analyses suggested changes in task-related informational connectivity, which occurred primarily between cortical areas and mostly involved putative cognitive control networks. These results suggest that cognitive interventions target more transient configurations in network connectivity rather than more durable structural changes.

## Introduction

Improving cognitive skills can contribute positively to multiple aspects of daily life, such as education and professional endeavors (*Edwards et al., 2018*; *Kraft et al., 2023*). Yet, the neural changes underlying cognitive skill acquisition remain an unresolved question in cognitive neuroscience,

**eLife digest** Learning new cognitive skills, such as navigating an unfamiliar city or remembering long lists of words, is a fundamental part of daily life. Yet, scientists still do not fully understand how the brain changes as these skills are acquired. Previous studies have often focused on experts, like London taxi drivers or memory champions, whose brains may have developed structural differences over many years of training. But it remains unclear how short-term training affects the brains of typical healthy adults.

To explore this question, Zheng et al. designed two intensive training programs that lasted for several weeks. One taught people how to navigate complex virtual environments, and the other trained them to remember long sequences of words. Young adults completed brain scans before and after the training. The researchers then measured brain structures (such as the size of the hippocampus, a region important for memory and navigation) and assessed how different brain regions interacted during cognitive tasks.

The results revealed that participants became significantly better at skills they had practiced, showing faster learning and improved performance. However, their brain structure did not change; there were no differences in hippocampal volume, cortical thickness, or the microstructure of white matter pathways. Instead, the most noticeable changes occurred in how brain regions communicated with one another during memory and navigation tasks. Networks of brain regions reorganized their patterns of activity, especially in areas involved in cognitive control, suggesting that the brain adapts to new challenges primarily by altering how its existing circuits work together rather than by growing new tissue.

These findings reveal that, at least in the early stages of learning complex skills, the brain's functional networks are more flexible and responsive than its physical structure. This insight could help inform the design of cognitive training programs and rehabilitation strategies by highlighting the importance of targeting dynamic brain activity rather than structural changes.

limiting our ability to target interventions designed to improve these skills. Some past studies have suggested a relationship between cognitive skills and focal gray matter volume (*Boyke et al., 2008*; *Draganski et al., 2004*; *Draganski et al., 2006*; *Engvig et al., 2010*; *Hänggi et al., 2010*; *Maguire et al., 2000*; *Wenger et al., 2021*), although many of these studies have involved comparison of experts with non-experts or specialized populations (e.g. older adults with cognitive decline). Therefore, it remains unclear whether gray matter volume differences underlie the behavioral changes that occur as part of new cognitive skill acquisition and whether the same mechanisms apply to healthy younger adults. Other studies have suggested the importance of changes in either the recruitment of specific brain areas (i.e. functional activation) or interactions between brain regions (i.e. functional connectivity) that may not result in macroscopic structural changes (*Braun et al., 2015*; *Dresler et al., 2017*; *Kraft et al., 2023*). Resolving the neural mechanism(s) underlying the acquisition of new cognitive skills in healthy populations, with the potential for maximal behavioral improvement, is important so we can determine how to target interventions in future studies.

One possibility is that training different cognitive skills leads to changes in similar underlying neural mechanisms by training a cognitive process or enhancing function in a brain region common to both skills. Past studies suggest such overlap for a brain region thought to be broadly important to memory, the hippocampus (*Buzsáki and Moser, 2013*; *Spiers et al., 2001*). People with better navigation and/or verbal memory skills typically have larger hippocampi (*Brunec et al., 2019*; *Maguire et al., 2000*; *Poppenk et al., 2013*), although this finding has not been consistently replicated (*Clark et al., 2020*; *Weisberg et al., 2019*). Alternatively, it might be that learning different cognitive skills targets only partially overlapping or even independent brain networks, which in turn might hint at why transfer of newly trained cognitive skills to other cognitive domains is often modest (*Sala and Gobet, 2017*). For example, some studies suggest only partial overlap between navigation and verbal memory (*Ekstrom and Hill, 2023*; *Fan et al., 2021*; *Weisberg and Newcombe, 2016*), and it is possible that some of the observed differences between experts and non-experts in past studies arose due to partial overlap between verbal memory and navigation. For example, when comparing expert navigators, street and

landmark names would be intertwined with wayfinding skills, and thus it is important to determine whether superior verbal memory or navigation skills underlie greater hippocampal volume.

One issue with comparing experts and non-experts is that they involve between-participant designs, which may be underpowered compared to within-participant approaches (*Marek et al., 2022*) and may introduce other uncontrolled variables. One potential way to address this issue is to employ a within-participant manipulation to measure how a brain region such as the hippocampus (and connected brain networks) might change in the same participant as a result of training. Successful cognitive interventions suggest that targeted within-participant cognitive training, even for as little as 1–2 weeks, can result in improvements to specific cognitive functions, including changes in focal gray matter (*Chavan et al., 2015*; *Colom et al., 2016*; *Draganski et al., 2004*; *Driemeyer et al., 2008*; *Lövdén et al., 2012*; *Miró-Padilla et al., 2021*); but see *Ganesan et al., 2024*. In some instances, interventions may even generalize to areas not explicitly trained but closely related to the training (termed 'near transfer'; *Jaeggi et al., 2014*; *Judd and Klingberg, 2021*; *Karbach and Verhaeghen, 2014*; *Schmiedek et al., 2010*; *Schweizer et al., 2013*). We developed and pre-registered two cognitive training regimens in which we attempted to separately target navigation and verbal memory skills. This allowed us to address the following novel theoretical questions: (1) What neural changes, if any, result from an intensive within-participant intervention that improves memory or navigation skills in healthy young adults? (2) If such changes occur, what is the degree of neural overlap between the acquisition of these cognitive skills?

Here, we tested three different cohorts of healthy young adults using two different cognitive interventions, each consisting of 10 training sessions targeting either navigation or verbal memory. A third active control group watched educational videos containing information about navigation and verbal memory but did not receive explicit training (*Gobet and Sala, 2023*; *Simons et al., 2016*). Before and after training, participants underwent high-resolution structural imaging targeting the hippocampus (T2), diffusion-weighted imaging (DWI), whole brain structural imaging (T1), and task-related functional imaging during a spatial and temporal associative memory task to understand how the hippocampus and related brain networks changed. We pre-registered three novel and specific hypotheses, which are described in more detail here (https://osf.io/etxvj): (1) targeted training over two weeks will result in improvements in navigation or verbal memory, (2) improvements will be specific to the trained task and will transfer to a separate behavioral measure of that skill ('near transfer'), (3 a) if the hippocampus is central to navigation, hippocampal volume should increase from pre-test to post-test following navigation training but not verbal memory training or the video control condition, (3b) if the hippocampus is central to verbal memory, hippocampal volume should increase from pre-test to post-test in the Verbal Memory group but not in the Navigation or Video Control groups, or (3 c) alternatively, hippocampal volume does not reflect acquisition of new verbal memory nor navigation skills. As part of planned exploratory analyses, we also examined changes in microstructural properties of major white matter tracts (DWI), gray matter volume outside of the medial temporal lobes, functional changes to individual brain regions (i.e. univariate functional activation), and changes to brain networks (i.e. task-related informational connectivity).

## Results

### Behavioral results

#### Training-related effects

Participants were randomly assigned to one of three conditions, each of which consisted of 10 training sessions, separated by at least one day over the course of approximately 4–8 weeks ($4.12\pm1.87$ weeks). Each training session lasted 2 hr. In the navigation training condition, we recruited a total of 29 participants, with 27 successfully completing 10 sessions of navigation training in a large city-scale virtual environment ('Virtual Arida', *Figure 1b* left). Briefly, participants were placed at one building and given the name of another building to find. Participants traveled to and from that pair of buildings until they were able to take a path not exceeding 120% of the shortest possible path. Upon reaching the criterion, they navigated between a new pair of landmarks. After one subsection of Virtual Arida was learned completely, participants were trained on integrating that subsection with previously learned subsections, with no more than one subsection learned in a day (see Methods). To quantify the training effect, we analyzed the average distance error traveled and normalized it by 120% of

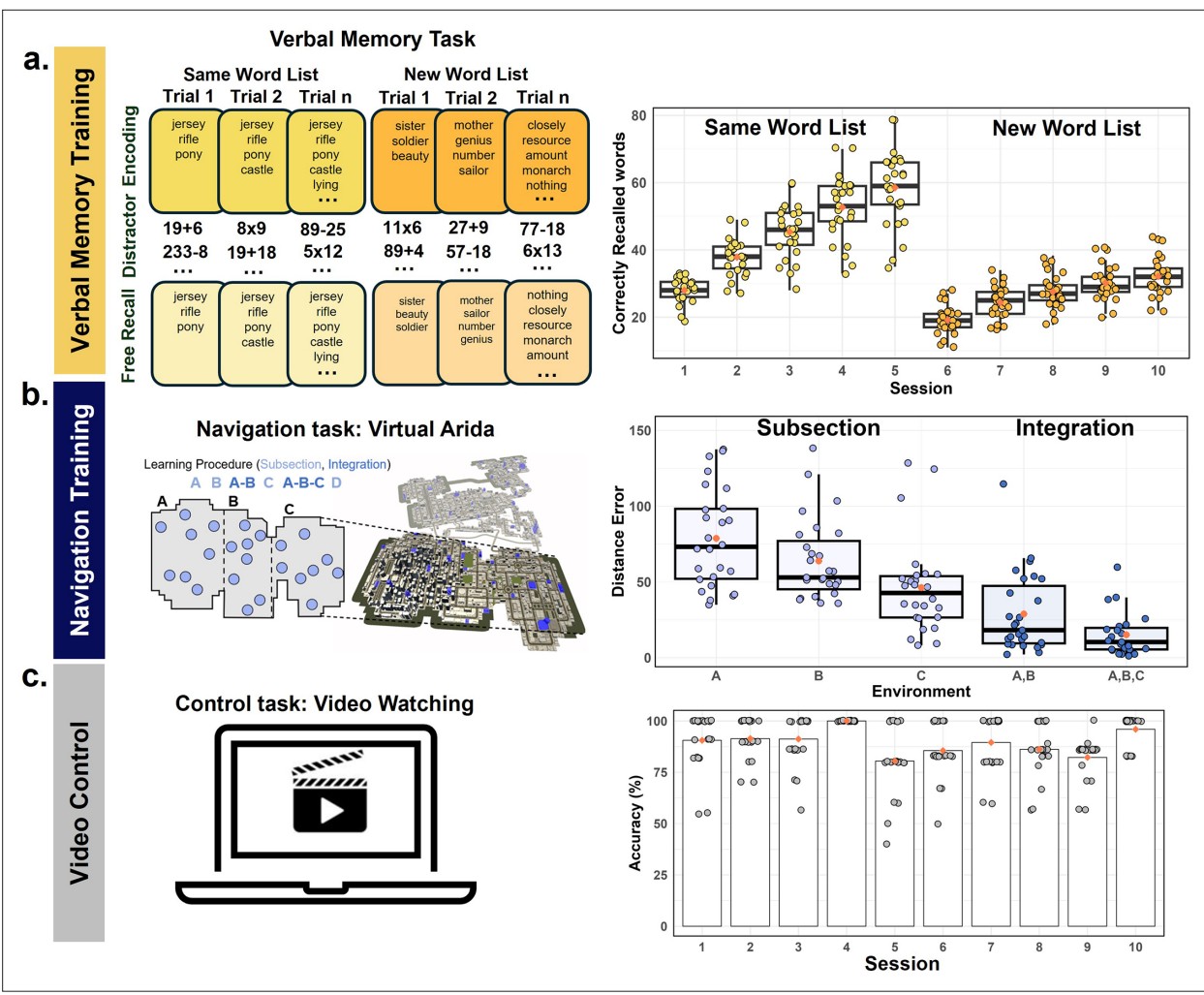

**Figure 1.** Experimental design. Participants were randomly assigned to one of three training conditions as follows. (**a**) In the verbal memory training condition (n=27), participants underwent 10 sessions of verbal memory training involving word free recall after a distractor task, with list complexity increasing over sessions. To quantify the training effect, we conducted a linear regression analysis on the maximum number of correctly recalled words for each day. The slopes of the regression lines for the first 5 days and the last 5 days were calculated separately, and both slopes were significantly greater than zero, indicating a significant improvement in memory performance over the training period: first 5 days, t(26) = 16.971, p<0.001; last 5 days, t(26) = 23.579, p<0.001. (**b**) In the navigation training condition (n=27), participants trained in a large virtual environment, navigating between buildings until optimal paths were learned. Subsections of the environment were integrated progressively across sessions. Participants improved over the course of the training, showing higher error initially for subsection A (M=79.09, SD = 33.22) compared to subsection B (M=65.52, SD = 26.89), which was significantly reduced for subsection C (M=47.44, SD = 31.22). (**c**) In the active control condition (n=21), participants watched videos related to memory and navigation, answering multiple-choice questions afterward, with accuracy consistently above 50%, indicating engagement. Training schedules spanned 4–8 weeks, with each session lasting 2 hr. Notes: Boxplots are centered on the median, boxes extend to first and third quartiles, whiskers extend to 1.5 times the interquartile range or minima/maxima in the absence of outliers. Each individual dot represents data from an individual subject. Red diamonds represent the mean value.

the length of the shortest possible path. While all 27 participants learned subsections A (although data was lost due to technical reasons for one participant), B, and A-B integration, 26 participants progressed to C, while 23 made it to A-B-C integration. 19 participants made it to subsection D, and 14 participants made it to A-B-C-D integration. Thus, to determine the training effect, we performed a repeated-measures ANOVA with subsection as the only factor and threshold-normalized distance error as the dependent measure.

We observed a significant main effect of subsection $F(2, 42)=13.46$, p<0.001, $\eta^2=0.39$; removing sex and site from the model did not change the effect (Appendix 1). Participants improved over the course of the training, showing higher error initially for subsection A (*M*=79.09, SD = 33.22) compared to subsection B (*M*=65.52, SD = 26.89, *Figure 1b* right), which was significantly reduced for subsection

C (*M*=47.44, SD = 31.22). This was also the case for subsections involving integration $F(1,22) = 5.16$, p=0.03, $\eta^2$=0.19, although this finding did not survive inclusion of sex and site in the model (p=0.06). The distance error was reduced from subsections A-B (*M*=22.03, SD = 18.47) to A-B-C (*M*=15.17, SD = 14.47, *Figure 1b* right). In sum, participants improved both in learning new subsections of our large virtual training environment and in integrating multiple areas of the training environment together, suggesting that the navigation training resulted in significant reductions in path error over the course of the training.

In the verbal memory condition, we recruited a total of 32 participants, with 27 successfully completing 10 sessions of training. Briefly, participants viewed words for one second (one at a time), completed math problems as a distractor after all words were presented, and then recalled as many words as they could by typing them in any order they wished. They were trained on a time-based method of loci ('temporal method of loci') to improve their verbal memory (*Bouffard et al., 2018*). During the first five sessions, participants learned the same list of words, with the list growing in length after each successful trial (Same Word List, *Figure 1a* left). In the last five sessions, a new list of words was used for each trial (New Word List, *Figure 1a* left; see Methods). To quantify the training effect, we conducted a linear regression analysis on the maximum number of correctly recalled words for each day. The slopes of the regression lines for the first 5 days and the last 5 days were calculated separately (*Figure 1a* right), as the training paradigm differed between these two phases (see Methods). Both slopes were significantly greater than zero, indicating a significant improvement in memory performance over the training period: first 5 days, t(26) = 16.971, p<0.001; last 5 days, t(26) = 23.579, p<0.001. This effect remained significant after controlling for sex and site (Appendix 1), suggesting robust improvements in verbal memory performance as a result of the training.

In the active control condition, we recruited a total of 22 participants, with 21 successfully completing 10 sessions of watching videos. These videos included interviews, lectures, documentaries, and TV series on topics related to memory and navigation. After watching the videos for that session, participants responded to multiple-choice questions. We did not expect to find a training effect (i.e. improved performance on the questions over sessions), but did find that the percentage of correct responses each day was significantly higher than a 50% guess rate (*Figure 1c* right, *t*s(20) > 8.28, p*s* <0.001, mean = 88.58%, SD = 5.82%), indicating that participants were engaged with the video content.

## Pre-test and post-test behavioral transfer

We administered a set of cognitive tests related to navigation and verbal memory before and after training to track transfer from the training. This involved learning a large virtual environment not related to the training (i.e. Navigation Transfer task: Virtual Silcton *Weisberg et al., 2014*) and learning an untrained list of 12 words (i.e. Verbal Memory Transfer task) until all could be recalled perfectly (or after five trials). When we considered our pre-registered dependent measures specific to verbal memory (average number of words recalled, number of trials to criteria, and slope) and navigation (path error of Navigation Transfer task, pointing error of Navigation Pointing task, and map accuracy of Navigation Model Building task), we found some evidence of improvements in slope specific to the verbal memory training group and path error specific to the navigation training group. The other dependent measures, however, did not show any clear evidence of a training-specific benefit (see Appendix 1). We derived an exploratory measure that allowed us to compare between navigation and verbal memory, termed '*learning rate*'. The learning rate attempts to capture how quickly a participant might learn new information at post-test given that they were trained in improving their acquisition of this information. For example, participants who underwent navigation training should show a smaller difference in path distance on the first ten trials compared to the last ten at post-test compared to pre-test, reflective of faster learning due to the training (see Methods).

To quantify the learning rate for the Navigation Transfer task, a mixed-design ANOVA was conducted with condition (Navigation/Verbal Memory/Video Control) as a between-subjects factor and session (pre/post) as a within-subjects factor. This analysis revealed a significant main effect of session (F(1,70) = 37.52, p<0.001, $\eta^2$=0.202) and a marginally significant main effect of condition (F(2, 70)=2.66, p=0.08, $\eta^2$=0.028), with no significant interaction between condition and session (F(2, 70)=1.82, p=0.17, $\eta^2$=0.020). Paired-sample t-tests indicated that all three groups demonstrated an increase in learning rate from pre to post, with the Navigation group showing the largest

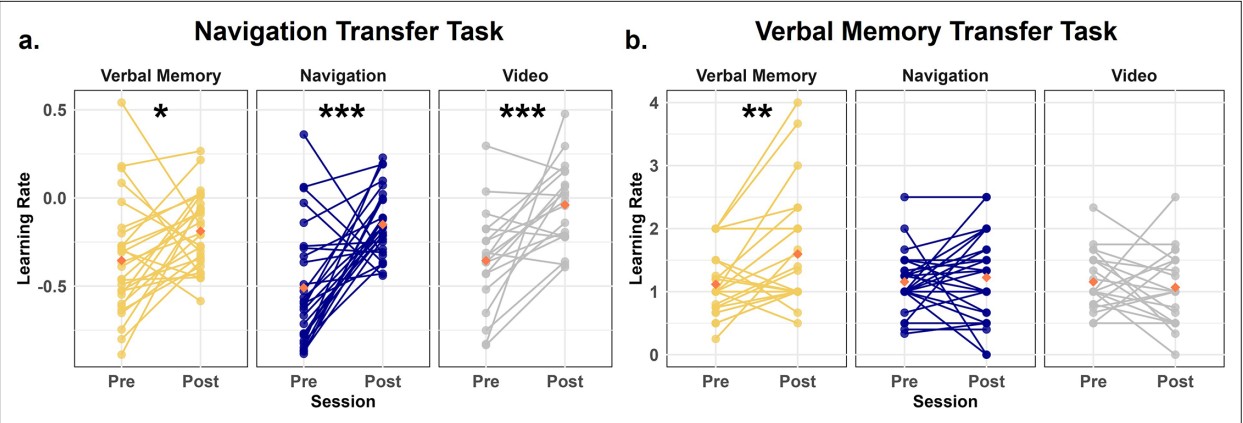

**Figure 2.** Changes in learning rates across training conditions from pre-test to post-test. (**a**) Learning rate for spatial navigation improvement on the Navigation Transfer task. All three groups improved from pre-test to post-test, with the Navigation group demonstrating the largest effect (paired-sample t-tests; Navigation: t(26) = 4.43, p<0.001, Cohen's d=1.32; Verbal Memory: t(26) = 2.31, p=0.03, Cohen's d=0.596; Video Control: t(18) = 3.97, p<0.001, Cohen's d=1.20.) (**b**) Learning rate for verbal memory improvement in the Verbal Memory transfer task. Only the Verbal Memory group significantly improved from pre-test to post-test (paired-sample t-tests; Verbal Memory: t(25) = 3.32, p=0.003, Cohen's d=0.608; Navigation: t(26) = 0.488, p=0.63, Cohen's d=0.113; Video Control: t(20) = –0.55, p=0.588, Cohen's d=0.164). Each individual dot represents data from an individual subject. Red diamonds represent the mean value. *p<0.05, **p<0.01, ***p<0.001.

effect (*Figure 2a*; Navigation: t(26) = 4.43, p<0.001, Cohen's d=1.32; Verbal Memory: t(26) = 2.31, p=0.03, Cohen's d=0.596; Video Control: t(18) = 3.97, p<0.001, Cohen's d=1.20). Results held when controlling for sex and site (see Appendix 1). The larger effect size for navigation suggests a greater benefit to the learning rate for the navigation condition, although this conclusion is tempered by the lack of a significant interaction effect.

To quantify the learning rate for the Verbal Memory Transfer task, we employed a mixed-design ANOVA, with condition (Navigation/Verbal Memory/Video Control) as a between-subjects factor and session (pre/post) as a within-subjects factor. We found a main effect of session ($F_{(1, 71)}$=3.21, p=0.078, $\eta^2$=0.014) and a significant interaction between condition and session ($F_{(2, 71)}$=2.9, p=0.025, $\eta^2$=0.034). Paired-sample t-tests indicated that only the Verbal Memory group showed a significant increase in learning rate from pre-test to post-test (*Figure 2b*; Verbal Memory: t(25) = 3.32, p=0.003, Cohen's d=0.608; Navigation: t(26) = 0.488, p=0.63, Cohen's d=0.113; Video Control: t(20) = –0.55, p=0.588, Cohen's d=0.164). This effect remained significant after controlling for sex and site (see Appendix 1). These findings suggest that verbal memory training resulted in significant improvements from pre-test to post-test in the Verbal Memory Transfer task, specific to the Verbal Memory group.

## Brain structural results
### Medial temporal lobe volume did not change due to the training
As our task provided a separation of navigation and verbal memory skills, with some evidence of their independence in the transfer tests, we next tested the hypothesis that navigation training or verbal memory training (or both) would result in increases in hippocampal volume (as measured with T2 high-resolution imaging, see Methods). We conducted a mixed-design ANOVA with condition (Navigation/Verbal Memory/Video Control) as a between-subjects factor and session (pre/post) as a within-subjects factor. The analysis revealed no significant main effect of session ($F_{(1, 71)}$=0.199, p=0.657, $\eta^2$=0.0002, $BF_{10}$=0.151, moderate evidence against the inclusion of the main effect of session), no significant main effect of condition ($F_{(2, 71)}$=0.113, p=0.893, $\eta^2$=0.003, $BF_{10}$=0.241, moderate evidence against the inclusion of the main effect of condition), and no significant interaction between session and condition ($F_{(2, 71)}$=1.224, p=0.300, $\eta^2$=0.003, $BF_{10}$=0.054, strong evidence against the inclusion of the interaction effect). Paired t-tests further confirmed that none of the three training groups exhibited significant changes in total hippocampal volume from pre to post-test (*Figure 3b*, ps >0.265). This nonsignificant result persisted even after controlling for sex and site as covariates (ps >0.256).

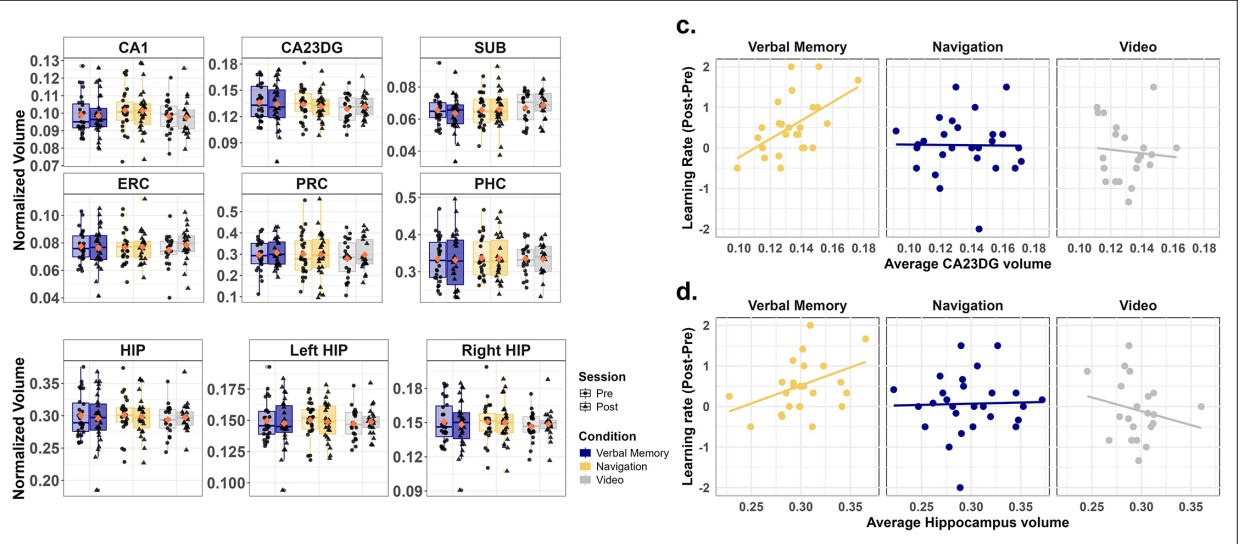

**Figure 3.** Relationship between medial temporal lobe volumes and training-induced learning rate changes across conditions. (**a**) Normalized MTL subregion volumes across conditions (Verbal Memory/Navigation/Video Control) and sessions (pre-test/post-test). No significant changes were detected in CA1, CA23DG, SUB, ERC, PRC, or PHC volumes between pre-test and post-test (paired-sample t-tests, all ps >0.588, FDR-corrected; Verbal Memory: n=26, Navigation: n=27, Video Control: n=21). (**b**) Normalized hippocampal volumes (HIP), including left and right hippocampus, across conditions and sessions. No significant changes in total, left, or right hippocampal volume from pre-test to post-test (paired-sample t-tests, all ps >0.256, FDR-corrected; Verbal Memory: n=26, Navigation: n=27, Video Control: n=21). (**c**) Correlations between changes in learning rate (post-test minus pre-test) and average CA23DG volume across groups. A significant positive correlation was observed in the Verbal Memory group (partial correlation controlling for sex and site: r(23) = 0.493, p=0.017, FDR corrected), suggesting that participants with larger CA23DG volumes exhibited greater improvements in verbal memory performance. No such correlation was observed in the Navigation group (r(25) = –0.083, p=0.953, FDR corrected) or Video Control group (r(19) = –0.12, p=0.953, FDR corrected). (**d**) Correlations between the changes in learning rate (post-test minus pre-test) and average total hippocampal volume across groups. A significant positive correlation was observed only in the Verbal Memory group (partial correlation controlling for sex and site: r(23) = 0.605, p=0.006, FDR corrected). No such correlation was observed in the Navigation group (r(25) = 0.038, p=0.858, FDR corrected) or Video Control group (r(19) = –0.123, p=0.858, FDR corrected). Boxplots are centered on the median, boxes extend to first and third quartiles, whiskers extend to 1.5 times the interquartile range or minima/maxima in the absence of outliers. Each individual dot represents data from an individual subject. Red diamonds represent the mean value.

We further investigated changes in hippocampal volume by conducting a mixed-design ANOVA, with a particular focus on lateralization. Specifically, we included regions of interest (ROI: left/right hippocampus) and session (pre/post) as a within-subjects factors, alongside condition (Navigation/Verbal Memory/Video Control) as a between-subjects factor. Consistent with the previous analysis, no significant main effects or interactions were observed (session: F(1, 71)=0.199, p=0.657, $\eta^2$=0.0002, $BF_{10}$=0.131, moderate evidence against the inclusion of the main effect of session; condition: F(2, 71)=0.113, p=0.893, $\eta^2$=0.003, $BF_{10}$=0.189, moderate evidence against the inclusion of the main effect of condition; ROI: F(1, 71)=0.208, p=0.650, $\eta^2$=0.0001, $BF_{10}$=0.115, moderate evidence against the inclusion of the main effect of ROI; and session ×condition × ROI, F(2, 71)=0.072, p=0.930, $\eta^2$<0.001, $BF_{10}$=0.0002, strong evidence against the inclusion of the interaction effect). Paired-sample t-tests confirmed that none of the three training groups exhibited significant changes in hippocampal volume from pre-test to post-test in either the left hippocampus (*Figure 3b*, ps >0.316) or the right hippocampus (ps >0.265). These findings remained consistent after controlling for sex and site as covariates (left hippocampus: ps >0.306; right hippocampus: ps >0.256).

Because some studies have shown long-axis specialization in correlations between hippocampal volume and behavior (*Angeli et al., 2025*; *Brunec et al., 2018*; *Maguire et al., 2000*), we considered whether anterior and posterior hippocampus might change differentially due to the training, which we analyzed based on the T1 whole brain structural image (see Methods). The previously described mixed-design ANOVA model was expanded to incorporate ROI (anterior and posterior hippocampus) as a within-subjects factor, thereby allowing for the examination of volume differences along the hippocampal anterior-posterior axis. Consistent with the above result, although we found a significant main effect of ROI (F(1,71) = 22.311, p<0.001, $\eta^2$=0.052, $BF_{10}$ >100, strong evidence for the

inclusion of the main effect of ROI), we found no significant main effect of session (F(1, 71)=2.914, p=0.092, $\eta^2$<0.001, BF$_{10}$=0.439, weak evidence against the inclusion of the main effect of session), no significant main effect of condition (F(2, 71)=1.045, p=0.357, $\eta^2$<0.022, BF$_{10}$=0.277, moderate evidence against the inclusion of the main effect of session), and no significant interaction effect between session ×condition × ROI (F(2, 71)=0.032, p=0.968, $\eta^2$<0.001, BF$_{10}$=0.005, strong evidence against the inclusion of the main effect of condition). Simple main effects confirmed that none of the three training groups exhibited significant changes in hippocampal volume from pre-test to post-test neither in the anterior hippocampus (ps >0.272) nor the posterior hippocampus (ps >0.115). These findings remained consistent after controlling for sex and site as covariates (anterior hippocampus: ps >0.260; posterior hippocampus: ps >0.106).

Next, we divided the medial temporal lobe into six subregions: three hippocampal subfields (CA1, CA23DG, and SUB) and three adjacent subregions (PRC, ERC, and PHC). We then conducted a mixed-design ANOVA to examine changes in these subfields and subregions. Although results revealed a significant main effect of ROI (F(1.766, 125.390)=602.143, p<0.001, $\eta^2$=0.895, Greenhouse-Geisser corrected; BF$_{10}$ >100, strong evidence for the inclusion of the main effect of ROI), we did not find any significant main effect of session (F(1,71) = 0.766, p=0.384, $\eta^2$<0.001, BF$_{10}$=0.047, strong evidence against the inclusion of the main effect of session), condition (F(2,71) = 0.131, p=0.878, $\eta^2$<0.001, BF$_{10}$=0.002, strong evidence against the inclusion of the main effect of session) and no significant interaction effect between session ×condition × ROI (F(3.21,110.806)=0.888, p=0.453, $\eta^2$<0.001, Greenhouse-Geisser corrected; BF$_{10}$ <0.001, extremely strong evidence against the inclusion of the main effect of session). Paired-sample t-tests confirmed that none of the three training groups exhibited significant changes from pre to post-test in any of the six MTL subregions (*Figure 3a*, ps >0.588, FDR corrected). These findings remained nonsignificant after controlling for sex and site as covariates (ps >0.559, FDR corrected).

We also analyzed gray matter volume changes outside of the medial temporal lobe using Free-Surfer (see Methods) to determine if any cortical brain areas might have been affected by the training. We applied a vertex-wise analysis of cortical volume, again finding no significant differences across the entire cortex (see Methods). This finding was further validated using the Destrieux atlas (*Destrieux et al., 2010*), which included 74 cortical parcellations per hemisphere (148 ROIs in total). Paired-sample t-tests revealed that none of the ROIs exhibited significant volume changes from pre- to post-test in any of the three groups (ps >0.697, FDR-corrected). These findings suggest that training did not result in any measurable cortical volumetric changes.

## Diffusion-weighted imaging (DWI) metrics did not change due to the training

For DWI, we separately evaluated free water (FW) and free water-corrected fractional anisotropy (fwcFA), as these measures capture different aspects of tissue microstructure. We also chose to analyze ROIs in gray matter (GM) and white matter (WM) separately, as these tissue types have dramatically different values for both FW and fwcFA, which involved eight GM ROIs and seven WM ROIs (see Methods). We conducted a mixed-design ANOVA with condition (Navigation/Verbal Memory/Video Control) as a between-subjects factor, with ROI and session (pre/post) as within-subjects factors. For the WM-fwcFA analysis, we observed a significant main effect of ROI (F(4.093, 233.279)=2340.878, p<0.001, Greenhouse-Geisser corrected, $\eta^2$=0.628, BF$_{10}$ >100, extremely strong evidence for the inclusion of the main effect of ROI). Yet, we found no main effect of session (F(1,57) < 0.001, p=0.981, $\eta^2$<0.001, BF$_{10}$=0.090, strong evidence against the inclusion of the main effect of session), no significant main effect of condition (F(2,57) = 0.025, p=0.975, $\eta^2$<0.001, BF$_{10}$=0.011, strong evidence against including the main effect of condition), and no significant interaction between session ×condition × ROI (F(6.924, 197.328)=1.954, p=0.064, $\eta^2$<0.001, BF$_{10}$ <0.001, strong evidence against including this interaction). Paired-sample t-tests confirmed that none of the three training groups exhibited significant changes from pre- to post-test in any of the seven ROIs (ps >0.083, FDR corrected). These findings remained nonsignificant after controlling for sex and site as covariates (ps >0.063, FDR corrected). Analyses involving WM-FW and GM regions (GM-fwcFA and GM-FW) were also not significant (see Appendix 1). These findings suggest that no DWI measures changed as a result of the training. Together, these findings suggest that neither navigation nor verbal memory

training for 20 hr across 10 sessions was sufficient to induce changes in macroscopic hippocampal volume or microstructure of relevant gray matter regions and white matter tracts.

## Improvements in the learning rate of the verbal memory transfer task, but not the navigation transfer task, correlated with both baseline total hippocampal volume and the baseline volume of the CA23DG subfield

We then performed an exploratory analysis to determine whether individual differences in hippocampal volume independent of the training might predict how well they learned from pre-test to post-test based on the different trainings. We conducted analyses employing more stringent corrections for multiple comparisons to examine the association between hippocampal volume and the change in learning rate from pre- to post-test based on the Verbal Memory and Navigation Transfer tasks. This analysis addresses the idea of whether those with larger hippocampi might improve more from pre- to post-test based on the training. Because there were no pre-post differences in hippocampal volume, we averaged the pre- and post-test hippocampal volume. We consider other behavioral correlations with baseline hippocampal volume in the Appendix material; none of these were significant (see Appendix 1).

For the Navigation Transfer task, the learning rate did not correlate with hippocampal volume, hemispheric volume, anterior and posterior hippocampus, or any hippocampal subfields, which was also not significant when controlling for sex and site in all three groups (Appendix 1 and *Appendix 2— table 5*). For the Verbal Memory Transfer task, the changes in learning rate did not correlate with hippocampal volume (r(23) = 0.367, p=0.071). When accounting for sex and site as covariates, however, we found a significant positive correlation (r(23) = 0.605, p=0.006, FDR corrected). This effect was specific to the Verbal Memory group; no significant correlation was identified for either the Navigation group (r(25) = 0.028, p=0.889) or the Video Control group (r(19) = –0.206, p=0.371); these findings remained consistent when controlling for sex and site as covariates (Navigation: r(25) = 0.038, p=0.858; Video Control: r(19) = –0.123, p=0.858, FDR corrected).

We further examined the correlation between learning rate and hippocampal subfield volumes after correcting for multiple comparisons (see Methods), specifically focusing on CA1 and CA23DG. Only the CA23DG subfield showed a positive correlation with the change in learning rate from pre- to post-test in the Verbal Memory Group (r(23) = 0.532, p=0.036, FDR corrected, *Figure 3c*), and this correlation persisted after controlling for sex and site as covariates (r(23) = 0.493, p=0.017). No significant correlations were observed for the CA23DG subfield in the Navigation group (r(25) = –0.083, p=0.953, FDR corrected) or the Video Control group (r(19) = –0.12, p=0.953, FDR corrected), regardless of whether sex and site were included as covariates. For correlations by hemisphere, anterior/posterior hippocampus and other subfields, please see Appendix 1 and *Appendix 2—table 4*. These findings suggested that individuals in the Verbal Memory group with larger CA23DG volumes exhibited greater improvement in verbal memory performance from pre- to post-test.

Fisher's z-tests revealed that the positive correlation in the Verbal Memory group was significantly stronger than that in the Navigation and Video Control group even after controlling for sex and site (Navigation: total hippocampus: Z=2.48, p=0.018; CA23DG: Z=1.972, p=0.036; Video Control group: total hippocampus: Z=2.596, p=0.015; CA23DG: Z=2.091, p=0.036; FDR corrected, one-side); there was no significant difference between the Navigation and Video Control group (total hippocampus: Z=–0.518, p=0.698; CA23DG: Z=–0.264, p=0.604; FDR corrected, one-sided). Together, these findings suggest that baseline hippocampal volume and CA23DG baseline subfield volume correlated with the change in learning rate on the Verbal Memory Transfer task following verbal memory training, but not after navigation training or video control. We highlight, however, that these analyses are exploratory and somewhat underpowered for between-subject comparisons.

## Task-related fMRI results

### Task-related informational connectivity analyses suggested network changes specific to the Navigation and Verbal Memory training groups

Although we did not observe changes in hippocampal volume as a result of training, it is likely that other brain-related changes might be mediating the improvements specific to the navigation and verbal memory training. We investigated this issue by collecting fMRI data both pre- and post-training

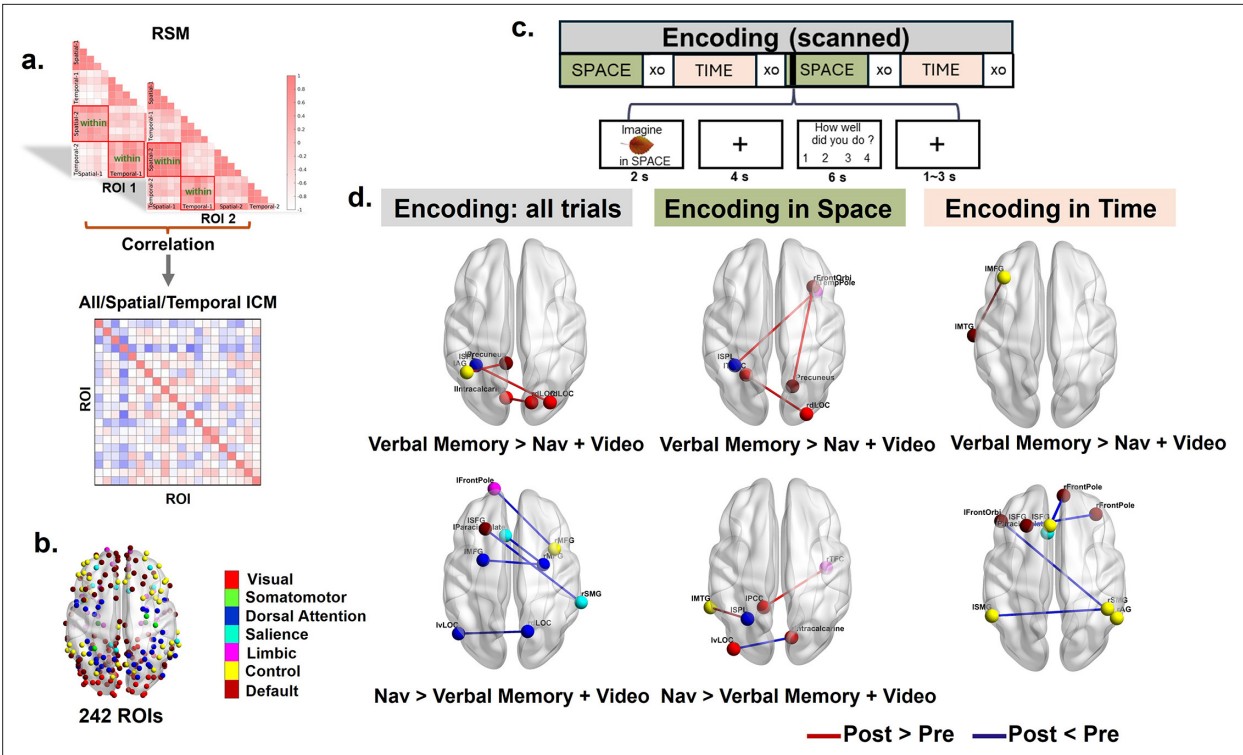

**Figure 4.** Task-related informational connectivity changes during encoding as a result of verbal memory and navigation interventions. (**a**) Representational similarity matrices (RSMs) that illustrate within-context correlations (spatial and temporal) for each of the 242 brain ROIs. Informational connectivity between 242 ROIs was derived by correlating the RSMs across regions within different contexts, resulting in three types of informational connectivity matrices (ICMs): the spatial ICM, the temporal ICM, and the combined ICM for all within-context trials. (**b**) Visualization of the 242 predefined ROIs, color-coded by functional networks. (**c**) Schematic of the experimental design for the source memory task during the encoding stage, highlighting tasks related to spatial and temporal contexts. (**d**) Differences in task-related informational connectivity during encoding (Verbal Memory (n=25)>Navigation (n=26)+Video (n=20); Navigation >Verbal Memory + Video) across training conditions. Results are shown for all trials, as well as separately for spatial and temporal encoding contexts. Red lines indicate regions with significantly increased connectivity (post >pre), while blue lines indicate regions with significantly decreased connectivity (post <pre). The top panels display results for the Verbal Memory group, while the bottom panels display results for the Navigation group. All Pearson correlation coefficients (r values) were Fisher-Z transformed prior to statistical analysis. Significance thresholds for informational connectivity were determined using 10,000 permutation simulations. The reported results have been corrected for multiple comparisons using the FDR method, with a q-value threshold of less than 0.05. Nav: Navigation.

during a source memory encoding (*Figure 4c*) and retrieval task (*Figure 5a*). During the encoding phase, participants learned objects by placing them within either an autobiographical timeline (temporal) or a familiar spatial layout (spatial). During the retrieval phase, they saw the objects again and indicated which of the 'sources' (spatial or temporal) they had placed the object in. Because we trained the Verbal Memory group to use the temporal method of loci but trained the Navigation group to learn spatial locations, we reasoned that training-specific strategies might lead to context-specific changes (i.e. depending on whether the object was encoded in time or space) that would be reflected in network changes during the source memory tasks. We performed a univariate analysis across the whole brain and an informational connectivity analysis using 242 pre-defined brain regions involved in spatial processing and memory (see Methods, *Figure 4b*). Informational connectivity involved correlating multivariate patterns between trials within each ROI and then correlating again between all 242 ROIs (*Figure 4a*). Informational connectivity could reflect changes in network configurations during source memory encoding and retrieval based on skills gained from training.

## Encoding phase: decreases in informational connectivity specific to the Navigation group and increases specific to the Verbal Memory group

We first considered both spatial and temporal source memory during the encoding phase, which could reveal how training affected the encoding of novel object-source pairings more broadly. For the

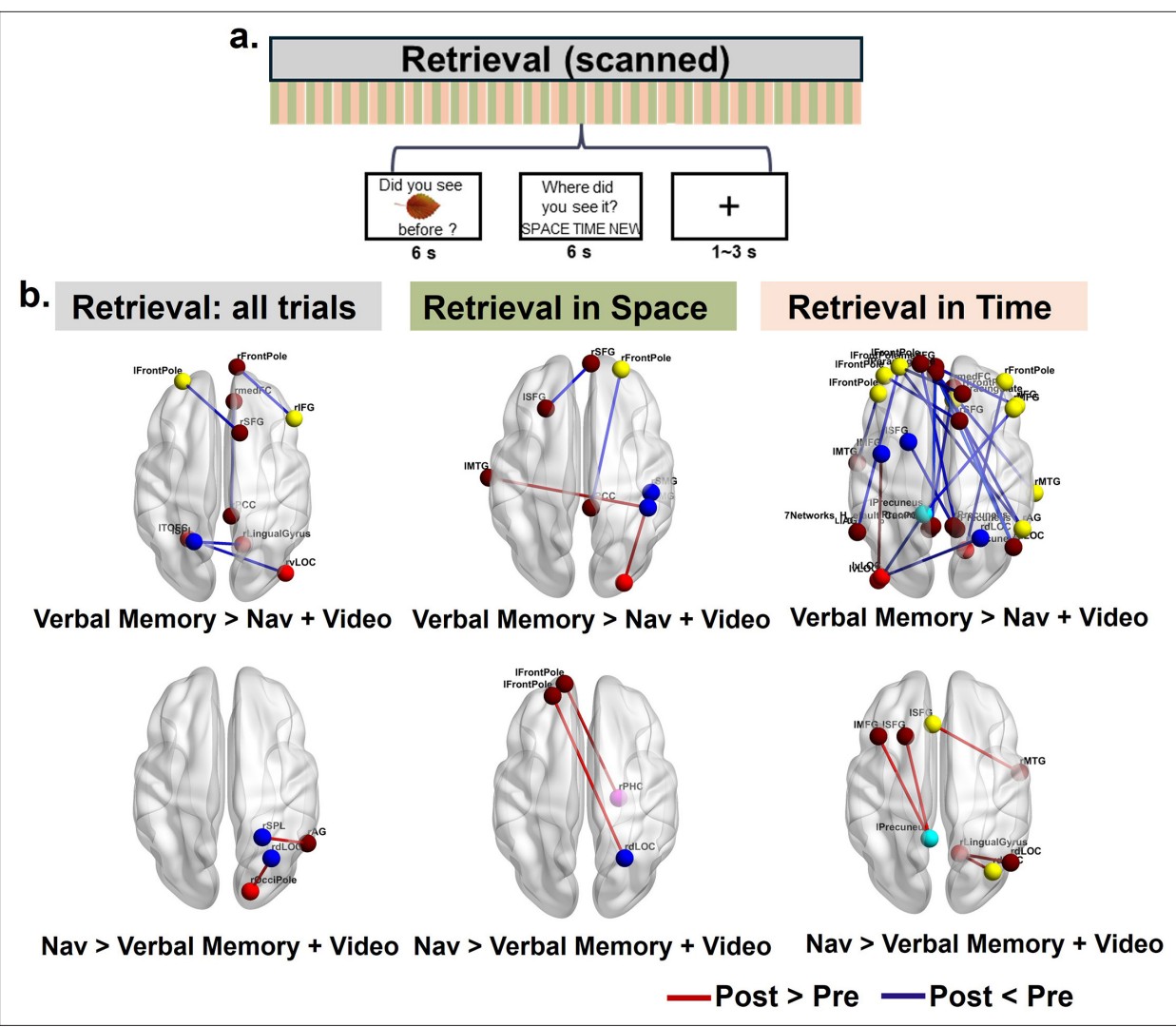

**Figure 5.** Task-related informational connectivity changes during source retrieval as a result of verbal memory and navigation interventions. (a) Schematic of the source memory task structure during retrieval. Participants were scanned during source retrieval, in which they identified whether an item was previously seen and determined its spatial or temporal context. (b) Differences in task-related informational connectivity during source retrieval (Verbal Memory (n=24)>Navigation (n=22)+Video (n=20); Navigation >Verbal Memory + Video) across training conditions. Results are shown for all trials, as well as separately for spatial and temporal encoding contexts. Red lines indicate regions with significantly increased connectivity (post >pre), while blue lines indicate regions with significantly decreased connectivity (post <pre). The top panels display results for the Verbal Memory group, while the bottom panels display results for the Navigation group. All Pearson correlation coefficients (r values) were Fisher-Z transformed prior to statistical analysis. Significance thresholds for informational connectivity were determined using 10,000 permutation simulations. The reported results have been corrected for multiple comparisons using the FDR method, with a q-value threshold of less than 0.05. Nav: Navigation.

univariate analyses, we found some changes from pre-test to post-test, but no interactions between session and training condition (see Appendix 1, *Appendix 2—table 16* and *Appendix 3—figures 6–8*). When considering spatial or temporal encoding separately, we also found changes from pre-test to post-test, but no interaction between session and training group (see Appendix 1, *Appendix 2— table 16* and *Appendix 3—figure 6*). All statistical analyses were corrected for multiple comparisons (see Methods).

Informational connectivity changes between pre-test and post-test were then compared across the three training groups. For the Verbal Memory group, we observed significantly increased informational connectivity at post-test compared to pre-test (i.e. post >pre), relative to the combined Navigation and Video Control groups (*Figure 4*). Specifically, we found enhanced connectivity between the left precuneus and left angular gyrus (t=3.94, p<0.05), the right lateral occipital cortex (LOC) and left intracalcarine (visual) cortex (t=3.63, p<0.05), and the right LOC and left superior parietal lobule (SPL)

(t=4.16, p<0.05). In contrast, no significant decreases in informational connectivity (i.e. post <pre) were observed. Considering spatial and temporal encoding separately, for the Verbal Memory group during spatial encoding, we found enhanced informational connectivity from pre-test to post-test (i.e. post >pre) between the frontal cortex and visual cortex (see Appendix 1 and *Figure 4d*). For temporal encoding, increased connectivity from pre- to post-test (i.e. post >pre) was observed between the frontal cortex and temporal gyrus (*Figure 4d*). No significant decreases in informational connectivity from pre-test to post-test (i.e. post <pre) were observed in the Verbal Memory group relative to the combined Navigation and Video Control groups during either spatial or temporal encoding (*Figure 4d*).

Conversely, for the Navigation group compared to the combined Verbal Memory and Video Control groups, we observed significantly decreased informational connectivity at post-test compared to pre-test (i.e. post <pre). Specifically, reduced connectivity was observed between the left paracingulate gyrus and right middle frontal gyrus (MFG) (t=3.80, p<0.05), left superior frontal gyrus (SFG) and right supramarginal gyrus (SMG) (t=5.14, p<0.05), left frontal pole and right MFG (t=3.85, p<0.05), left dorsal LOC and right ventral LOC (t=3.7, p<0.05) and right MFG and left MFG (t=3.79, p<0.05). In contrast, no significant increases in connectivity were observed from pre- to post-test (i.e. post >pre) for the Navigation group relative to the combined Verbal Memory and Video Control groups (*Figure 4d*, bottom-left). When considering spatial and temporal encoding separately, during spatial encoding, reduced connectivity was observed from pre-test to post-test (i.e. post <pre) between the primary visual cortex and higher-order visual areas, while increased connectivity was identified from pre-test to post-test (i.e. post >pre) between the parietal and temporal cortices. For temporal encoding, the Navigation group exhibited significant decreases in connectivity from pre-test to post-test (i.e. post <pre) across multiple regions, including the frontal and parietal cortices (see Appendix 1), with no significant increases in connectivity from pre-test to post-test (i.e. post >pre). No significant changes in task-related informational connectivity to the hippocampus survived after multiple comparisons correction for either navigation or verbal memory training. Broadly, these findings suggest changes in informational connectivity specific to each training group (i.e. decreases in informational connectivity from pre-test to post-test in the Navigation group and increases in the Verbal Memory group), with no changes specific to the hippocampus.

## Retrieval phase: increases in information connectivity specific to the Navigation group and decreases specific to the Verbal Memory group

For the univariate analyses for both spatial and temporal source memory retrieval, no significant changes were observed from pre-test to post-test, nor were there any interactions between session and training condition (see *Appendix 2—table 16*). When examining spatial and temporal retrieval separately, we similarly found no significant changes from pre-test to post-test and no interactions between session and training group (see *Appendix 2—table 16*).

We next examined informational connectivity changes from pre-test to post-test during the source memory retrieval stage (*Figure 5a*), finding a pattern somewhat opposite to that seen during encoding. In the Verbal Memory group, we observed significantly decreased informational connectivity in post-test compared to pre-test (i.e. post <pre) relative to the combined Navigation and Video Control groups. These reductions were identified between the right inferior frontal gyrus (IFG) and right frontal pole (t=4.01, p<0.05), left frontal pole and right SFG (t=4.48, p<0.05), right medial frontal gyrus and right posterior cingulate cortex (PCC) (t=4.42, p<0.05), right ventral LOC and left temporo-occipital fusiform cortex (TOFC) (t=4.30, p<0.05), and right lingual gyrus and left SPL (t=3.82, p<0.05). No significant increases in connectivity from pre-test to post-test (i.e. post >pre) were observed in the Verbal Memory group relative to the combined Navigation and Video Control groups (*Figure 5b*, top-left). Considering spatial and temporal source retrieval separately, spatial retrieval showed decreased connectivity from pre- to post-test (i.e. post <pre) between frontal regions (e.g. left and right superior frontal gyri) and between the right frontal pole and PCC, and increased connectivity from pre to post (i.e. post >pre) between temporal and parietal regions (e.g. left middle temporal gyrus [LMTG] and right SMG) and between the visual cortex and right SMG. For temporal retrieval, decreased connectivity from pre-test to post-test (i.e. post <pre) was prominent across multiple regions, including the frontal pole, angular gyrus, precuneus, and lateral occipital cortices. One increase specific to the

Verbal Memory group was observed from pre-test to post-test (i.e. post >pre); however, in the left ventral lateral occipital cortex and left MFG (*Figure 5b*, see details in Appendix 1).

In the Navigation group during retrieval, we found significantly increased informational connectivity in the post-test compared to the pre-test (i.e. post >pre) relative to the combined Verbal Memory and Video Control groups. These increases were found between the right occipital pole and dorsal LOC (t=4.27, p<0.05), and the right SPL and right angular gyrus (AG) (t=4.09, p<0.05). Considering spatial and temporal retrieval separately, spatial retrieval showed enhanced connectivity from pre-test to post-test (i.e. post >pre) between regions such as the left frontal pole and the occipital and parahippocampal cortices. Temporal retrieval revealed increases from pre- to post-test (i.e. post >pre) between the precuneus and SFG, lingual gyrus and occipital gyrus, and SFG and MTG. No significant decreases in connectivity were observed from pre-test to post-test (i.e. post <pre) in the Navigation group relative to the combined Verbal Memory and Video Control groups (*Figure 5b*, bottom).

Again, no significant changes in task-related informational connectivity during source retrieval were present to the hippocampus when controlling for multiple comparisons. To summarize, verbal memory training resulted in decreased connectivity during retrieval across frontal, parietal, and occipital regions, especially during temporal retrieval, with limited increases. In contrast, navigation training consistently increased connectivity for both spatial and temporal retrieval, particularly between frontal, occipital, parietal, and temporal regions, with no significant decreases observed. These findings suggest that verbal memory primarily alters network connectivity during temporal contextual retrieval, whereas navigation training enhances connectivity more broadly across both contexts.

## Specific changes in network-wide informational connectivity pattern distance during retrieval: Enhanced due to verbal memory but not navigation training

The above analysis focused on changes in informational connectivity between pre-test and post-test for individual connectivity edges between pairs of ROIs. Next, we examined changes in multivariate informational connectivity patterns, which we defined as the connectivity between a single ROI, and the remaining 241 ROIs. This analysis helped address network-wide changes in connectivity rather than those restricted to two different ROI pairs. Multivariate informational connectivity patterns were extracted separately for spatial and temporal informational connectivity matrices (ICMs), and the distance between the two patterns was computed using 1 r (Pearson's, see Methods).

While no significant changes in pattern distance were observed during the encoding stage for any of the three conditions, a greater distance between spatial and temporal patterns in the left SFG network connectivity during source retrieval was found in the Verbal Memory group at post-test compared to pre-test (t(23) = –4.275, p=0.043, FDR corrected; *Figure 6b*). This effect was not observed in the Navigation and Video Control groups (*Figure 6c*). A similar result was obtained using the Spearman correlation to compute the distance, with a greater distance in the left SFG for the Verbal Memory group at post-test relative to pre-test (t(23) = –4.400, p=0.050, FDR corrected). No

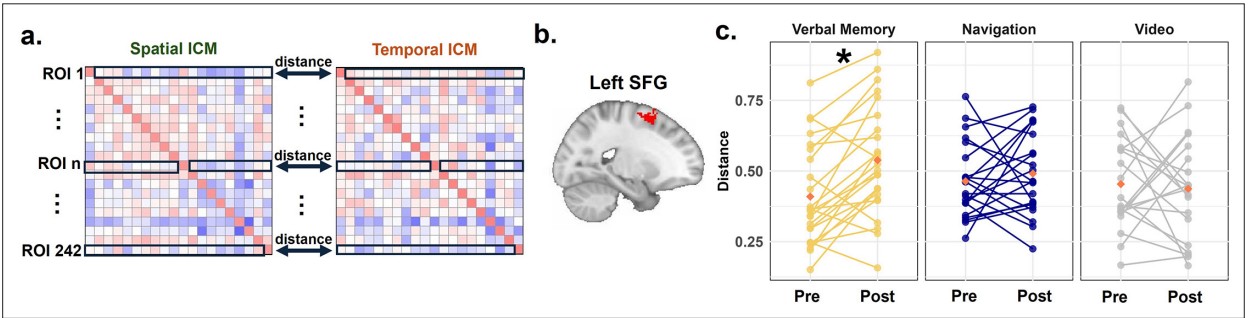

**Figure 6.** Multivariate informational connectivity pattern distance between pre-test and post-test. (**a**) Schematic of multivariate distance analysis. For each ROI, informational connectivity between the current ROI and the remaining 214 ROIs was extracted from both spatial and temporal ICMs, and distances were calculated as 1 minus the Pearson or Spearman correlation coefficient. (**b**) and (**c**) Among all 242 ROIs, only the SFG showed a greater distance between spatial and temporal ICMs in post-test compared to pre-test in the Verbal Memory (n=24) group, but not in the Navigation (n=22) or Video Control (n=20) groups. ICM: informational connectivity matrix. Paired t-tests were conducted to evaluate statistical differences between pre and post. *p<0.05, FDR corrected.

significant differences were found in the Navigation or Video Control groups. These findings also suggest some network-wide changes that are specific to verbal memory training.

## Discussion

Acquiring complex cognitive skills is a fundamental human capability, but we have little knowledge about the neural changes that accompany such new learning. As suggested in past work, we are capable of remarkable improvements in performance across a wide range of domains, yet the neural basis for these changes remains unclear. Here, we build on expert training regimens in verbal memory (e.g. the method of loci) and spatial navigation (e.g. London taxi driver training) by employing a within-subject intervention approach to determine the neural correlates of training two related but distinct complex skills: navigation and verbal memory. Our results show that non-expert young adults substantially improved in the task they were trained on and exhibited some limited near transfer (to new lists of words or new environments). Unlike the neuroimaging literature on experts or impaired populations (like older adults or individuals with dementia who cannot perform these tasks as well), we found no evidence that training modified brain structure, as measured by hippocampal volume and DWI microstructure in relevant GM regions and WM tracts. Rather, analyses of task-based functional imaging data showed training-specific differences in informational connectivity, a neural metric that indexes how similar representations are across brain regions (*Mooraj et al., 2025*). Our findings have implications for both basic research and clinical translation. From a basic research perspective, our results suggest that structural changes in gray and white matter are relatively insensitive to cognitive intervention but are instead reflected in more transient changes to network-based connectivity patterns. From a clinical-translational perspective, our results suggest promise in training new complex cognitive skills such as navigation and verbal memory, although the efficacy of our specific training regimen in clinical populations remains to be tested.

Our study involved two novel and extensive in-lab training regimens, which shared key features but allowed for dissociable cognitive training. While previous training studies have examined network configurations before and after participants learned the method of loci or evaluated navigation training, these experiments often involved expert groups (memory athletes or London taxi drivers; *Maguire et al., 2000*; *Maguire et al., 2003*), or older adults (*Engvig et al., 2010*). In our experiment, we evaluated healthy young adults while carefully controlling the duration and nature of the verbal memory and navigation trainings to match the intensity and the degree to which the skill was scaffolded. In this way, one training serves as a control for the other, while also allowing us to compare with an active control group who did not gain a complex skill (video control) but was passively exposed to similar material.

Both training regimens were effective: participants in the Verbal Memory group were able to learn longer word lists, while participants in the Navigation group were able to reduce the length of their paths and required fewer trials to learn landmarks in new subsections of the environment. Yet, it was difficult to ascertain whether either training generalized beyond immediate transfer. Our near transfer measures exhibited strong individual differences at baseline, which were difficult to overcome with 10 sessions of training. Both near transfer measures also likely had practice and ceiling effects, which dampened our ability to measure the effects of training (e.g. *Appendix 3—figure 2*). Exploratory analyses, however, revealed training-specific differences in the learning rate on the transfer tasks from pre- to post-test. In addition to good face validity as a match for the skills we were training (that is, for example, in the navigation training, participants learned how to quickly acquire spatial information about the whole environment after only 10 trials compared to those who did not undergo that training), the change in learning rate from pre- to post-test was also the only dependent measure we found to correlate with baseline hippocampal volume. These findings, though exploratory, suggest that baseline hippocampal volume may enable more receptiveness to training, aligning with concepts like neural reserve (*Stern, 2012*). In addition to replicating these findings, future research using these training regimens should develop a more comprehensive near-transfer measure and evaluate far transfer to completely untrained tasks.

Our navigation training is particularly notable in that it partially mimics the training undertaken by London taxi drivers, who first learn individual subsections of an environment before connecting these subsections to one another. While other virtual navigation trainings exist (*Lövdén et al., 2012*; *McLaren-Gradinaru et al., 2020*), their environments are often small, simplified, or non-hierarchical.

Thus, a key contribution of this work is the development of a navigation training environment that is suitable for training wayfinding skills. We also developed a novel verbal memory training paradigm that has not been reported previously in the literature. Using this training, we were able to dramatically increase the number of freely recalled words of the average participant, with the typical participant showing an improvement from about 25–30 words freely recalled (repeated) words on day 1 to 60–65 words after only five days of training.

Despite the efficacy of the training, neither the Navigation nor the Verbal Memory group showed macroscopic neural structural differences in the hippocampus, which has been linked to verbal memory and/or navigation function in past studies (*Brunec et al., 2018*; *Burgess et al., 2002*; *Maguire et al., 2000*). These results supply causal evidence supporting the notion that hippocampal structural changes may not impart superior navigation or verbal memory skills. Although previous work has shown effects of training, to our knowledge, this work has occurred in specialized populations or involved between-subject comparisons. For example, London taxi drivers who undergo extensive training to learn the city of London show increased hippocampal volume from pre- to post-training compared to taxi drivers who fail to complete the training (*Maguire et al., 2000*). Older adults who undergo an exercise-only training or a navigation + exercise training show reduced hippocampal atrophy (i.e., less volume loss) compared to a no-contact control (*Lövdén et al., 2012*). Here, in healthy younger adults, who nevertheless showed robust individual differences in navigation and verbal memory and showed strong evidence of improvement across both types of training, there were no observed structural changes to hippocampal gray or white matter tracts, nor were there any significant structural changes to other brain regions.

In exploratory analyses, informational connectivity analyses on fMRI data from a spatial and temporal source memory task revealed unique network reconfiguration signatures as a result of navigation and verbal memory training. These changes were based on a strict correction for multiple comparisons and suggest that verbal memory and navigation training results in robust changes in task-related functional connectivity patterns. Interestingly, the vast majority of these changes were to areas outside of the medial temporal lobe, an area often considered central to both navigation and memory. During encoding, we observed increases in connectivity due to the verbal memory training primarily in the frontal and occipital lobes. Conversely, we observed decreases in informational connectivity in some of these same areas as a result of navigation training. We observed somewhat of the opposite pattern during retrieval: increases in informational connectivity during navigation and decreases during verbal memory training. Again, many of these connectivity changes were areas within the frontal cortex, with some longer-range changes to parietal, parahippocampal, and occipital cortex. Finally, a multivariate approach centered around superior frontal gyrus as a hub revealed differences in network connectivity patterns for spatial compared to temporal retrieval, suggesting task-related network-level changes due to the training as well. Similarly, we note that two studies, which trained participants on an n-back task, also observed changes in dynamic functional connectivity in frontal and parietal cortices (*Bassett et al., 2015*; *Finc et al., 2020*).

Our results suggest that the networks most malleable during acquisition of complex cognitive skills largely involve areas related to cognitive control, with one possibility being that such changes allow modulation of existing networks that include the medial temporal lobe (*Schedlbauer and Ekstrom, 2019*). At the same time, navigation and verbal memory training resulted in differential changes within these frontal-parietal networks, suggesting that such cognitive skills involve at least partially dissociable changes. While it is reasonable to conjecture that microstructural changes in brain tissue (as measured with DWI) or even macrostructural gray matter changes may occur after considerable expertise is developed, such structural reorganization would likely be preceded by network changes in functional connectivity. In this scenario, functional changes that reflect training effects could potentially become solidified through long-term maintenance or the development of domain-specific expertise.

## Methods

### Key resources table

| Reagent type (species) or resource | Designation | Source or reference | Identifiers | Additional information |
|---|---|---|---|---|
| Software, algorithm | PsychoPy v2022.2 | https://www.psychopy.org | RRID:SCR_006571 | Used for behavioral task presentation |

*Continued on next page*

*Continued*

| Reagent type (species) or resource | Designation | Source or reference | Identifiers | Additional information |
|---|---|---|---|---|
| Software, algorithm | Unity 3D (2021) | https://unity.com | | Used to develop the virtual navigation environment |
| Software, algorithm | FSL v5.0.1 | https://fsl.fmrib.ox.ac.uk | RRID:SCR_002823 | Used for preprocessing and registration of MRI data |
| Software, algorithm | Advanced Normalization Tools | https://stnava.github.io/ANTs/ | RRID:SCR_004757 | Used for registration fMRI and DWI data |
| Software, algorithm | MRtrix3 | https://www.mrtrix.org/ | RRID:SCR_024123 | Used for DWI data analysis |
| Software, algorithm | Automatic hippocampal subfield segmentation | https://sites.google.com/view/ashs-dox/home | RRID:SCR_005996 | Used for hippocampus segmentation |
| Software, algorithm | FreeSurfer v7.4.1 | https://surfer.nmr.mgh.harvard.edu | RRID:SCR_001847 | Used for cortical reconstruction and volumetric segmentation |
| Software, algorithm | MATLAB | https://www.mathworks.com/products/matlab.html | RRID:SCR_001622 | Used for custom scripts for behavioral and fMRI data analysis |
| Software, algorithm | R | https://www.r-project.org/ | RRID:SCR_001905 | Used for custom scripts for figures and statistical analysis |

## Preregistration and power analysis

This study was preregistered on the Open Science Framework (OSF; https://osf.io/etxvj). The preregistration was completed on October 30, 2023, after approximately 80% of data collection had been completed, but prior to any analysis of the primary outcome variables. The preregistration outlines the study hypotheses, design, target sample size, and planned behavioral and neuroimaging analyses, including longitudinal ROI comparisons and statistical correction procedures.

A priori power analysis was conducted using G*Power 3.1 to estimate the required sample size for detecting a Group ×Time interaction in a mixed-design ANOVA. Assuming a small-to-medium effect size ($f$=0.35), we determined that 24 participants per group would provide 80% power to detect a significant effect at $\alpha$=0.05. To allow for potential attrition and data exclusion (e.g., due to excessive motion or incomplete datasets), we targeted recruitment of 30 participants per group across two study sites.

All primary hypotheses, analytic plans, and inference criteria are documented in the preregistration. Exploratory analyses are clearly delineated in both the preregistration and the present manuscript.

## Participants

83 participants between the ages of 18–45 were recruited for enrollment in the study across two sites (University of Florida and University of Arizona). Participants were recruited through advertisements and flyers posted on campus. After recruitment, participants were screened for: (1) being right-handed; (2) having no metal in their bodies or other MRI contraindications; (3) having no COVID-19 symptoms. Participants also completed and passed an MRI prescreen before fMRI scans. All participants provided written informed consent. The study was approved by the University of Arizona Institutional Review Board (Primary Reviewing IRB: 1805589537) and was ceded to the University of Florida Institutional Review Board (Ceded IRB: CED000000489). This study included 75 participants (*Appendix 2—table 16*) who fully completed the entire study: 27 in the verbal memory condition, 27 in the navigation condition, and 21 in the video control condition. One verbal memory participant, despite completing training, was excluded from MRI scanning due to ear piercings, resulting in 26 participants for brain structure and connectivity analyses. Some participants were excluded from certain tests due to outlier performance, excessive head movement during scanning, or missing data. For detailed sample size information for each test, please refer to *Appendix 2—table 2*.

## Overview

Participants were recruited via flyers or word of mouth. During prescreening, each participant was randomly assigned to one of three training conditions (Navigation, Verbal Memory, or Video Control). Participants then completed a prescreening phone call for MRI eligibility, COVID safety, and availability to determine participation in the study. They then received instructions for their first session verbally and via email. The study took place over 12 sessions, each of which occurred on a separate day. These 12 sessions took place over the course of 3–4 weeks (average duration = 4.116 weeks).

The first and last sessions were 3 hr sessions, referred to as pre-test and post-test, and consisted of behavioral and neuroimaging measures. These sessions were identical for all participants, regardless of which condition they were assigned to. Following the pre-test session, participants completed ten 2 hr training sessions, during which they completed training based on their assigned condition. Following training, participants completed the post-test session, were debriefed, and released. Participants were paid for their participation. See *Appendix 2—table 3* for more details about conditions and scheduling.

## Training conditions
### Navigation training

Participants assigned to the Navigation group navigated a virtual environment in Unity 3D called 'Virtual Arida' using an Xbox game controller on a desktop computer. Virtual Arida has six individual subsections (A, B, C, D, E, F), with each containing eight different target locations. Participants began training on day 1 in subsection A and were spawned in front of a target location. They were then asked to navigate using the shortest possible route to a second target location. If the participants did not perform a sufficiently direct route, they repeated the route; their route was determined to be sufficiently direct if it did not exceed 120% of the shortest possible path between two buildings. Once they successfully employed a sufficiently direct route, they moved on to the next route. Participants learned all 28 possible routes in the subsection. When they learned all routes successfully, they completed random target location pairings for the remainder of the 2 hr session. After completing a subsection, participants moved on to a new subsection. In some cases, the new 'subsection' involved navigating between two previously learned subsections. For example, once individual subsections A and B were completed, participants navigated between targets in subsection A and subsection B. This involved 'integrating' their previously learned knowledge from prior subsections (A+B). Participants then learned the next individual subsection (subsection C), and after that, the next integrated subsection A+B+C. This continued in this manner through the final integrated subsection A+B+C+D+E+F (for those participants that reached this far).

## Training condition
### Verbal memory training

Participants placed in the verbal memory condition completed free recall training on a desktop computer using a task built with Unity 3D. Similar to the verbal memory transfer task during pre/post-test, in this task participants were shown a list of words from the Toronto word pool one at a time, which they memorized using a trained strategy (the 'temporal' method of loci)(*Bouffard et al., 2018*). After being shown the list of words, they completed a distractor task by typing the numeric answer for five simple addition/subtraction questions. Then, they recalled the words by typing them into the provided space. The strategy that participants were given to memorize the words was a modified method of loci termed the 'temporal method of loci' (*Bouffard et al., 2018*). At the beginning of Day 1 of training, participants wrote 30 of their most memorable, vivid, and non-traumatic memories from their life thus far. They were asked to order these memories into a mental timeline to help them with the word recall task. Participants were asked to associate each word they were given with a memory from their mental timeline. Then, when they reached the recall portion of the task, participants were to use their mental timeline memories to help them recall the associated words in the current list. The number of words increased with each successful trial. If there were multiple unsuccessful trials, the number of words decreased. Once they were successful with this shorter list, the previous list was repeated. For example, if the list was 'red', 'orange', 'yellow', and the participant did not recall all three words, the next trial's list would be 'red', 'orange'. On successful completion, the next trial would be 'red', 'orange', 'yellow', 'blue'.

### Sessions 1-5 of verbal memory training (same words list)

For the first five sessions of training, participants recalled a list that increased in length on each trial when they successfully recalled all the words. This was done using an algorithm based on a power law learning function. For example, they could start with the words: 'Jersey', 'rifle', 'pony'. Upon successful completion of this list, the list would grow to 'Jersey', 'rifle', 'pony', 'castle' (*Figure 1a*).

### Sessions 6–10 of verbal memory training (new word list)

For the final five sessions of training, once participants successfully recalled a list, they learned a completely new list of words that was increased in length in the same manner as described above. For example, they could start with the words: 'sister', 'soldier', 'beauty'. Upon successful recall of this list, the next trial, the next list could be 'mother', 'genius', 'number', 'sailor' (*Figure 1b*).

### Active control condition - video watching

Participants in this condition watched a wide range of videos on a desktop computer. The types of videos varied, but included interviews, lectures, documentaries, shows, and YouTube videos covering different aspects of memory and navigation. Participants were given a set of questions for each video to ensure that they paid attention. Videos and quiz questions were administered using Qualtrics.

### Pre-test overview

In the Pre-test session, participants provided informed consent, completed a demographics question-naire, and confirmed their MRI prescreen information. Participants then completed pre-test behavioral measures, which included a navigation transfer task (Virtual Silcton), a verbal memory transfer task (a new set of words from the Toronto word pool), and an attention task (Posner cueing task). Next, the experimenter described the source memory task that would take place during the fMRI scan. The participants created the temporal and spatial contexts for this task, then after a short break to prepare for the scanner, they completed the source memory task in the scanner (see details in Appendix 1 and *Appendix 2—table 3*).

### Post-test overview

The post-test session was similar to the pre-test session. The only changes in the post-test session were different words for the verbal memory transfer task, a different section of Virtual Silcton, and different images for the source memory task. After reviewing the MRI prescreening form for any changes, participants completed the post-test measures (navigation, verbal memory, and attention tasks). Participants then reviewed the spatial and temporal contexts created in the pre-test session, took a short break, and completed the source memory task in the scanner. Finally, participants completed a debriefing survey and received payment (see details in Appendix 1 and *Appendix 2—table 2*).

## Pre-post tests

### Navigation transfer task: Virtual Silcton

The pre-test and post-test navigation transfer task was completed on a laptop computer with an Xbox game controller using the 'Virtual Silcton' environment recreated in Unity 3D (*Weisberg et al., 2014*) (http://www.virtualsilcton.com/). Participants were instructed to freely navigate from their starting location to one of the eight possible destination buildings within the environment. If the participant walked for more than 60 s during a trial, a compass appeared at the bottom of the screen to point them to the target building. Over the trials, participants learned the spatial environment from the routes taken between the buildings. Participants completed trials until all routes were completed, or until 30 min elapsed. After the learning phase, participants completed an on-site pointing task, in which they were placed next to one of the eight buildings and had to point to each of the other seven in a random order. Next, they completed a model-building task, in which they dragged and dropped images of the eight buildings around an on-screen box. These tasks demonstrate configural knowledge of the environment.

### Verbal memory transfer task

Participants memorized and recalled a list of 12 words from the Toronto Word pool. The structure of the task was the same as that for the verbal memory training task, although it was programmed in PsychoPy. All participants had a maximum of five attempts to recall all 12 of the words correctly.

### Source memory task

Prior to entering the scanner, participants were asked to generate a *temporal* and a *spatial* context to use during the source memory task. For their temporal context, they were asked to create a list of

10–15 salient memories of events from their life. We then asked them to write these down on a piece of paper and arranged them into a mental timeline of life events. For their spatial context, participants drew a blueprint of their living space (e.g. dorm room or apartment) on a second piece of paper. Participants were given time to review their contexts to ensure they knew them well enough prior to entering the scanner. They were then instructed on how they would use these contexts to recall objects. They were provided with an example of how to use each context and practiced verbally by using these contexts verbally.

Participants underwent functional MRI scanning while performing the source memory task. This task was comprised of two phases, each with four runs. During the encoding phase (runs 1–4), participants were presented with images and instructed to encode them within either a temporal or spatial context, consistent with prior out-of-scanner practice. During the retrieval phase (runs 5–8), participants were presented with images and asked to make two judgments: first, whether the image was new, and second, if recognized, the context in which it had been previously encoded. All participants viewed the same images, but in counterbalanced orders.

### Source memory encoding

This source memory task included four encoding and four retrieval runs. In the encoding task, participants viewed an image and were asked to imagine it in either 'time' or 'space', referring to their previously generated temporal or spatial contexts. Participants were to either visualize the image as a part of their mental timeline if it was to be imagined in 'time' or visualize this object existing in their living space if it was indicated that it was to be imagined in 'space'. Following this visualization period, participants were asked to indicate how vividly they visualized this item on a Likert scale by pressing 1–4 on the button box (see image below). All trials were jittered. At the end of the block, participants completed a short odd/even task. In this 60 s task, an 'X' or an 'O' would be displayed. Participants pressed '1' if it was an 'X' and '2' if it was an 'O'. This served as a baseline task for hippocampal activation (*Squire et al., 2004*).

### Source memory retrieval

In the retrieval task, participants were shown an image of an object and asked whether they recognized it from the encoding phase or if it was a new image by pressing '1' for recognition and '2' for new. They were then asked if this image was previously viewed in 'time', or 'space', or if it was new by pressing '1', '2', or '3', respectively. Four retrieval runs were completed. All trials were jittered. Detailed descriptions of other pre-post tests are provided in Appendix 1.

### Navigation pointing task: Virtual Silcton

Using the Virtual Silcton website (http://www.virtualsilcton.com/), participants were placed at the goal location in front of each of the 8 buildings within the Virtual Silcton environment. Participants were not capable of translation in this portion of the task, but fully capable of rotation. Participants were told to point the target on the screen in the direction of the other seven target buildings, then click the mouse to record their response (see *Appendix 3—figure 1a*).

### Navigation model building task: Virtual Silcton

Continuing on the Virtual Silcton website, participants were presented with a blank box and instructed that this box represents the boundaries of the Virtual Silcton environment. To the right of the square, they were shown bird's-eye view images of the 8 destinations from the Navigation Task. Using the mouse, they dragged and dropped each building into the blank square to create a map of how they recalled the virtual environment (see *Appendix 3—figure 1b*). Once finished, participants clicked a button that indicated that they had completed the task.

### Attention task

Participants completed a Posner cueing task (*Posner, 1980*) in which they were told to respond to an appearance of a star that would either appear to the left or to the right of a fixation cross through valid and invalid highlighting cues. Participants were instructed to be accurate, but quick. This task was completed via the Testable website (https://www.testable.org/project).

## Behavioral data analysis

### Navigation task (navigation training in Virtual Arida)

To quantify the training effect, we analyzed the average distance error traveled, normalized by the threshold (120% of the length of the shortest possible path). We removed all trials that occurred after all unique pairs had been learned. Because we controlled the amount of time per task (2 hr per session), not all participants progressed to the same extent. Repeated-measures ANOVAs were conducted with subsection or integration as the within-subject factors and threshold-normalized distance error as the dependent measure.

### Verbal memory task (verbal memory training with the modified method of loci)

To quantify the training effect, we performed linear regression analysis on the maximum number of correctly recalled words per day. Regression slopes were calculated separately for the first 5 and last 5 days (*Figure 1a* right), reflecting the distinct training paradigms implemented during these two phases (see Methods).

### Learning rate in the navigation transfer task (Virtual Silcton) during the pre-/post-test

To assess navigation performance in the Navigation Transfer task, for each trial, we calculated distance error by subtracting the optimal distance from the participant's actual navigated distance between their starting location and the target building. Since distance errors are inherently influenced by the distance between the starting location and the target building, we normalized these errors by dividing each distance error by the corresponding optimal distance. The task consisted of 20 trials, during which participants progressively learned the environment. As expected, normalized distance errors decreased over time, indicating improved navigation performance in later trials. To quantify this learning effect, we calculated a learning rate, defined as the negative of the difference between the normalized distance errors of the first 10 trials and the last 10 trials, divided by their sum (see *Equation 1*). A higher/positive learning rate suggests relatively stable and consistent performance across early and late trials, whereas a lower/negative learning rate reflects a greater difference in performance between early and late trials.

$$Learning\ Rate = -\left( \frac{Distance\ errors\ of\ first\ 10\ trials\ -\ Distance\ errors\ of\ last\ 10\ trials}{Distance\ errors\ of\ first\ 10\ trials\ +\ Distance\ errors\ of\ last\ 10\ trials} \right) \quad (1)$$

### Learning rate in the verbal memory transfer task during the pre-/post-test

For the Verbal Memory Transfer task, we calculated a learning rate to quantify memory performance. To determine the learning rate, we first calculated the difference in the number of words correctly recalled between the first and last trials. This difference was then divided by the total number of trials to yield the learning rate (see *Equation 2*).

$$Learning\ Rate = \frac{Last\ trial\ -\ First\ trial}{number\ of\ trials} \quad (2)$$

### Correlation analysis

To examine the relationship between learning rate of either task and brain structure (volume/DWI, see below), we collapsed the session factor by averaging values from pre- and post-training sessions for each ROI, given the absence of significant session effects or interactions between session and training condition. We conducted Pearson correlation analyses (parametric) for data that met the normality assumption based on the Shapiro-Wilk test. If normality was violated, Spearman correlation analysis (nonparametric) was used instead. Sex and site were included as covariates in the corresponding partial correlation analyses.

## fMRI data analysis

### MRI data acquisition

Scanning was performed using a 32-channel 3T Siemens 'Skyra' scanner at the University of Arizona and 32-channel 3T Siemens 'Prisma' scanner at the University of Florida. Visual stimuli were presented on a screen positioned behind the scanner and viewed by participants through a mirror attached to the head coil. Stimuli and responses were presented and collected using PsychoPy (https://www.psychopy.org) running on a Windows 10 laptop. High-resolution anatomical images of the hippocampus and surrounding cortex were acquired with a T2-weighted turbo-spin echo (TSE) anatomical sequence (FOV = 200 mm × 200 mm, matrix = 448 × 448, TR = 4200.0 ms, TE = 93.0 ms, flip angle = 139 degree, slice thickness = 1.9 mm, 28 slices, bandwidth=199 Hz/pixel). High-resolution structural images of the whole brain were obtained using a 3D, T1-weighted MPRAGE (1 mm$^3$ isotropic) sequence (FOV = 256 mm, matrix = 256 × 256, slice thickness = 1 mm, TR = 2300 ms, TE = 2.41 ms, flip angle = 8 degree, bandwidth =330 Hz/pixel). Functional images were acquired using a whole-brain echo planar imaging (EPI) sequence (TR = 1560 ms, TE = 30 ms, flip angle = 70 degree, field of view (FOV) = 220 mm, matrix = 88 ×88, slice thickness = 2.5 mm, slices = 48, bandwidth = 2030 Hz/pixel), involving a voxel resolution of 2.5 × 2.5 × 2.5 mm. Diffusion Weighted Imaging (DWI) data were acquired using two sequences with opposite phase-encoding directions to correct for distortion artifacts without signal loss. Acquisition parameters included: TR = 9200 ms, TE = 86 ms, FOV = 256 mm, 30 diffusion directions, 60 slices with a thickness of 2.0 mm, and a voxel size of 2.0 × 2.0 × 2.0 mm. The diffusion-weighted images were obtained using an echo plane sequence with the following parameters: number of b0 images = 1, b-value=1000 s/mm$^2$, number of directions = 30, TR = 9200 ms, TE = 86 ms, and voxel size: 2 × 2 × 2 mm. High-resolution resting state images were acquired using another whole-brain EPI sequence (TR = 3000ms, TE = 36 ms, flip angle = 90 degree, field of view (FOV) = 240 mm, matrix = 160 ×160, slice thickness = 2.5 mm, slices = 48, bandwidth = 802 Hz/pixel), involving a voxel resolution of 1.5 × 1.5 × 2.5 mm. Resting state data were not considered in this manuscript.

### DWI data preprocessing

We performed pre-processing of DWI data using a customized pipeline that combines tools from the fMRIB Software Library 6.0 (*Jenkinson et al., 2012*), Advanced Normalization Tools (*Tustison et al., 2021*), and MRtrix3 (*Tournier et al., 2019*). First, we removed Gaussian noise present in the DWI data by fitting a Marchenko-Pastor distribution to the signal matrices to generate a threshold for PCA denoising (*Cordero-Grande et al., 2019*; *Veraart et al., 2016a*; *Veraart et al., 2016b*). We then removed Gibbs-ringing artifacts that can occur at tissue borders such as the outer surface of the brain and near ventricles (*Tournier et al., 2019*). Next, we generated brain masks for the DWI images using the dwi2mask function from MRtrix3, which uses information from both diffusion-weighted and non-diffusion weighted (b=0) volumes to generate an accurate brain mask. We then corrected for Eddy current and movement-related distortions using FSL Eddy (*Andersson et al., 2016*; *Bastiani et al., 2019*; *Smith et al., 2004*). Using the root mean square motion output provided by Eddy, we then applied a motion threshold, excluding participants with >2 mm absolute displacement or >0.5 mm relative displacement between diffusion weighted volumes. Application of this motion threshold at both timepoints resulted in 20 participants in the Verbal Memory group, 22 participants in the Navigation group, and 18 participants in the Video control group in final DWI analysis.

Consistent with prior work (*Mitchell et al., 2022*; *Pasternak et al., 2009*; *Shin et al., 2025*; *Wilkes et al., 2024*; *Wilkes et al., 2023*), we reconstructed the DWI data using a bi-tensor model, in which freely diffusing water is modeled by one tensor and anisotropic water diffusion is modeled with a separate tensor after removing the contribution from isotropic free water. In this model, the free water (FW) compartment of each voxel is interpreted primarily as originating from extracellular water diffusion, and the second tensor represents the tissue compartment after removing the contribution from FW. We performed whole brain estimation of FW and calculated free water corrected tensor metrics using custom MATLAB scripts (R2023a, The Mathworks, Natick, MA, USA). Briefly, we calculated FW from single shell diffusion data based on minimization of a variational regularization framework outlined in *Pasternak et al., 2009*. Initialization of the free water estimate in this pipeline used mean diffusivity (MD) maps calculated from a single tensor fit. Thus, prior to FW estimation, we performed conventional single tensor reconstruction using FSL's DTIFIT. Next, conventional MD maps were used

for initialization of the bi-tensor reconstruction in MATLAB. Voxels with MD values greater than 0.8 x $d$ (i.e. $d$ is constant diffusivity) were assumed to be comprised of CSF and omitted from the fitting process. Fitting a bi-tensor model with DWI data from a single non-zero diffusion weight (i.e. b-value) has been described as an ill-posed problem with nearly infinite solutions, thus we employed sensible biological constraints to the minimization process, as performed in *Pasternak et al., 2009*. We set the reference $MD_t$ at 0.6 $\mu m^2/ms$. and diffusivities were limited to $\lambda_{max}$ = 2.5 $\mu m^2/ms$ and $\lambda_{min}$ = 0.1 $\mu m^2/ms$. We also assumed isotropic water diffusion at 37 ° C, corresponding to human body temperature, as a constant ($d$=3.0 x 10$^{-3}$). After initialization, 100 iterations were used to refine the FW estimates corresponding to free water corrected metrics. We used an automated quality assurance procedure and visually inspected output images (e.g. FW and fwcFA) to confirm appropriate data quality.

We used ANTs to perform nonlinear registration to MNI standard space by warping each participant's FA image to the HCP 1065 template (*Yeh, 2022*). We used participants' uncorrected FA for the registration process because the HCP 1065 template was created with FA images from single tensor reconstruction, not fwcFA images. We then applied the same transformation matrix to align FW and fwcFA images from the same participant into MNI space. Following registration, we extracted FW and fwcFA values from ROIs in MNI space. We separately evaluated ROIs from gray matter (GM) regions and white matter (WM) tracts, selected based on prior literature and relevance to navigation. Eight bilateral GM ROIs from the Mayo Clinic Adult Lifespan Template (*Schwarz et al., 2017*) were used: hippocampus, parahippocampal gyrus, entorhinal cortex, inferior temporal cortex, cuneus, superior parietal cortex, retrosplenial cortex, and caudate nucleus (*Ekstrom et al., 2017*). Seven bilateral WM ROIs from the Johns Hopkins University (JHU) white-matter tractography atlas were used: three subregions of the corpus callosum (body, genu, and splenium), fornix, posterior thalamic radiation, cingulum (hippocampus), and the fornix cres / stria terminalis (these two small tracts are not able to be differentiated at 2 mm voxel resolution).

### DWI statistics
Similar to previous work (*Wilkes et al., 2024*), we performed separate statistical analyses of ROIs from GM and WM regions, as these tissue types have dramatically different values for DWI metrics. We also separately evaluated FW and fwcFA, as these metrics provide separate but complementary information about extracellular and intracellular contributions to tissue microstructure. For each of these four combinations (GM-FW, GM-fwcFA, WM-FW, WM-fwcFA), we performed mix-designed ANOVAs with condition (Navigation/Verbal Memory/Video Control) as a between-subjects factor and ROI and session (pre/post) as within-subjects factors with corresponding post-hoc comparisons. We also performed correlations (Pearson or Spearman, as appropriate) to evaluate the relationship between DWI metrics and performance on behavioral measures. Correlations were corrected for multiple comparisons using the false discovery rate (FDR) method (*Benjamini and Hochberg, 1995*), which was applied across p-values for each ROI, performed separately for FW and fwcFA analyses.

### MTL subfield demarcation
Automatic hippocampal subfield segmentation software (ASHS) (*Yushkevich et al., 2015a*; *Yushkevich et al., 2015b*) was used to segment the subfields of the MTL based on each participant's high-resolution T2-weighted MRI image. We used ASHS with the ASHS-Princeton-1.0.0-Young-Adult (*Yushkevich et al., 2015a*). The MTL was segmented into CA1, CA2/3, DG, and subiculum (SUB), as well as the perirhinal cortex (PRC), entorhinal cortex (ERC), and parahippocampal cortex (PHC). We combined the CA2/3 and DG subfields as finer distinctions could not be made at the acquired resolution (*Zeineh et al., 2001*). Each participant's subfield segmentations were manually inspected to ensure accuracy of the segmentation protocol. The hippocampus was further subdivided into anterior (head) and posterior (body +tail) regions along its longitudinal axis using the T1-weighted MPRAGE sequence and FreeSurfer 7.4.1 software (https://surfer.nmr.mgh.harvard.edu/). To ensure reliable volume estimates, images were processed using FreeSurfer's longitudinal pipeline (*Reuter et al., 2012*). This method creates an unbiased within-subject template using a robust, inverse-consistent registration approach (*Reuter et al., 2010*). Visual quality control was conducted by three trained raters who inspected the skull stripping, surface reconstruction, and segmentation accuracy at both the within-subject template and individual timepoints. Manual edits were primarily performed on the within-subject template to correct segmentation errors—particularly in challenging regions such

as the hippocampus—since edits to the template automatically propagate to all associated time-points. Raters followed standardized FreeSurfer longitudinal editing guidelines to ensure consistent correction across subjects and time points. Discrepancies between raters were resolved via consensus discussions. This approach enhanced segmentation accuracy and consistency across longitudinal scans, thereby improving the reliability of volumetric and morphometric analyses. These quality control procedures ensured that anatomical registration was accurate and that atlas-based ROI extractions were not confounded by segmentation or alignment errors. To account for individual differences in brain size, we normalized the volume of each MTL subregion by dividing each volume by intracranial volume (ICV) and scaling by a factor of 100.

## Whole-brain volumetric analysis

We conducted whole-brain structural analyses using FreeSurfer (version 7.4.1; https://surfer.nmr.mgh.harvard.edu). T1-weighted anatomical images were processed with the longitudinal processing pipeline (*Reuter et al., 2012*).

Vertex-wise analyses of cortical volume were conducted using FreeSurfer's general linear modeling tool, mri_glmfit. Group-level statistical comparisons were corrected for multiple comparisons using mri_glmfit-sim, which implements cluster-wise correction based on Monte Carlo simulations. A vertex-wise threshold of Z>3.0 (corresponding to p<0.001, two-sided) was used to identify both positive and negative effects. Clusters were considered significant if they survived a cluster-wise corrected p-value of less than 0.05.

In addition to vertex-wise analyses, cortical parcellation was performed using the Destrieux atlas (*Destrieux et al., 2010*) (aparc.a2009s), which delineates 74 cortical regions per hemisphere, resulting in a total of 148 cortical regions of interest (ROIs). To account for inter-individual variability in brain size, each ROI volume was normalized by the estimated intracranial volume (ICV) and scaled by a factor of 100. Longitudinal statistical comparisons were conducted using paired-sample t-tests. To control for multiple comparisons across ROIs, FDR correction was applied at q<0.05.

## Regions of interest

To identify brain regions associated with memory and spatial navigation, we conducted meta-analyses using Neurosynth (https://neurosynth.org/). We first defined 11 maps based on Neurosynth association tests, using the following 11 key terms: 'memory', 'episodic memory', 'autobiographical memory', 'recognition memory', 'subsequent memory', 'memory encoding', 'memory retrieval', 'memory performance', 'memory processes', 'memory tasks', and 'memory test'. These maps were then combined to create a single, comprehensive memory-related brain mask. Then, following the same steps, we created a single, comprehensive spatial navigation-related mask using six key terms: 'navigation', 'spatial', 'spatial information', 'spatial temporal', 'time task', and 'visual spatial'. The memory-spatial map was then created by summing up the memory-related and navigation-related maps. Finally, this combined map was overlapped with the 400 ROIs from the 'Schaefer 2018 parcellation' to yield 242 ROIs, each containing a minimum of 30 voxels. Prior to ROI extraction, all registration steps—from individual subject space to MNI space—were visually inspected for each participant to ensure accurate alignment between the functional data and the Schaefer atlas.

## fMRI data preprocessing

We performed fMRI data preprocessing using FEAT (FMRI Expert Analysis Tool), version 6.00, implemented in FSL (http://www.fmrib.ox.ac.uk/fsl). The EPI images underwent motion correction, slice-timing correction, and temporal filtering with a nonlinear high-pass filter (100 s cutoff). Six motion parameters were included as confounding regressors in the model. Additionally, outlier time-points which were identified using the FSL's motion outlier detection tool (framewise displacement [FD]>0.9 mm) were incorporated as additional confounds in the first-level general linear model (GLM) analysis. If more than 20% of the volumes in a run had an FD exceeding 0.9 mm, or if the absolute head movement exceeded half the voxel size (i.e. 1.25 mm), the entire run was excluded from further analyses. For single-trial estimation, no spatial smoothing was applied. All functional images were linearly registered to the middle image of the first run, and all analyses were conducted in MNI standard space.

## Univariate activation analysis

We examined encoding and retrieval-related neural activity using the general linear model (GLM) within the FILM module of FSL. During the encoding stage, the GLM included five regressors: (1) the encoding stage of all pictures (i.e. the 6 s of the trial in which the participants were presented with an image and asked to encode that image using a temporal or spatial context); (2) rating period (i.e. reaction time, RT) when the participant made a rating judgment of all trials; (3) the remaining time period after the participant made a rating judgment of all trials (i.e. 6 s minus RT) while the stimulus stayed on the screen; (4) The duration of inter-trial interval (ITI) fixations; (5) all the instruction period within the same run. The 'X' or 'O' judgement baseline period was not coded and thus was treated as an implicit baseline. Events were modeled at the time of the stimulus onset and convolved with canonical hemodynamic response function (double gamma function). The encoding phase consisted of four runs, equally divided between spatial and temporal encoding runs. For the retrieval phase, remembered and forgotten pictures were separately modeled for both item memory retrieval and source memory retrieval. Source memory retrieval was further categorized into spatial and temporal retrieval trials. The source memory retrieval effect was defined as the difference in activity between correctly retrieved contextual pictures and the implicit baseline.

A second-level analysis was conducted to compute cross-run averages for spatial encoding effects, temporal encoding effects, combined encoding effects (spatial and temporal concatenated), spatial source memory retrieval effects, temporal source, and combined source memory retrieval effects (spatial and temporal concatenated) using a fixed-effects model. These contrasts were then used in a group analysis with a random-effects model using full FMRIB's Local Analysis of Mixed Effect 1 with automatic outlier detection (*Beckmann et al., 2003*; *Woolrich, 2008*). Unless otherwise noted, group images were thresholded using cluster detection statistics, with a height threshold of z>3.1 and a cluster probability of p<0.05, corrected for whole-brain multiple comparisons using Gaussian Random Field Theory. To specifically examine univariate activation within the hippocampus, a key region of interest in our experiment, we conducted small volume correction using the hippocampal mask derived from the Harvard-Oxford subcortical atlas in MNI152 space.

## Single-trial response estimates

General linear models (GLMs) were conducted separately to estimate the activation patterns for each of 80 encoding trials and 100 retrieval trials. In each single-trial model, a Least Square–Separate (LS-S) approach was used, in which the trial of interest was modeled as one regressor, with all other trials modeled as a separate regressor (*Mumford et al., 2012*). Specifically, during the encoding stage, each single-trial GLM included six regressors: (1) the trial of interest (i.e. the 6 s of the trial where the participants were presented with an image and asked to encode that image using a temporal or spatial context); (2) all other remaining trials within the same run; (3) rating period (i.e. reaction time, RT) when the participant made a rating judgment of all trials; (4) the remaining time period after the participant made a rating judgment of all trials (i.e. 6 s minus RT) while the stimulus stayed on the screen; (5) The duration of inter-trial interval (ITI) fixations; (6) the instruction period within the same run. The 'X' or 'O' judgement period was not coded and thus was treated as an implicit baseline.

Similarly, for the retrieval stage, each single-trial GLM included six regressors: (1) the trial of interest (i.e. the RT of the trial in which participants were asked to retrieve source information for the image); (2) all other remaining source retrieval trials within the same run; (3) the remaining time period after the participant made a source judgment (i.e. 6 s minus RT) while the stimulus remained on the screen; (4) all item retrieval periods in which participants judged whether they could recognize the item from the encoding phase by pressing 'Yes' or 'No' within the same run; (5) the remaining time period after the participant made a Yes/No judgment (i.e. 6 s minus RT) while the stimulus stayed on the screen; and (6) all instruction periods within the same run. The ITI fixation period was not modeled and was treated as an implicit baseline. Each event was modeled at the time of stimulus onset and convolved with a canonical hemodynamic response function (double gamma). To control the effects of head motion, six motion parameters were included in the GLM model as a covariate, as well as a regressor for each TR that was flagged as having greater framewise displacement (FD) than 0.9 during preprocessing. The t-map for each trial was used for multivariate pattern similarity analysis to increase the reliability by normalizing for noise (*Walther et al., 2016*).

## Multivariate pattern similarity analysis (MPS)

Multi-voxel pattern similarity (MPS) was used to measure the similarity of activation patterns by calculating the correlation between trials that were either encoded or correctly retrieved, within each ROI (*Kriegeskorte et al., 2006*). Following the approach of Power (*Power et al., 2014*), volumes with a framewise displacement greater than 0.9 mm were censored, and trials containing any censored frames within the duration of the modeled GLM response were excluded. Specifically, within-context pattern similarity (PS) during encoding was calculated using pairwise Pearson correlation coefficients between trials within the same context (i.e. spatial or temporal). Within-spatial context was calculated using only spatial trials, while within temporal context was calculated using only temporal trials. For source memory retrieval, we performed analogous calculations for within-context PS, separately measuring within-spatial context PS and within-temporal context PS, using correct retrieval trials. All MPS analyses were performed across trials from different runs to avoid temporal autocorrelations that could otherwise inflate or bias results. The resulting correlation coefficients were then transformed into Fisher's z-scores.

## Informational connectivity analysis

We performed informational connectivity among the 242 predefined ROIs by calculating pairwise correlations between representational similarity matrices (RSMs) (*Aly and Turk-Browne, 2016*; *Gao et al., 2022*; *Kriegeskorte et al., 2008*; *Zheng et al., 2023*). This was done by correlating the within-context PS for each ROI with another ROI. In contrast to traditional functional connectivity (FC) analyses, which quantify temporal correlations of BOLD signal fluctuations between brain regions, the informational connectivity approach examines the similarity of information coding patterns across regions (*Aly and Turk-Browne, 2016*; *Gao et al., 2022*; *Zheng et al., 2023*). While FC analyses rely on averaged BOLD signals within regions, representational similarity analysis (RSA) captures the fine-grained spatial patterns of neural responses to different experimental conditions (*Kriegeskorte et al., 2008*). Specifically, RSA quantifies how similarly or differently a brain region responds to different stimuli or conditions by comparing multivariate activity patterns, creating a RSM that characterizes each region's representational structure (*Kriegeskorte et al., 2008*). By correlating these RSMs between regions, we can assess whether different brain areas encode information in similar ways, providing insights into information sharing and processing that complement traditional FC measures.

Three types of within-context RSMs were calculated: one based on all trials, one based on spatial trials, and one based on temporal trials (*Figure 4a* top). All Pearson correlation coefficients (r values) were Fisher-Z transformed prior to statistical analysis. Significance thresholds for informational connectivity were determined using 10,000 permutation simulations. The reported results have been corrected for multiple comparisons using the FDR method, with a q-value threshold of less than 0.05.

## Multivariate informational connectivity pattern distance analysis

To assess changes in multivariate informational connectivity patterns between pre and post, distances were computed for each ROI by comparing spatial and temporal informational connectivity patterns before and after training. Specifically, for each ROI, the informational connectivity with the remaining 241 ROIs was extracted separately from the spatial and temporal informational connectivity matrices (ICM). The distance between the two connectivity patterns was calculated using 1 minus the Pearson or Spearman correlation coefficient (*Figure 6a*). Paired t-tests were conducted to evaluate statistical differences between pre and post, with FDR correction applied across the 242 ROIs at a q-value threshold of <0.05.

---

## **Additional information**

### Funding

| Funder | Grant reference number | Author |
| --- | --- | --- |
| National Institutes of Health | R21NS120237-01 | Arne Ekstrom<br>Steven M Weisberg |

---

| Funder | Grant reference number | Author |
|--------|------------------------|--------|
| National Institutes of Health | K01AG070333-01 | Steven M Weisberg |
| Arizona Alzheimer's Consortium | Pilot Project grant program | Li Zheng |

The funders had no role in study design, data collection and interpretation, or the decision to submit the work for publication.

## Author contributions

Li Zheng, Data curation, Software, Formal analysis, Funding acquisition, Validation, Visualization, Methodology, Writing – original draft, Writing – review and editing; Zachary Boogaart, Data curation, Validation, Investigation, Methodology, Project administration; Andrew McAvan, Data curation, Software, Validation, Methodology; Joshua Garren, Validation, Investigation, Project administration; Stephanie G Doner, Data curation, Formal analysis, Validation, Investigation, Visualization, Methodology, Project administration; Bradley J Wilkes, Software, Formal analysis, Investigation, Visualization; Will Groves, Investigation, Visualization, Project administration; Ece Yuksel, Data curation, Software, Formal analysis, Validation, Visualization; Lucia Cherep, Supervision, Investigation, Methodology, Project administration; Arne Ekstrom, Conceptualization, Resources, Formal analysis, Supervision, Funding acquisition, Investigation, Methodology, Writing – original draft, Project administration, Writing – review and editing; Steven M Weisberg, Conceptualization, Resources, Data curation, Software, Formal analysis, Supervision, Funding acquisition, Validation, Investigation, Visualization, Methodology, Writing – original draft, Project administration, Writing – review and editing

## Author ORCIDs

Li Zheng ⓘ https://orcid.org/0000-0002-9358-4353
Andrew McAvan ⓘ https://orcid.org/0000-0002-1253-884X
Stephanie G Doner ⓘ https://orcid.org/0000-0002-0338-5052
Arne Ekstrom ⓘ https://orcid.org/0000-0002-6812-2368
Steven M Weisberg ⓘ https://orcid.org/0000-0001-7003-6082

## Ethics

Human subjects: All participants provided written informed consent. The study was approved by the University of Arizona Institutional Review Board (Primary Reviewing IRB: 1805589537) and was ceded to the University of Florida Institutional Review Board (Ceded IRB: CED000000489).

Reviewer #1 (Public review): https://doi.org/10.7554/eLife.106873.3.sa1
Author response https://doi.org/10.7554/eLife.106873.3.sa2

# Additional files

## Supplementary files

MDAR checklist

## Data availability

The raw data supporting the findings of this study are publicly available at OpenNeuro (https://openneuro.org/datasets/ds004639). The code used for the analyses is available at https://osf.io/72p3y.

The following datasets were generated:

| Author(s) | Year | Dataset title | Dataset URL | Database and Identifier |
|---|---|---|---|---|
| Ekstrom A, Weisberg S | 2025 | nav_training | https://doi.org/10.18112/openneuro.ds004639.v1.0.0 | OpenNeuro, 10.18112/openneuro.ds004639.v1.0.0 |
| Zheng L | 2025 | Newly trained navigation and verbal memory skills elicit changes in task-related networks but not brain structure | https://osf.io/72p3y/ | Open Science Framework, 72p3y |

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

## Appendix 1

### Extended results

#### Control analysis for the Navigation task (navigation training in virtual Arida)

Regarding average distance error traveled for individual subsections, we observed a significant main effect of subsection $F(2,42) = 13.46$, p<0.001, $\eta^2$=0.39. We also observed significant effects of sex and site (men outperformed women and participants at the University of Arizona outperformed participants at the University of Florida), but these did not interact with each other, nor with the within-participant training effects (ps >0.41). Removing sex and site from the model did not change the effect. Removing outliers (3 standard deviations away from the grand mean) did not affect results. Regarding integrating multiple subsections, we observed a significant main effect of subsections, $F(1,22) = 5.16$, p=0.03, $\eta^2$=0.19. This finding was more fragile. Including sex and site in the model reduced the finding to non-significant (p=0.06). This analysis was also not robust to outlier removal.

#### Control analysis for verbal memory task (verbal memory training) performance

Significant improvements in verbal memory performance were observed throughout 10 days training, with slopes significantly greater than zero after controlling for sex and site for the first 5 days (t(26) = 3.964, p=0.001) and the last 5 days (t(26) = 4.877, p<0.001).

#### Control analysis for learning rate of verbal memory transfer task

When controlling for sex and site using linear mixed effect model, the Verbal Memory group still showed an increase in learning rate from pre-test to post-test (t(26) = 3.381, p=0.002, Marginal $R^2$=0.117) compared to the Navigation group (t(27) = 0.497, p=0.623, Marginal $R^2$=0.016) and Video Control group (t(21) = 0.564, p=0.588, Marginal $R^2$=0.025).

In further analysis of the learning rate for the Verbal Memory Transfer task, a potential issue arises with participants who met the criteria within a single trial, making it problematic to calculate the difference between the last and first trials. In the initial analysis, these participants were assigned a value of zero. However, in the subsequent control analysis, we excluded these participants, one from the Video Control group and two from the Navigation group, all of whom met the criteria within one trial in the post-test. All results remained consistent after excluding those participants: A mixed-design ANOVA, with condition (Navigation/Verbal Memory/Video Control) as a between-subjects factor and session (pre/post) as a within-subjects factor, revealed a main effect of session (F(1, 68)=6.724, p=0.012, $\eta^2$=0.028) and a significant interaction between condition and session (F(2, 68)=3.410, p=0.039, $\eta^2$=0.028). Paired-sample t-tests indicated that only the Verbal Memory group showed a significant increase in learning rate from pre to post (Verbal Memory: t(25) = 3.316, p=0.003, Cohen's d=0.608; Navigation: t(24) = 1.697, p=0.103, Cohen's d=0.347; Video Control: t(19) = –0.263, p=0.795, Cohen's d=–0.081). When controlling for sex and site in linear mixed effect models, the Verbal Memory group still showed the greatest increase in learning rate from pre to post (t(26) = 3.381, p=0.002, Marginal $R^2$=0.117) compared to the Navigation group (t(25) = 1.732, p=0.096, Marginal $R^2$=0.06) and Video Control group (t(20) = 0.270, p=0.790, Marginal $R^2$=0.007).

#### Slope of verbal memory transfer task

In a complementary analysis, we calculated the slope across trials as an alternative measure of the learning rate and observed similar results. The slope was derived using linear regression on the number of words recalled in each trial. A mixed-design ANOVA, with condition (Navigation/Verbal Memory/Video Control) as a between-subjects factor and session (pre/post) as a within-subjects factor, revealed a main effect of session (F(1, 71)=7.144, p=0.009, $\eta^2$=0.032) but no significant interaction between condition and session (F(2, 71)=2.579, p=0.083, $\eta^2$=0.023). Paired-sample t-tests indicated that only the Verbal Memory group showed a significant increase in learning rate from pre to post (Verbal Memory: t(25) = 3.56, p=0.002, Cohen's d=0.631; Navigation: t(26) = 0.909, p=0.372, Cohen's d=0.212; Video Control: t(20) = 0.369, p=0.716, Cohen's d=–0.121). When controlling for sex and site in linear mixed effect models, the Verbal Memory group still showed a significant increase in learning rate from pre-test to post-test (t(26) = 3.627, p=0.004, Marginal $R^2$=0.135) compared to the Navigation group (t(27) = 0.926, p=0.362, Marginal $R^2$=0.075) and Video Control group (t(38) = 0.384, p=0.703, partial $\eta^2$=0.0039). During the pre-test session and post-test

session, we also calculated classic indices of verbal memory performance, including the average number of words recalled for the 12-word list and the number of trials required to reach the criterion (i.e. 12) in the Verbal Memory Transfer task.

A mixed-design ANOVA was conducted with average number of words correctly recalled as the dependent variable, condition (Navigation/Verbal Memory/Video Control) as a between-subjects factor, and session (pre/post) as a within-subjects factor. The analysis revealed a significant interaction between condition and session (F(2, 71)=8.163, p<0.001, $\eta^2$=0.049). Surprisingly, paired-sample t-tests showed that only the Video Control group demonstrated a significant increase in average number of words recalled from pre-test to post-test (t(20) = 3.28, p=0.004, Cohen's d=0.69). In contrast, the Verbal Memory group exhibited a significant decrease in average number of words recalled (t(25) = –2.26, p=0.033, Cohen's d=–0.389), while the Navigation group showed no significant change (t(26) = 0.381, p=0.706, Cohen's d=0.08). When controlling for sex and site, the Video Control group continued to show the greatest increase in average number of words recalled from pre-test to post-test (t(21) = 3.362, p=0.003, Marginal $R^2$=0.134) compared to the Navigation group (t(27) = 0.388, p=0.701, Marginal $R^2$=0.021) and the Verbal Memory group (t(26) = –2.307, p=0.029, Marginal $R^2$=0.083).

A mixed-design ANOVA was conducted with the number of trials to criterion as the dependent variable, condition (Navigation/Verbal Memory/Video Control) as a between-subjects factor, and session (pre/post) as a within-subjects factor. The analysis revealed a significant main effect of Session (F(1, 71)=25.157, p<0.001, $\eta^2$=0.071) and a marginally significant interaction between session and condition (F(2,71) = 2.548, p=0.085, $\eta^2$=0.014). Paired-sample t-tests indicated that both the Video Control group (t(20) = –3.65, p=0.002, Cohen's d=0.856) and the Navigation group (t(26) = 3.36, p=0.002, Cohen's d=0.579) demonstrated significant decreases in the number of trials from pre-test to post-test. In contrast, the Verbal Memory group did not show a significant decrease (t(25) = –1.43, p=0.166, Cohen's d=0.238). When controlling for sex and site, the Video Control group (t(21) = –3.74, p=0.001, Marginal $R^2$=0.261) and the Navigation group (t(27) = –3.422, p=0.002, Marginal $R^2$=0.207) continued to exhibit significant decreases in learning rate from pre to post-test, while the Verbal Memory group (t(26) = –1.456, p=0.157, Marginal $R^2$=0.070) did not show a significant change.

## Control analysis for learning rate of navigation transfer task

When controlling for sex and site using linear mixed effect models or linear regression models, the Navigation group still showed the increase in learning rate from pre to post t(50) = 4.766, p<0.001, partial Eta$^2$=0.312; Verbal Memory group t(50) = 2.265, p=0.028, partial Eta$^2$=0.093; Video Control group (t(19) = 4.081, p<0.001, marginal $R^2$=0.358).

## Path error of navigation transfer task

A mixed-design ANOVA was conducted with path error as the dependent variable, condition (Navigation/Verbal Memory/Video Control) as a between-subjects factor, and session (pre/post) as a within-subjects factor. The analysis revealed a significant main effect of session (F(1,70) = 144.97, p<0.001, $\eta^2$=0.443), indicating a substantial change across sessions. However, there was no significant main effect of condition (F(2,70) = 0.726, p=0.487, $\eta^2$=0.007) and no significant session ×condition interaction (F(2,70) = 0.59, p=0.557, $\eta^2$=0.004). Paired-sample t-tests indicated that all three groups demonstrated an increase in overall pointing error from post-test to pre-test, with the Navigation group showing the largest effect (Navigation: t(26) = 8.15, p<0.001, Cohen's d=1.97; Verbal Memory: t(26) = 6.47, p<0.001, Cohen's d=1.48; Video Control: t(18) = 6.33, p<0.001, Cohen's d=1.25). When controlling for sex and site, the Navigation group still showed the greatest increase in overall pointing error from pre-test to post-test (t(26) = 8.307, p<0.001, Marginal $R^2$=0.582) compared to the Verbal Memory group (t(26) = 6.595, p<0.001, Marginal $R^2$=0.416) and the Video Control group (t(18) = 6.506, p<0.001, Marginal $R^2$=0.517).

## Navigation pointing task performance

For the pointing task, the error for each trial was determined by calculating the absolute angular difference between the participant's response and the correct angle. These differences were then averaged across all trials to produce an overall pointing error score. We also examined performance differences between participants based on the two types of pointing trials: within-environment

and between-environment. We separated trials based on whether the target building was in the environment that the participant was currently standing in (within-environment) or in the other environment (between-environment).

A mixed-design ANOVA was conducted with overall pointing error as the dependent variable, condition (Navigation/Verbal Memory/Video Control) as a between-subjects factor, and session (pre/post) as a within-subjects factor. The analysis revealed a significant main effect of session (F(1,70) = 81.66, p<0.001, $\eta$²=0.165), but no significant main effect of condition (F(2,70) = 2.618, p=0.080, $\eta$²=0.048) and no significant interaction between condition and session (F(2,70) = 0.63, p=0.54, $\eta$²=0.003). Paired-sample t-tests indicated that all three groups demonstrated a decrease in overall pointing error from pre-test to post-test, with the Verbal Memory group showing the largest effect (Navigation: t(26) = 4.76, p<0.001, Cohen's d=0.925; Verbal Memory: t(24) = 5.97, p<0.001, Cohen's d=0.983; Video Control: t(20) = 5.55, p<0.001, Cohen's d=0.758, *Appendix 3—figure 2a*). When controlling for sex and site in linear mixed effect models, both the Navigation (t(26) = 4.849, p<0.001, Marginal R²=0.32) and Verbal Memory group (t(26) = 6.091, p<0.001, Marginal R²=0.322) showed the greatest decrease in overall pointing error from pre-test to post-test compared to Video Control group (t(20) = 5.684, p<0.001, Marginal R²=0.183).

A mixed-design ANOVA was conducted with within-environment pointing error as the dependent variable, condition (Navigation/Verbal Memory/Video Control) as a between-subjects factor, and session (pre/post) as a within-subjects factor. The analysis revealed a significant main effect of session (F(1,70) = 27.58, p<0.001, $\eta$²=0.059), but no significant interaction between condition and session (F(2,70) = 0.182, p=0.834, $\eta$²<0.001). Paired-sample t-tests indicated that all three groups demonstrated a decrease in within-environment pointing error from pre to post (Navigation: t(26) = 2.36, p=0.03, Cohen's d=0.479; Verbal Memory: t(24) = 3.74, p=0.001, Cohen's d=0.5; Video Control: t(20) = 3.34, p=0.003, Cohen's d=0.435; *Appendix 3—figure 2c*). When controlling for sex and site, all three groups showed a significant decrease in <u>within-</u>environment pointing error from pre to post-test (Navigation: t(26) = 2.403, p=0.023, Marginal R²=0.165); Verbal Memory group t(24) = 3.819, p<0.001, Marginal R²=0.213; Video Control group (t(20) = 3.418, p=0.003, Marginal R²=0.168).

A mixed-design ANOVA was conducted with between-environment pointing error as the dependent variable, condition (Navigation/Verbal Memory/Video Control) as a between-subjects factor, and session (pre/post) as a within-subjects factor. The analysis revealed a significant main effect of session (F(1,70) = 76.11, p<0.001, $\eta$²=0.201), but no significant interaction between condition and session (F(2,70) = 0.97, p=0.39, $\eta$²=0.005). Paired-sample t-tests indicated that all three groups demonstrated a decrease in between-environment pointing error from pre to post-test (Navigation: t(26) = 5.16, p<0.001, Cohen's d=1.07; Verbal Memory: t(24) = 5.88, p<0.001, Cohen's d=1.13; Video Control: t(20) = 4.22, p<0.001, Cohen's d=0.871; *Appendix 3—figure 2b*). When controlling for sex and site, all three groups showed a significant increase in between-environment pointing error from pre-test to post-test, with the Navigation group showing the greatest effect (Navigation: t(26) = 5.256, p<0.001,=1.17, Marginal R²=0.360) Verbal Memory group (t(24) = 6.00, p<0.001, Marginal R²=0.333; Video Control group t(20) = 4.320, p<0.001, Cohen's d=0.183).

## Navigation model building task performance

Map accuracy on the model-building task was measured using a bidimensional regression analysis. A mixed-design ANOVA, with condition (Navigation/Verbal Memory/Video Control) as a between-subjects factor and session (pre/post) as a within-subjects factor, revealed a significant main effect of session (F(1,72) = 58.799, p<0.001, $\eta$²=0.144) with no significant interaction between condition and session (F(2,72) = 0.957, p=0.389, $\eta$²=0.005). Paired-sample t-tests indicated that all three groups demonstrated an improvement in map accuracy from pre to post-test (Navigation: t(26) = 4.95, p<0.001, Cohen's d=0.81; Verbal Memory: t(26) = 5.72, p<0.001, Cohen's d=0.94; Video Control: t(20) = 2.97, p=0.008, Cohen's d=0.66; *Appendix 3—figure 2d*). When controlling for sex and site, all three groups still showed the increase in map accuracy from pre-test to post-test (Navigation: t(26) = 5.042, p<0.001, Marginal R²=0.272; Verbal Memory: t(26) = 5.825, p<0.001, Marginal R²=0.371; Video Control group: t(20) = 3.039, p=0.006, Marginal R²=0.119).

## Source memory task performance

A series of mixed-design ANOVAs was conducted to analyze the effects of condition (Navigation/Verbal Memory/Control; between-subjects factor) and session (Pre/Post; within-subjects factor) on source memory measures: Hit rate, false alarm (FA), and reaction time (RT).

Hit Rate for Source Memory: The ANOVA revealed a significant main effect of session (F(1, 63)=6.382, p=0.014, $\eta$²=0.017), but no interaction between session and condition (F(2, 63)=0.206, p=0.815, $\eta$²=0.001). Paired-sample t-tests indicated that none of the three conditions demonstrated a significant increase in hit rate from pre to post-test (*Appendix 3—figure 4a*): Verbal Memory: t(23) = 1.429, p=0.166, Cohen's d=0.223; Navigation: t(21) = 1.847, p=0.079, Cohen's d=0.316; Control: t(19) = 1.082, p=0.293, Cohen's d=0.208. These effects remained non-significant after controlling for sex and site using linear mixed-effect models (ts <1.891, ps >0.216).

FA for Source Memory: The ANOVA showed no significant main effect of session (F(1, 63)=0.604, p=0.440, $\eta$²=0.005) or interaction between session and condition (F(2, 63)=1.445, p=0.244, $\eta$²=0.022). Paired-sample t-tests showed no significant changes in FA across sessions for any condition: Verbal Memory: t(23) = 1.477, p=0.153, Cohen's d=0.451; Navigation: t(21) = –0.699, p=0.492, Cohen's d=–0.168; Control: t(19) = –0.0005, p=0.999, Cohen's d=–0.0001, *Appendix 3—figure 4b*. These effects remained non-significant after controlling for sex and site using linear mixed-effect models (ts <1.575, ps >0.122).

Hit Rate for Spatial Source Memory: The ANOVA revealed a significant main effect of session (F(1, 63)=5.824, p=0.019, $\eta$²=0.016), with no interaction between session and condition (F(2, 63)=0.137, p=0.872, $\eta$²<0.001). Paired-sample t-tests indicated no significant increase in hit rate from pre-test to post-test in any condition: Verbal Memory: t(23) = 1.102, p=0.282, Cohen's d=0.217; Navigation: t(21) = 1.690, p=0.106, Cohen's d=0.304; Control: t(19) = 1.433, p=0.168, Cohen's d=0.214, *Appendix 3—figure 4c*. These effects remained non-significant after controlling for sex and site using linear mixed-effect models (ts <1.730, ps >0.098).

Hit Rate for Temporal Source Memory: The ANOVA showed a marginal main effect of session (F(1, 63)=3.577, p=0.063. $\eta$²=0.012), with no interaction between session and condition (F(2, 63)=0.199, p=0.820, $\eta$²=0.001). Paired-sample t-tests revealed no significant changes in hit rate across sessions: Verbal Memory: t(23) = 1.288, p=0.210, Cohen's d=0.196; Navigation: t(21) = 1.513, p=0.145, Cohen's d=0.288; Control: t(19) = 0.564, p=0.580, Cohen's d=0.147, *Appendix 3—figure 4d*. These effects remained non-significant after controlling for sex and site using linear mixed-effect models (ts <1.548, ps >0.136).

RT for Source Memory: Although no training effects were observed for hit rate or FA, RT in the source memory task was further examined. A mixed-design ANOVA revealed a marginal main effect of session (F(1, 63)=3.685, p=0.059, $\eta$²=0.017) but no significant interaction between session and condition (F(2, 63)=1.642, p=0.202, $\eta$²=0.015). Paired-sample t-tests showed no significant changes in RT from pre-test to post-test for the Navigation condition (t(21) = –0.453, p=0.655, Cohen's d=–0.107) or the Video Control condition (t(19) = 1.454, p=0.162, Cohen's d=0.363). However, the Verbal Memory condition demonstrated a marginally significant decrease in RT from pre to post-test (t(23) = 2.042, p=0.053, uncorrected, Cohen's d=0.447, *Appendix 3—figure 5a*). This effect remained the same after controlling for sex and site using linear mixed-effect model (Verbal Memory: t(24) = 2.086, p=0.048, Marginal R²=0.172; Navigation: t(22) = 0.464, p=0.647, Marginal R²=0.028; Video Control: t(20) = 1.492, p=0.151, Marginal R²=0.111).

Subsequent analyses of spatial and temporal source memory RT revealed a nuanced pattern. Specifically, the Verbal Memory group demonstrated a trend toward decreased RT in temporal source memory from pre to post-test (t(23) = 1.910, p=0.069, uncorrected, Cohen's d=0.434, *Appendix 3—figure 5c*), while no such trends were observed in the Navigation (t(21) = –0.622, p=0.541, Cohen's d=–0.163) or Video Control conditions (t(19) = 1.708, p=0.103, Cohen's d=0.445). This effect remained the same after controlling for sex and site using linear mixed effect model (Verbal Memory: t(24) = 1.951, p=0.063, Marginal R²=0.129; Navigation: t(22) = 0.636, p=0.531, Marginal R²=0.038; Video Control: t(20) = 1.752, p=0.095, Marginal R²=0.128). Conversely, for spatial source memory, paired-sample t-tests showed no statistically significant RT changes across conditions: Verbal Memory (t(23) = 1.444, p=0.162, Cohen's d=0.321), Navigation (t(21) = –0.189, p=0.852, Cohen's d=–0.041), and Video Control (t(19) = 1.073, p=0.297, Cohen's d=0.264; *Appendix 3—*

figure 5b). These effects remained not significant after controlling for sex and site using linear mixed-effect models (ts <1.475, ps >0.153).

## Diffusion MRI metrics did not change due to the training

For DWI, we separately evaluated FW and fwcFA as these measures capture different aspects of tissue microstructure. We also chose to analyze ROIs in gray matter (GM) and white matter (WM) separately, as these tissue types have dramatically different values for both FW and fwcFA. There were 8 GM ROIs and 7 WM ROIs (see Methods). For each metric and tissue type, we performed a mixed-design ANOVA with condition (Navigation/Verbal Memory/Video Control) as a between-subjects factor, ROI and session (pre/post) as within-subjects factors.

For the GM FW analysis, we found no main effect of session ($F_{(1,57)} = 0.039$, p=0.844, $\eta^2<0.001$, $BF_{10}=0.067$, strong evidence against the inclusion of the main effect of session). As expected, we observed a significant main effect of ROI ($F_{(4.397,250.643)}=436.227$, p<0.001, Greenhouse-Geisser corrected, $\eta^2=0.933$, $BF_{10}>100$, extremely strong evidence for the inclusion of the main effect of ROI) and a significant main effect of condition ($F_{(2,57)} = 3.418$, p=0.040, $\eta^2=0.007$, $BF_{10}=0.879$, strong evidence against including the main effect of condition). However, we found no significant interaction between session ×condition × ROI ($F_{(6.169,175.825)}=0.376$, p=0.898, $BF_{10}<0.001$, strong evidence against this interaction). Paired-sample t-tests confirmed that none of the three training groups exhibited significant changes from pre to post-test in any of the 8 ROIs (ps >0. 842, FDR corrected). These findings remained nonsignificant after controlling for sex and site as covariates (ps >0.826, FDR corrected).

For the GM fwcFA analysis, we found no main effect of session ($F_{(1,57)} = 0.471$, p=0.495, $\eta^2<0.001$, $BF_{10}=0.090$, strong evidence against the inclusion of the main effect of session). As expected, we observed a significant main effect of ROI ($F_{(3.584,204.312)}=207.954$, p<0.001, Greenhouse-Geisser corrected, $\eta^2=0.637$, $BF_{10}>100$, extremely strong evidence for the inclusion of the main effect of ROI). There was no significant main effect of condition ($F_{(2,57)} = 1.147$, p=0.325, $\eta^2=0.005$, $BF_{10}=0.011$, strong evidence against including the main effect of condition). We found no significant interaction between session ×condition × ROI ($F_{(4.928,140.441)}=0.288$, p=0.917, $\eta^2<0.001$, $BF_{10}<0.001$, strong evidence against this interaction). Paired-sample t-tests confirmed that none of the three training groups exhibited significant changes from pre to post-test in any of the 8 ROIs (ps >0. 958, FDR corrected). These findings remained not significant after controlling for sex and site as covariates (ps >0.834, FDR corrected).

For the WM FW analysis, we found no main effect of session ($F_{(1,57)} = 0.012$, p=0.913, $\eta^2<0.001$, $BF_{10}=0.767$, strong evidence against the inclusion of the main effect of session). As expected, we observed a significant main effect of ROI ($F_{(2.938, 167.445)}=1659.245$, p<0.001, Greenhouse-Geisser corrected, $\eta^2=0.791$, $BF_{10}>100$, extremely strong evidence for the inclusion of the main effect of ROI). There was no significant main effect of condition ($F_{(2,57)} = 0.1$, p=0.905, $\eta^2<0.001$, $BF_{10}=0.431$, strong evidence against including the main effect of condition). We found no significant interaction between session ×condition × ROI ($F_{(7.623, 217.255)}=0.923$, p=0.495, $\eta^2<0.001$, $BF_{10}<0.001$, strong evidence against this interaction). Paired-sample t-tests confirmed that none of the three training groups exhibited significant changes from pre to post-test in any of the 7 ROIs (ps >0.400, FDR corrected). These findings remained not significant after controlling for sex and site as covariates (ps >0.597, FDR corrected).

For the WM fwcFA analysis, we found no main effect of session ($F_{(1,57)} < 0.001$, p=0.981, $\eta^2<0.001$, $BF_{10}=0.090$, strong evidence against the inclusion of the main effect of session). As expected, we observed a significant main effect of ROI ($F_{(4.093, 233.279)}=2340.878$, p<0.001, Greenhouse-Geisser corrected, $\eta^2=0.628$, $BF_{10}>100$, extremely strong evidence for the inclusion of the main effect of ROI). There was no significant main effect of condition ($F_{(2,57)} = 0.025$, p=0.975, $\eta^2<0.001$, $BF_{10}=0.011$, strong evidence against including the main effect of condition). We found no significant interaction between session ×condition × ROI ($F_{(6.924, 197.328)}=1.954$, p=0.064, $\eta^2<0.001$, $BF_{10}<0.001$, strong evidence against this interaction). Paired-sample t-tests confirmed that none of the three training groups exhibited significant changes from pre to post-test in any of the 7 ROIs (ps >0.083, FDR corrected). These findings remained not significant after controlling for sex and site as covariates (ps >0.063, FDR corrected).

## The correlation between hippocampal volume and behavioral performance

Improvements in the learning rate of the Verbal Memory Transfer task, but not the Navigation Transfer task, were found to correlate with lateral hippocampal volume but not with anterior and posterior hippocampus (using average volume between pre and post-test).

Consistent findings were obtained when analyzing hemispheric hippocampal volumes (see MTL subregions and either the average number *Appendix 3—figure 3*). Specifically, a significant positive correlation was observed between the learning rate in the Verbal Memory group and both left hippocampal volume (r(23) = 0.568, p-FDR=.014) and right hippocampal volume (r(23) = 0.601, p-FDR=.007), while no significant correlations were found for the Navigation (left: r(25) = 0.038, p-FDR=0.858; right: r(25) = 0.008, p-FDR=0.970) or Video Control group (left: r(19) = –0.123, p-FDR=0.858; right: r(19) = –0.163, p=0.758), after controlling for sex and site as covariates. Fisher's z-tests revealed that the positive correlation in the Verbal Memory group was significantly stronger than that in the Navigation group even after controlling for sex and site (left hippocampus: Z=1.942, p-FDR=0.039; right hippocampus: Z=2.326, p-FDR=0.015) and Video Control group (left hippocampus: Z=2.234, p-FDR=0.038; right hippocampus: Z=2.702, p-FDR=0.010) while no significant difference was observed between the Navigation group and Video Control group (left hippocampus: Z=–0.439, p=0.669; right hippocampus: Z=–0.553, p-FDR=0.710).

An analysis was conducted to assess the correlation between anterior and posterior hippocampal volumes and changes in learning rate on the verbal memory transfer task. However, the improvement in learning rate demonstrated no significant correlation with either anterior or posterior hippocampal volume in any of the three groups (ps >0.232, *Appendix 2—table 4*). This lack of correlation remained consistent after accounting for the influence of sex and site as covariates (ps >0.114).

We also correlated hippocampal volume and MTL subregions with the change in the average number of words recalled, number of trials to criterion, and slope from linear regression (see Appendix 1) between the post-test and pre-test for the verbal memory transfer task. The analysis revealed no significant correlations between hippocampal volume or MTL subregions and either the average number of words recalled or the number of trials to criterion (*Appendix 2—tables 6 and 7*), regardless of whether sex and site were included as covariates. However, when correlating slope with hippocampal volume, a positive correlation was found for the Verbal Memory group (total hippocampal volume: *r*=0.552, p=0.006; left hippocampus: *r*=0.502, p=0.015;right hippocampus: *r*=0.561, p=0.005, *Appendix 2—table 8*), however, no such correlation was found for the Navigation group (total hippocampus: *r*=0.094, p=0.656; left hippocampus: *r*=0.114, p=0.586; right hippocampus: *r*=0.073, p=0.727) and Video Control (total hippocampus: *r*=–0.239, p=0.325; left hippocampus: *r*=–0.217, p=0.372; right hippocampus: *r*=–0.238, p=0.326) even after controlling for sex and site as covariates.

We also correlated hippocampal volume with the change in learning rate from the post-test and pre-test for the navigation transfer task. No significant correlations were found between total hippocampal volume and changes in learning rate in any of the three conditions (Navigation: r(25) = –0.27, p=0.181; Verbal Memory: r(24) = –0.31, p=0.129; Video Control: r(17) = 0.124, p=0.612), regardless of whether sex and site were included as covariates. Similar no significant correlations were observed for both left hippocampal volume (Navigation: r(25) = –0.25, p=0.214; Verbal Memory: r(24) = –0.29, p=0.160; Video Control: r(17) = 0.25, p=0.310) and right hippocampal volume (Navigation: r(25) = –0.27, p=0.167; Verbal Memory: r(24) = –0.302, p=0.142; Video Control: r(17) = –0.018, p=0.940), regardless of whether sex and site were included as covariates. We also did not find any significant correlations when examining hippocampal subfields or surrounding MTL subregions (p's>0.20, *Appendix 2—table 5*).

The relationship between total hippocampal volume, MTL subregion volumes, and changes in performance on the navigation transfer task was assessed through correlational analyses, focusing on path error, overall pointing error, between-environment pointing error, within-environment pointing error, and map accuracy. These analyses demonstrated a lack of significant associations between hippocampal volume and MTL subregions and any of the behavioral metrics under consideration (*Appendix 2—table 9*, *Appendix 2—tables 10–13*), regardless of the inclusion of sex and site as covariates.

Improvements in the learning rate of the Verbal Memory Transfer task, but not the Navigation Transfer task, were found to correlate with both total hippocampal volume and the volume of the

CA2/3/DG subfield (using volume data only from the pre-test). We examined the hypothesis that baseline hippocampal volume might be associated with either verbal or navigation performance. For this analysis, we utilized the hippocampal volume (or subfield volume) obtained for each participant at pre-test, rather than averaging volumes across pre and post-test. We then assessed the relationship between hippocampal volumes at pre-test and the change in learning rate from pre-test to post-test for both the Verbal Memory transfer task and the Navigation Transfer task.

We found a marginal correlation in the verbal memory training group between the pre-test total hippocampal volume and the observed improvement in verbal memory performance from pre- to post-test (r(23) = 0.352, p=0.084). Further analysis accounting for sex and site as covariates revealed a significant positive correlation between total hippocampal volume and the learning rate in the Verbal Memory condition (r(23) = 0.545, p=0.007). This effect was specific to the verbal memory training; no significant correlation was identified for either the Navigation condition (r(25) = 0.072, p=.721) or the Video condition (r(19) = −0.209, p*P*=0.362); the same result was observed when controlling for covariates (Navigation condition: r(25) = 0.082, p=0.695; Video condition: r(19) = −0.144, p=0.555). Consistent findings were obtained when analyzing hemispheric hippocampal volumes. Specifically, a significant positive correlation was observed between the learning rate in the Verbal Memory condition and both left hippocampal volume (r(23) = 0.504, p=0.014) and right hippocampal volume (r(23) = 0.545, p=0.007), while no significant correlations were found for the Navigation condition (left: r(25) = 0.122, p=0.561; right: r(25) = 0.045, p=0.832) or Video Control condition (left: r(19) = −0.084, p=0.734; right: r(19) = −0.187, p=0.444), after controlling for sex and site as covariates.

We further examined the correlation between CA23DG volume and learning rate change. The CA23DG subfield showed a positive correlation with the change in learning rate from pre to post-test in the verbal memory transfer task of the Verbal Memory condition (r(23) = 0.504, p=0.01), suggesting that individuals in the Verbal Memory condition with larger CA23DG volumes exhibited greater improvements in memory performance from pre- to post-test. This correlation persisted even after controlling for sex and site as covariates (r(23) = 0.439, p=0.036). No significant correlations were observed for the CA23DG subfield in the Navigation (r(25) = 0.007, p=0.972) or Video Control conditions (r(19) = −0.05, p=0.828), regardless of whether sex and site were included as covariates.

Fisher's z-tests revealed that the positive correlation in the Verbal Memory condition was significantly stronger than that in the Navigation even after controlling for sex and site (total hippocampus: Z=1.793, p=0.037; left hippocampus: Z=1.464, p=0.072; right hippocampus: Z=1.919, p=0.027; CA23DG: Z=1.687, p=0.046) and Video conditions (total hippocampus: Z=2.382, p=0.009; left hippocampus: Z=2.009, p=0.022; right hippocampus: Z=2.518, p=0.006; CA23DG: Z=1.766, p=0.038), while no significant difference was observed between the Navigation and Video Control conditions (total hippocampus: Z=−0.731, p=0.768; left hippocampus: Z=−0.662, p=0.746; right hippocampus: Z=−0.750, p=0.773; CA23DG: Z=−0.214, p=0.585).

Improvements in the learning rate of the Verbal Memory Transfer task, but not the Navigation Transfer task, were found to correlate with both total hippocampal volume and the volume of the CA2/3/DG subfield (using volume data only from the post-test). We examined the hypothesis that baseline hippocampal volume may be associated with either verbal or navigation performance. For this analysis, we utilized the hippocampal volume (or subfield volume) obtained for each participant at post-test, rather than averaging volumes across pre-test and post-test. We then assessed the relationship between hippocampal volumes at post-test and the change in learning rate from pre-test to post-test for both the Verbal Memory transfer task and the Navigation transfer task.

We found a marginal correlation in the verbal memory training group between the post-test total hippocampal volume and the observed improvement in verbal memory performance from pre- to post-test (r(23) = 0.360, p=0.078). Further analysis accounting for sex and site as covariates revealed a significant positive correlation between total hippocampal volume and the learning rate in the Verbal Memory condition (r(23) = 0.623, p=0.001). This effect was specific to the verbal memory training; no significant correlation was identified for either the Navigation condition (r(25) = −0.009, p=0.965) or the Video Control condition (r(19) = −0.178, p=0.439); the same was true when controlling for covariates (Navigation condition: r(25) = 0.000, p=0.999; Video Control condition r(18) = −0.083, p=0.736). Consistent findings were obtained when analyzing hippocampal volumes by hemisphere. Specifically, a significant positive correlation was observed between the learning rate

in the Verbal Memory condition and both left hippocampal volume (r(23) = 0.597, p=0.003) and right hippocampal volume (r(23) = 0.594, p=0.003), while no significant correlations were found for the Navigation (left: r(25) = 0.026, p=0.901; right: r(24) = –0.023, p=0.912) or Video Control conditions (left: r(19) = –0.036, p=0.883; right: r(18) = –0.109, p=0.656), after controlling for sex and site as covariates.

We further examined the correlation between CA23DG volume and learning rate change. The CA23DG subfield showed a positive correlation with the change in learning rate from pre to post-test in the verbal memory transfer task of the Verbal Memory condition (r(23) = 0.545, p=0.005), suggesting that individuals in the Verbal Memory condition with larger CA23DG volumes exhibited greater improvement in memory performance from pre to post-test. This correlation persisted even after controlling for sex and site as covariates (r(23) = 0.537, p=0.008). No significant correlations were observed for the CA23DG subfield in the Navigation (r(25) = –0.027, p=0.894) or Video Control conditions (r(19) = –0.110, p=0.634), regardless of whether sex and site were included as covariates.

Fisher's z-tests revealed that the positive correlation in the Verbal Memory condition was significantly stronger than that in the Navigation group even after controlling for sex and site (total hippocampus: Z=2.473, p=0.007; left hippocampus: Z=2.243, p=0.012; right hippocampus: Z=2.398, p=0.008; CA23DG: Z=2.209, p=0.014) and the Video Control condition (total hippocampus: Z=2.558, p=0.005; left hippocampus: Z=2.280, p=0.011; right hippocampus: Z=2.498, p=0.006; CA23DG: Z=2.337, p=0.014), while no significant difference was observed between the Navigation and the Video conditions (total hippocampus: Z=–0.266, p=0.605; left hippocampus: Z=–0.2, p=0.841; right hippocampus: Z=–0.276, p=0.609; CA23DG: Z=–0.291, p=0.615).

## The correlation between the verbal memory training task, hippocampus volume and MTL subfields

Additionally, we calculated correlations between the slopes from the first 5 days and the last 5 days of the Verbal Memory training task with hippocampal volume and MTL subregions separately. No significant correlations were identified between the training effect and any MTL subregion or the total hippocampal volume (*Appendix 2—tables 14 and 15*).

## Informational connectivity changes specific to spatial or temporal context encoding in the verbal memory and navigation interventions

In the Verbal Memory condition, increased informational connectivity during encoding was observed during the post-test stage compared to the pre-test stage (i.e. post >pre), relative to the combined Navigation and Video conditions. Specifically, enhanced connectivity was identified between the right frontal orbital cortex and right precuneus (t=3.93, p<0.05), the right dorsal lateral occipital cortex (dLOC) and left temporal occipital fusiform cortex (TOFC) (t=3.98, p<0.05), and the right temporal pole and left superior parietal lobule (SPL) (t=4.33, p<0.05). Conversely, no significant decreases in connectivity were observed from post to pre-test (i.e. post <pre) in the Verbal Memory condition, relative to the combined Navigation and Video conditions (*Figure 4d*, top-middle). In contrast, the Navigation condition showed significantly decreased informational connectivity during the post-test compared to the pre-test stage (i.e. post <pre), relative to the combined Verbal Memory and Video conditions. Specifically, reduced connectivity was observed between the right intracalcarine cortex and the left ventral LOC (t=3.85, p<0.05). However, the Navigation condition also demonstrated significantly increased connectivity during post compared to pre (i.e. post >pre) between the left posterior cingulate gyrus (PCC) and right temporal fusiform cortex (TFC) (t=4.44, p<0.05), as well as between the left middle temporal gyrus and left SPL (t=4.96, p<0.05), relative to the combined Verbal Memory and Video conditions (*Figure 4d*, bottom-middle).

Similar patterns emerged when analyzing trials encoded within temporal contexts. In the Verbal Memory condition, increased informational connectivity was observed during the post-test compared to the pre-test (i.e. post >pre), relative to the combined Navigation and Video conditions. Specifically, enhanced connectivity was identified between the left middle frontal gyrus (MFG) and left middle temporal gyrus (MTG) (t=4.01, p<0.05). Conversely, no significant decreases in connectivity were observed from post to pre (i.e. post <pre) in the Verbal Memory condition, relative to the combined Navigation and Video conditions (*Figure 4d*, top-right). In contrast, the Navigation condition exhibited significantly decreased informational connectivity during the post-test compared to the pre-test (i.e. post <pre), relative to the combined Verbal Memory and Video

conditions. Specifically, reductions were observed between the left frontal orbital cortex and right angular gyrus (AG) (t=4.07, p<0.05), right supramarginal gyrus (RSMG) and left supramarginal gyrus (LSMG) (t=3.63, p<0.05), left superior frontal gyrus (LSFG) and right frontal pole (t=4.18, p<0.05), left paracingulate gyrus and right frontal pole (t=4.04, p<0.05), and LSFG and right frontal pole (t=4.14, p<0.05). Conversely, no significant increases in connectivity were observed from post to pre (i.e. post >pre) in the Navigation condition, relative to the combined Verbal Memory and Video conditions (*Figure 4d*, bottom-right).

## Informational connectivity changes specific to spatial or temporal context retrieval in the verbal memory and navigation interventions

When analyzing only spatial RSMs during source retrieval, we observed decreased informational connectivity between the left superior frontal gyrus (LSFG) and right superior frontal gyrus (SFG) (t=4.50, p<0.05), as well as between the right frontal pole and right posterior cingulate cortex (PCC) (t=4.10, p<0.05) in the post-test compared to the pre-test (i.e. post <pre) in the Verbal Memory condition, relative to the combined Navigation and Video conditions. Conversely, we observed increased connectivity between the left middle temporal gyrus (LMTG) and right supramarginal gyrus (RSMG) (t=3.51, p<0.05), and between the right dorsal lateral occipital cortex (RdLOC) and right supramarginal gyrus (RSMG) (t=3.73, p<0.05) from pre to post (i.e. post >pre) in the Verbal Memory condition, relative to the combined Navigation and Video conditions (*Figure 5b*, top-middle).

We observed significantly increased informational connectivity between the left frontal pole and right dorsal lateral occipital cortex (dLOC) (t=3.42, p<0.05), as well as between the left frontal pole and right parahippocampal cortex (PHC) (t=4.32, p<0.05) in the post-test compared to the pre-test (i.e. post >pre) in the Navigation condition, relative to the combined Verbal Memory and Video conditions. Conversely, no significant decreases in connectivity were observed from pre to post (i.e. post <pre) in the Navigation condition, relative to the combined Verbal Memory and Video conditions (*Figure 5b*, bottom-middle).

When analyzing only temporal RSMs during source retrieval, we observed significantly decreased informational connectivity from pre-test to post-test (i.e. post <pre) in the Verbal Memory condition, relative to the combined Navigation and Video conditions, across multiple brain regions. Specifically, decreased connectivity was found between the following regions: left frontal pole and left middle temporal gyrus (t=3.71, p<0.05), left frontal pole and left angular gyrus (t=4.03, p<0.05), left paracingulate gyrus and left precuneus (t=3.62, p<0.05), left precuneus and left ventral lateral occipital cortex (t=3.60, p<0.05), left ventral lateral occipital cortex and right dorsal lateral occipital cortex (t=4.12, p<0.05), left frontal pole and right angular gyrus (t=4.70, p<0.05), left paracingulate gyrus and right angular gyrus (t=4.39, p<0.05), left paracingulate gyrus and right middle temporal gyrus (t=4.12, p<0.05), right precuneus and right frontal pole (t=4.68, p<0.05), left medial frontal gyrus and right inferior frontal gyrus (t=3.91, p<0.05), left precuneus and right middle frontal gyrus (t=3.91, p<0.05), left precuneus and right dorsal lateral occipital cortex (t=3.49, p<0.05), right paracingulate gyrus and right dorsal lateral occipital cortex (t=4.06, p<0.05), left retrosplenial cortex and right medial frontal cortex (t=3.99, p<0.05), left frontal pole and right frontal pole (t=4.81, p<0.05), left frontal pole and right superior frontal gyrus (t=3.90, p<0.05), and left frontal pole and right precuneus (t=4.14, p<0.05), as well as right precuneus and left superior frontal gyrus (t=4.14, p<0.05). In contrast, we observed an increase in connectivity between the left ventral lateral occipital cortex and left middle frontal gyrus (t=3.96, p<0.05) from pre to post (i.e. post >pre) in the Free Recall condition, relative to the combined Navigation and Video conditions (*Figure 5b*, top-right).

We observed a significant increase in informational connectivity from pre-test to post-test (i.e. post >pre) in the Navigation condition, relative to the combined Verbal Memory and Video conditions, between the following regions: left precuneus and left middle frontal gyrus (t=4.10, p<0.05), left precuneus and left superior frontal gyrus (t=3.89, p<0.05), right lingual gyrus and right dorsal lateral occipital cortex (t=4.07, p<0.05), right lingual gyrus and right dorsal lateral occipital cortex (t=3.68, p<0.05), and left superior frontal gyrus and right middle temporal gyrus (t=3.77, p<0.05). In contrast, no significant decreases in connectivity were observed from pre to post (i.e. post <pre) in the Navigation condition, relative to the combined Verbal Memory and Video conditions (*Figure 5b*, bottom-right).

### Task-related changes in brain activation during encoding as a result of both the verbal memory and navigation interventions

We investigated how different training interventions influenced univariate brain activation during the encoding phase by examining whole-brain activity changes between the pre-test and post-test stages for the Verbal Memory, Navigation, and Video conditions. In the Verbal Memory condition, a significant decrease in activation was observed in several regions during the post-test compared to the pre-test (post <pre). These regions included the left dorsolateral occipital cortex (dLOC; Z=4.66, MNI: −46,−62, 30), the left middle frontal gyrus (MFG; Z=4.32, MNI: −42, 12, 50), the left frontal pole (Z=4.20, MNI: −24, 40, 46), the left middle temporal gyrus (MTG; Z=4.28, MNI: −60,−24, −12), and the left precuneus (Z=3.80, MNI: −10,−52, 38) (see *Appendix 3—figure 6a*). In contrast, in the Navigation condition, post-test activation was significantly increased in the right middle temporal gyrus (MTG; Z=4.03, MNI: 42,−54, 6) compared to the pre-test (post >pre) (see *Appendix 3—figure 6b*).

However, no group-specific changes in activity were observed when comparing the pre-test and post-test stages across the three conditions: Verbal Memory, Navigation, and Video Control (*Appendix 2—table 16*).

### Activity changes specific to spatial or temporal context encoding in the verbal memory and navigation training conditions

We also investigated whether navigation and Verbal Memory training differentially affected spatial versus temporal encoding. In the Verbal Memory condition, a significant decrease in spatial encoding activation was observed during the post-test compared to the pre-test (post <pre, *Appendix 3—figure 6c*). Specifically, reduced activation was detected in the left lateral occipital cortex (LOC; Z=3.97, MNI: −44,−62, 32) and the left frontal pole (Z=3.99, MNI: −18, 48, 36). Similarly, in the Verbal Memory group, a decrease in temporal encoding activation was observed in the left LOC (Z=3.97, MNI: −40,−62, 32) during the post-training stage relative to the pre-test (post <pre, *Appendix 3—figure 6d*). In contrast, no significant differences between pre and post were found in the Navigation condition for either spatial or temporal encoding.

### No task-related changes in brain activation during retrieval as a result of either the verbal memory or navigation interventions

See *Appendix 2—table 16*

### No activity changes specific to spatial or temporal context retrieval in the Verbal Memory and Navigation training conditions

See *Appendix 2—table 16*

# Appendix 2

**Appendix 2—table 1.** Demographics information.

|  |  | Verbal Memory | Navigation | Video Control | All |
|---|---|---|---|---|---|
|  | Sample size | 27 | 27 | 21 | 75 |
| Sex | Male | 7 | 11 | 7 | 25 |
|  | Female | 20 | 16 | 14 | 50 |
| Site | Site 1 | 13 | 13 | 13 | 39 |
|  | Site 2 | 14 | 14 | 8 | 36 |
|  | Male-Site 1 | 2 | 4 | 5 | 11 |
|  | Female-Site 1 | 9 | 9 | 8 | 26 |
|  | Male-Site 2 | 5 | 7 | 2 | 14 |
|  | Female-Site 2 | 9 | 7 | 6 | 22 |
|  | Age (min-max/Year) | 20.89 (18-26) | 22.11 (18-32) | 22 (18–32) | 21.67 |
|  | Age-Site1 | 20.31 | 23.00 | 23.00 | 22.10 |
| Age | Age-Site2 | 21.43 | 21.29 | 20.38 | 21.03 |

**Appendix 2—table 2.** Sample size information by condition and test, reflecting exclusions due to outlier performance, excessive head movement during scanning, or missing data.

| Test | Verbal Memory | Navigation | Video |
|---|---|---|---|
| Verbal Memory Training | 27 | 27 | 21 |
| Navigation Training | 27 | 27 | 21 |
| Verbal Memory Transfer Task | 26 | 27 | 21 |
| Navigation Transfer Task | 27 | 27 | 19 |
| Navigation Pointing Task | 25 | 27 | 21 |
| Navigation Model Building Task | 27 | 27 | 21 |
| Source Memory Task-Encoding | 25 | 26 | 20 |
| Source Memory Task-Retrieval | 24 | 22 | 20 |
| Hippocampal volume | 26 | 27 | 21 |
| DWI | 20 | 22 | 18 |

**Appendix 2—table 3.** Study Timeline.

| Pre-test (Day 1) | Training (Days 2–11) | Post-test (Day 12) |
|---|---|---|
| Complete consent, MRI prescreen, and demographics survey | Comprehensive training varies based on which condition the participant was assigned | Review consent, MRI prescreen |
| Navigation Transfer task (Virtual Silcton, Unity) | Navigation Task (navigation training in virtual Arida, Unity) | Navigation Transfer task (Virtual Silcton, Unity) |
| Navigation Pointing task (Virtual Silcton, web-based) | Verbal memory Task (verbal memory training with the modified method of loci, Unity) | Navigation Pointing task (Virtual Silcton, web-based) |
| Navigation Model Building task (Virtual Silcton, web-based) | Control task (video control by viewing informative videos and answering questions, Qualtrics) | Navigation Model Building task (Virtual Silcton, web-based) |
| Verbal Memory Transfer task (Psychopy) |  | Verbal Memory Transfer task (PsychoPy) |
| Attention task (web-based) |  | Attention task (web-based) |

*Appendix 2—table 3 Continued on next page*

*Appendix 2—table 3 Continued*

| Pre-test (Day 1) | Training (Days 2–11) | Post-test (Day 12) |
|---|---|---|
| Source Memory task (PsychoPy, fMRI scanned) | | Source Memory task (PsychoPy, fMRI scanned) |
| | | Debrief survey (Quatrics) |

**Appendix 2—table 4.** Correlations between the volumes of MTL subregions and changes in learning rate from pre-test to post-test in the Verbal Memory Transfer task.

| Condition | ROI | r | p | Method |
|---|---|---|---|---|
| | Ant-HIP | 0.248 | 0.232 | Pearson |
| | Post-HIP | –0.043 | 0.837 | Spearman |
| | CA1 | 0.138 | 0.51 | Pearson |
| | CA23DG | 0.532 | 0.006* | Pearson |
| | SUB | 0.025 | 0.907 | Pearson |
| | ERC | 0.069 | 0.743 | Pearson |
| | PRC | 0.162 | 0.438 | Pearson |
| Verbal Memory N = 25 | PHC | –0.068 | 0.748 | Spearman |
| | Ant-HIP | –0.064 | 0.751 | Spearman |
| | Post-HIP | 0.183 | 0.362 | Pearson |
| | CA1 | 0.084 | 0.676 | Pearson |
| | CA23DG | –0.012 | 0.953 | Pearson |
| | SUB | 0.035 | 0.863 | Pearson |
| | ERC | –0.142 | 0.48 | Pearson |
| | PRC | 0.023 | 0.91 | Pearson |
| Navigation N = 27 | PHC | 0.223 | 0.264 | Pearson |
| | Ant-HIP | –0.105 | 0.651 | Pearson |
| | Post-HIP | 0.035 | 0.88 | Pearson |
| | CA1 | –0.131 | 0.572 | Pearson |
| | CA23DG | –0.083 | 0.719 | Pearson |
| | SUB | –0.24 | 0.294 | Pearson |
| | ERC | 0.149 | 0.52 | Spearman |
| | PRC | 0.121 | 0.6 | Pearson |
| Video N = 21 | PHC | –0.292 | 0.198 | Pearson |

*Significant results after FDR correction. Ant: anterior; Post: posterior

**Appendix 2—table 5.** Correlations between the volumes of MTL subregions and changes in learning rate from pre-test to post-test in the Navigation Transfer task.

| Condition | ROI | r | p | Method |
|---|---|---|---|---|
| | Ant-HIP | –0.328 | 0.102 | Pearson |
| | Post-HIP | –0.147 | 0.471 | Spearman |
| | HIP | –0.314 | 0.119 | Pearson |
| | LHIP | –0.291 | 0.149 | Pearson |
| | RHIP | –0.305 | 0.129 | Pearson |
| | CA1 | –0.181 | 0.377 | Pearson |
| | CA23DG | –0.261 | 0.198 | Pearson |
| | SUB | –0.257 | 0.205 | Pearson |
| | ERC | 0.027 | 0.897 | Pearson |
| | PRC | –0.018 | 0.931 | Pearson |
| Verbal Memory N = 26 | PHC | –0.243 | 0.23 | Spearman |
| | Ant-HIP | –0.482 | 0.012 | Spearman |
| | Post-HIP | –0.042 | 0.837 | Pearson |
| | HIP | –0.266 | 0.181 | Pearson |
| | LHIP | –0.247 | 0.214 | Pearson |
| | RHIP | –0.274 | 0.167 | Pearson |
| | CA1 | –0.246 | 0.215 | Pearson |
| | CA23DG | –0.334 | 0.088 | Pearson |
| | SUB | 0.087 | 0.665 | Pearson |
| | ERC | 0.008 | 0.967 | Pearson |
| | PRC | –0.021 | 0.917 | Pearson |
| Navigation N = 27 | PHC | 0.028 | 0.891 | Pearson |
| | Ant-HIP | –0.101 | 0.681 | Pearson |
| | Post-HIP | 0.048 | 0.846 | Pearson |
| | HIP | 0.124 | 0.612 | Pearson |
| | LHIP | 0.246 | 0.31 | Pearson |
| | RHIP | 0.028 | 0.911 | Spearman |
| | CA1 | 0.02 | 0.934 | Pearson |
| | CA23DG | 0.128 | 0.6 | Pearson |
| | SUB | 0.053 | 0.828 | Pearson |
| | ERC | 0.194 | 0.425 | Pearson |
| | PRC | 0.24 | 0.322 | Pearson |
| Video N = 19 | PHC | –0.089 | 0.716 | Pearson |

Ant: anterior. Post: posterior

**Appendix 2—table 6.** Correlations between the volumes of MTL subregions and changes in the average number of correctly recalled words from pre-test to post-test in the Verbal Memory Transfer task.

| Condition | ROI | r | p | Method |
|---|---|---|---|---|
| | Ant-HIP | –0.350 | 0.087 | Pearson |
| | Post-HIP | –0.042 | 0.842 | Spearman |
| | HIP | –0.235 | 0.258 | Pearson |
| | LHIP | –0.172 | 0.410 | Pearson |
| | RHIP | –0.267 | 0.197 | Pearson |
| | CA1 | –0.212 | 0.309 | Pearson |
| | CA23DG | –0.329 | 0.109 | Pearson |
| | SUB | 0.106 | 0.615 | Pearson |
| | ERC | 0.030 | 0.888 | Pearson |
| | PRC | –0.219 | 0.292 | Pearson |
| Verbal Memory N = 25 | PHC | –0.202 | 0.334 | Spearman |
| | Ant-HIP | 0.122 | 0.545 | Spearman |
| | Post-HIP | –0.131 | 0.516 | Pearson |
| | HIP | 0.012 | 0.953 | Pearson |
| | LHIP | –0.032 | 0.875 | Pearson |
| | RHIP | 0.049 | 0.809 | Pearson |
| | CA1 | 0.023 | 0.911 | Pearson |
| | CA23DG | 0.043 | 0.830 | Pearson |
| | SUB | –0.085 | 0.674 | Pearson |
| | ERC | 0.135 | 0.503 | Pearson |
| | PRC | –0.062 | 0.757 | Pearson |
| Navigation N = 27 | PHC | –0.256 | 0.197 | Pearson |
| | Ant-HIP | –0.111 | 0.633 | Pearson |
| | Post-HIP | –0.012 | 0.959 | Pearson |
| | HIP | –0.125 | 0.589 | Pearson |
| | LHIP | –0.156 | 0.499 | Pearson |
| | RHIP | –0.183 | 0.426 | Spearman |
| | CA1 | 0.116 | 0.616 | Pearson |
| | CA23DG | –0.147 | 0.525 | Pearson |
| | SUB | –0.212 | 0.357 | Pearson |
| | ERC | –0.365 | 0.104 | Spearman |
| | PRC | –0.189 | 0.413 | Pearson |
| Video N = 21 | PHC | –0.088 | 0.703 | Pearson |

Ant: anterior. Post: posterior

**Appendix 2—table 7.** Correlations between the volumes of MTL subregions and changes in the number of trials to criterion from pre-test to post-test in the Verbal Memory Transfer task.

| Condition | ROI | r | p | Method |
|---|---|---|---|---|
| | Ant-HIP | 0.095 | 0.653 | Spearman |
| | Post-HIP | 0.316 | 0.124 | Spearman |
| | HIP | –0.163 | 0.438 | Spearman |
| | LHIP | –0.111 | 0.598 | Spearman |
| | RHIP | –0.223 | 0.285 | Spearman |
| | CA1 | 0.028 | 0.896 | Spearman |
| | CA23DG | 0.074 | 0.726 | Spearman |
| | SUB | –0.398 | 0.049 | Spearman |
| | ERC | –0.334 | 0.103 | Spearman |
| | PRC | 0.139 | 0.509 | Spearman |
| Verbal Memory N = 25 | PHC | 0.041 | 0.846 | Spearman |
| | Ant-HIP | 0.084 | 0.679 | Spearman |
| | Post-HIP | 0.008 | 0.970 | Spearman |
| | HIP | –0.023 | 0.908 | Spearman |
| | LHIP | –0.042 | 0.837 | Spearman |
| | RHIP | –0.033 | 0.872 | Spearman |
| | CA1 | –0.015 | 0.943 | Spearman |
| | CA23DG | –0.028 | 0.890 | Spearman |
| | SUB | 0.099 | 0.622 | Spearman |
| | ERC | –0.011 | 0.956 | Spearman |
| | PRC | 0.137 | 0.495 | Spearman |
| Navigation N = 27 | PHC | 0.257 | 0.196 | Spearman |
| | Ant-HIP | 0.223 | 0.332 | Pearson |
| | Post-HIP | 0.242 | 0.290 | Pearson |
| | HIP | 0.359 | 0.110 | Pearson |
| | LHIP | 0.413 | 0.063 | Pearson |
| | RHIP | 0.222 | 0.334 | Spearman |
| | CA1 | 0.167 | 0.471 | Pearson |
| | CA23DG | 0.178 | 0.440 | Pearson |
| | SUB | 0.436 | 0.048 | Pearson |
| | ERC | 0.202 | 0.379 | Spearman |
| | PRC | 0.072 | 0.756 | Pearson |
| Video N = 21 | PHC | 0.461 | 0.035 | Pearson |

Ant: anterior. Post: posterior

**Appendix 2—table 8.** Correlations between the volumes of MTL subregions and changes in slope from pre-test to post-test in the Verbal Memory Transfer task.

| Condition | ROI | r | p | Method |
|---|---|---|---|---|
| | Ant-HIP | 0.30 | 0.15 | Pearson |
| | Post-HIP | −0.14 | 0.51 | Spearman |
| | HIP | 0.39 | 0.05 | Pearson |
| | LHIP | 0.32 | 0.12 | Pearson |
| | RHIP | 0.42 | 0.04 | Pearson |
| | CA1 | 0.13 | 0.54 | Pearson |
| | CA23DG | 0.50 | 0.01 | Pearson |
| | SUB | 0.15 | 0.46 | Pearson |
| | ERC | 0.11 | 0.59 | Pearson |
| | PRC | 0.19 | 0.37 | Pearson |
| Verbal Memory N = 26 | PHC | 0.01 | 0.95 | Spearman |
| | Ant-HIP | 0.04 | 0.83 | Spearman |
| | Post-HIP | 0.23 | 0.26 | Pearson |
| | HIP | 0.09 | 0.67 | Pearson |
| | LHIP | 0.10 | 0.63 | Pearson |
| | RHIP | 0.07 | 0.71 | Pearson |
| | CA1 | 0.18 | 0.36 | Pearson |
| | CA23DG | 0.02 | 0.92 | Pearson |
| | SUB | 0.06 | 0.75 | Pearson |
| | ERC | −0.10 | 0.63 | Pearson |
| | PRC | 0.01 | 0.97 | Pearson |
| Navigation N = 27 | PHC | 0.21 | 0.30 | Pearson |
| | Ant-HIP | −0.23 | 0.32 | Pearson |
| | Post-HIP | −0.05 | 0.82 | Pearson |
| | HIP | −0.29 | 0.20 | Pearson |
| | LHIP | −0.28 | 0.21 | Pearson |
| | RHIP | −0.28 | 0.21 | Spearman |
| | CA1 | −0.16 | 0.50 | Pearson |
| | CA23DG | −0.15 | 0.51 | Pearson |
| | SUB | −0.31 | 0.17 | Pearson |
| | ERC | 0.09 | 0.71 | Spearman |
| | PRC | 0.05 | 0.82 | Pearson |
| Video N = 21 | PHC | −0.33 | 0.14 | Pearson |

Ant: anterior. Post: posterior

**Appendix 2—table 9.** Correlations between the volumes of MTL subregions and changes in path errors from pre-test to post-test in the Navigation Transfer task.

| Condition | ROI | r | p | Method |
|---|---|---|---|---|
| | Ant-HIP | 0.080 | 0.698 | Pearson |
| | Post-HIP | 0.117 | 0.570 | Spearman |
| | HIP | 0.041 | 0.844 | Pearson |
| | LHIP | 0.117 | 0.570 | Pearson |
| | RHIP | 0.041 | 0.844 | Pearson |
| | CA1 | 0.163 | 0.426 | Pearson |
| | CA23DG | −0.214 | 0.294 | Pearson |
| | ERC | 0.274 | 0.176 | Pearson |
| | PHC | 0.047 | 0.820 | Pearson |
| | PRC | −0.452 | 0.020 | Pearson |
| Verbal Memory N = 26 | SUB | 0.396 | 0.045 | Spearman |
| | Ant-HIP | −0.137 | 0.493 | Spearman |
| | Post-HIP | −0.181 | 0.366 | Pearson |
| | HIP | −0.058 | 0.775 | Pearson |
| | LHIP | −0.032 | 0.873 | Pearson |
| | RHIP | −0.078 | 0.701 | Pearson |
| | CA1 | −0.294 | 0.137 | Pearson |
| | CA23DG | 0.011 | 0.956 | Pearson |
| | ERC | −0.072 | 0.723 | Pearson |
| | PHC | 0.027 | 0.892 | Pearson |
| | PRC | −0.083 | 0.681 | Pearson |
| Navigation N = 27 | SUB | 0.101 | 0.617 | Pearson |
| | Ant-HIP | 0.301 | 0.211 | Pearson |
| | Post-HIP | 0.210 | 0.389 | Pearson |
| | HIP | −0.193 | 0.429 | Pearson |
| | LHIP | −0.187 | 0.443 | Pearson |
| | RHIP | −0.130 | 0.595 | Spearman |
| | CA1 | −0.186 | 0.446 | Pearson |
| | CA23DG | 0.052 | 0.833 | Pearson |
| | ERC | 0.065 | 0.791 | Pearson |
| | PHC | −0.291 | 0.227 | Pearson |
| | PRC | 0.150 | 0.540 | Pearson |
| Video N = 19 | SUB | −0.277 | 0.252 | Pearson |

Ant: anterior. Post: posterior

**Appendix 2—table 10.** Correlations between the volumes of MTL subregions and changes in overall pointing errors from pre-test to post-test in the Navigation Pointing Error task.

| Condition | ROI | r | p | Method |
|---|---|---|---|---|
| | Ant-HIP | –0.143 | 0.504 | Pearson |
| | Post-HIP | 0.191 | 0.369 | Spearman |
| | HIP | –0.205 | 0.336 | Pearson |
| | LHIP | –0.127 | 0.554 | Pearson |
| | RHIP | –0.252 | 0.236 | Pearson |
| | CA1 | 0.000 | 1.000 | Pearson |
| | CA23DG | –0.201 | 0.346 | Pearson |
| | ERC | –0.086 | 0.690 | Pearson |
| | PHC | –0.134 | 0.531 | Spearman |
| | PRC | 0.314 | 0.136 | Pearson |
| Verbal Memory N=24 | SUB | –0.263 | 0.214 | Pearson |
| | Ant-HIP | –0.119 | 0.553 | Spearman |
| | Post-HIP | –0.317 | 0.107 | Pearson |
| | HIP | –0.159 | 0.429 | Pearson |
| | LHIP | –0.198 | 0.322 | Pearson |
| | RHIP | –0.121 | 0.548 | Pearson |
| | CA1 | –0.295 | 0.136 | Pearson |
| | CA23DG | –0.017 | 0.932 | Pearson |
| | ERC | –0.024 | 0.907 | Pearson |
| | PHC | –0.227 | 0.256 | Pearson |
| | PRC | –0.090 | 0.656 | Pearson |
| Navigation N=27 | SUB | –0.208 | 0.299 | Pearson |
| | Ant-HIP | –0.155 | 0.503 | Pearson |
| | Post-HIP | 0.041 | 0.860 | Pearson |
| | HIP | 0.200 | 0.385 | Pearson |
| | LHIP | 0.247 | 0.280 | Pearson |
| | RHIP | 0.042 | 0.859 | Spearman |
| | CA1 | –0.040 | 0.864 | Pearson |
| | CA23DG | 0.188 | 0.415 | Pearson |
| | ERC | –0.246 | 0.282 | Spearman |
| | PHC | 0.247 | 0.282 | Pearson |
| | PRC | 0.238 | 0.300 | Pearson |
| Video N=21 | SUB | 0.250 | 0.274 | Pearson |

**Appendix 2—table 11.** Correlations between the volumes of MTL subregions and changes in within-environment pointing error from pre-test to post-test in the Navigation Pointing Error task.

| Condition | ROI | r | p | Method |
|---|---|---|---|---|
| | Ant-HIP | −0.142 | 0.508 | Pearson |
| | Ant-HIP | 0.165 | 0.439 | Spearman |
| | HIP | 0.110 | 0.610 | Pearson |
| | LHIP | 0.163 | 0.447 | Pearson |
| | RHIP | 0.055 | 0.798 | Pearson |
| | CA1 | 0.040 | 0.854 | Pearson |
| | CA23DG | 0.239 | 0.261 | Pearson |
| | ERC | 0.192 | 0.369 | Pearson |
| | PHC | −0.145 | 0.497 | Spearman |
| | PRC | 0.411 | 0.046 | Pearson |
| Verbal Memory N=24 | SUB | −0.149 | 0.488 | Pearson |
| | Ant-HIP | −0.242 | 0.222 | Spearman |
| | Post-HIP | −0.249 | 0.211 | Pearson |
| | HIP | −0.166 | 0.408 | Pearson |
| | LHIP | −0.183 | 0.360 | Pearson |
| | RHIP | −0.147 | 0.466 | Pearson |
| | CA1 | −0.199 | 0.321 | Pearson |
| | CA23DG | −0.160 | 0.427 | Pearson |
| | ERC | 0.019 | 0.925 | Pearson |
| | PHC | −0.178 | 0.374 | Pearson |
| | PRC | −0.204 | 0.309 | Pearson |
| Navigation N=27 | SUB | −0.010 | 0.961 | Pearson |
| | Ant-HIP | 0.166 | 0.471 | Pearson |
| | Post-HIP | 0.181 | 0.432 | Pearson |
| | HIP | 0.060 | 0.796 | Pearson |
| | LHIP | 0.040 | 0.865 | Pearson |
| | RHIP | 0.044 | 0.850 | Spearman |
| | CA1 | −0.078 | 0.738 | Pearson |
| | CA23DG | 0.301 | 0.185 | Pearson |
| | ERC | −0.260 | 0.254 | Spearman |
| | PHC | −0.112 | 0.629 | Pearson |
| | PRC | 0.415 | 0.062 | Pearson |
| Video N=21 | SUB | −0.232 | 0.312 | Pearson |

**Appendix 2—table 12.** Correlations between the volumes of MTL subregions and changes in between-environment pointing error from pre-test to post-test in the Navigation Pointing Error Task.

| Condition | ROI | r | p | Method |
|---|---|---|---|---|
| Verbal Memory N=24 | Ant-HIP | –0.120 | 0.576 | Pearson |
| | Post-HIP | 0.190 | 0.373 | Spearman |
| | HIP | –0.285 | 0.177 | Pearson |
| | LHIP | –0.211 | 0.323 | Pearson |
| | RHIP | –0.321 | 0.127 | Pearson |
| | CA1 | –0.014 | 0.948 | Pearson |
| | CA23DG | –0.327 | 0.119 | Pearson |
| | ERC | –0.172 | 0.423 | Pearson |
| | PHC | –0.183 | 0.391 | Spearman |
| | PRC | 0.227 | 0.286 | Pearson |
| | SUB | –0.261 | 0.218 | Pearson |
| Navigation N=27 | Ant-HIP | –0.145 | 0.468 | Spearman |
| | Post-HIP | –0.295 | 0.136 | Pearson |
| | HIP | –0.125 | 0.534 | Pearson |
| | LHIP | –0.169 | 0.400 | Pearson |
| | RHIP | –0.084 | 0.676 | Pearson |
| | CA1 | –0.291 | 0.141 | Pearson |
| | CA23DG | 0.063 | 0.756 | Pearson |
| | ERC | –0.042 | 0.834 | Pearson |
| | PHC | –0.210 | 0.294 | Pearson |
| | PRC | –0.011 | 0.955 | Pearson |
| | SUB | –0.275 | 0.165 | Pearson |
| Video N=21 | Ant-HIP | –0.248 | 0.278 | Pearson |
| | Post-HIP | –0.047 | 0.838 | Pearson |
| | HIP | 0.182 | 0.430 | Pearson |
| | LHIP | 0.243 | 0.290 | Pearson |
| | RHIP | –0.021 | 0.930 | Spearman |
| | CA1 | –0.003 | 0.989 | Pearson |
| | CA23DG | 0.049 | 0.835 | Pearson |
| | ERC | –0.183 | 0.425 | Spearman |
| | PHC | 0.318 | 0.160 | Pearson |
| | PRC | 0.044 | 0.850 | Pearson |
| | SUB | 0.382 | 0.087 | Pearson |

**Appendix 2—table 13.** Correlations between the volumes of MTL subregions and changes in model-building accuracy from pre-test to post-test in the Navigation Model Building task.

| Condition | ROI | r | p | Method |
|---|---|---|---|---|
| | Ant-HIP | 0.125 | 0.544 | Pearson |
| | Post-HIP | −0.158 | 0.438 | Spearman |
| | HIP | 0.066 | 0.749 | Pearson |
| | LHIP | 0.025 | 0.905 | Pearson |
| | RHIP | 0.096 | 0.640 | Pearson |
| | CA1 | −0.048 | 0.816 | Pearson |
| | CA23DG | 0.137 | 0.504 | Pearson |
| | ERC | −0.009 | 0.965 | Pearson |
| | PHC | 0.074 | 0.721 | Spearman |
| | PRC | 0.345 | 0.085 | Pearson |
| Verbal Memory N=26 | SUB | 0.017 | 0.936 | Pearson |
| | Ant-HIP | −0.035 | 0.863 | Spearman |
| | Post-HIP | 0.008 | 0.969 | Pearson |
| | HIP | 0.078 | 0.701 | Pearson |
| | LHIP | 0.046 | 0.821 | Pearson |
| | RHIP | 0.102 | 0.612 | Pearson |
| | CA1 | 0.322 | 0.102 | Pearson |
| | CA23DG | −0.068 | 0.737 | Pearson |
| | ERC | −0.065 | 0.746 | Pearson |
| | PHC | −0.038 | 0.852 | Pearson |
| | PRC | −0.129 | 0.520 | Pearson |
| Navigation N=27 | SUB | 0.074 | 0.715 | Pearson |
| | Ant-HIP | −0.083 | 0.722 | Pearson |
| | Post-HIP | −0.231 | 0.313 | Pearson |
| | HIP | 0.039 | 0.868 | Pearson |
| | LHIP | −0.020 | 0.931 | Pearson |
| | RHIP | 0.092 | 0.690 | Spearman |
| | CA1 | 0.129 | 0.577 | Pearson |
| | CA23DG | −0.181 | 0.432 | Pearson |
| | ERC | 0.203 | 0.377 | Spearman |
| | PHC | 0.046 | 0.842 | Pearson |
| | PRC | −0.407 | 0.067 | Pearson |
| Video N=21 | SUB | 0.235 | 0.304 | Pearson |

**Appendix 2—table 14.** Correlations between the average volumes of MTL subregions and slope from Day 1 to Day 5 in the Verbal Memory Training task.

| Condition | ROI | r | p | Method |
|---|---|---|---|---|
| | Ant-HIP | –0.0323 | 0.8755 | Pearson |
| | Post-HIP | –0.1321 | 0.52 | Spearman |
| | HIP | 0.0912 | 0.6577 | Pearson |
| | LHIP | 0.2391 | 0.2395 | Pearson |
| | RHIP | –0.0455 | 0.8252 | Pearson |
| | CA1 | –0.0993 | 0.6292 | Pearson |
| | CA23DG | 0.0299 | 0.8846 | Pearson |
| | ERC | 0.1806 | 0.3773 | Pearson |
| | PHC | 0.1397 | 0.4962 | Spearman |
| | PRC | –0.3257 | 0.1044 | Pearson |
| Verbal Memory N=26 | SUB | 0.3239 | 0.1065 | Pearson |

**Appendix 2—table 15.** Correlations between the average volumes of MTL subregions and slope from Day 6 to Day 10 in the Verbal Memory Training task.

| Condition | ROI | r | p | Method |
|---|---|---|---|---|
| | Ant-HIP | –0.0882 | 0.6682 | Pearson |
| | Post-HIP | –0.0576 | 0.78 | Spearman |
| | HIP | –0.1188 | 0.5633 | Pearson |
| | LHIP | 0.0082 | 0.9685 | Pearson |
| | RHIP | –0.2188 | 0.2829 | Pearson |
| | CA1 | –0.0902 | 0.6612 | Pearson |
| | CA23DG | –0.2916 | 0.1484 | Pearson |
| | ERC | 0.1524 | 0.4573 | Pearson |
| | PHC | 0.1542 | 0.4521 | Spearman |
| | PRC | –0.3393 | 0.09 | Pearson |
| Verbal Memory N=26 | SUB | 0.2459 | 0.2259 | Pearson |

**Appendix 2—table 16.** Univariate activation changes from pre-test to post-test during encoding and retrieval, with whole-brain cluster correction and small-volume correction for the hippocampus reported separately.

**Pre vs. Post**

| Condition | Contrast | Results (Hippocampus) | Results (whole brain) |
|---|---|---|---|
| | Video | None | None |
| | Verbal Memory | None | T1>T2: left dLOC (Z=4.66, MNI: −46,−62, 30), left MFG (Z=4.32, MNI: −42, 12, 50), left frontal pole (Z=4.2, MNI: −24, 40, 46), left MTG (Z=4.28, MNI: −60,−24, −12), left precuneus (Z=3.8, MNI: −10,−52, 38). |
| | Navigation | None | T2>T1: right MTG (Z=4.03, MNI: 42,−54, ss6). |
| | Verbal Memory vs. Video | None | None |
| | Navigation vs. Video | None | None |
| | Verbal Memory vs. Navigation | None | None |
| | Verbal Memory vs. Video +Navigation | None | None |
| Encoding | Navigation vs. Video +Verbal Memory | None | None |
| | Video | None | None |
| | Verbal Memory | None | T1>T2: Left LOC (Z=3.97, MNI: −44,−62,32), Left frontal pole (Z=3.99, MNI: −18, 48, 36). |
| | Navigation | None | None |
| | Verbal Memory vs. Video | None | None |
| | Navigation vs. Video | None | None |
| | Verbal Memory vs. Navigation | None | None |
| | Verbal Memory vs. Video +Navigation | None | None |
| Spatial encoding | Navigation vs. Video +Verbal Memory | None | None |
| | Video | None | None |
| | Verbal Memory | None | T1>T2: Left LOC (Z=3.97, MNI: −40,−62, 32) |
| | Navigation | None | None |
| | Verbal Memory vs. Video | None | None |
| | Navigation vs. Video | None | None |
| | Verbal Memory vs. Navigation | None | None |
| | Verbal Memory vs. Video +Navigation | None | None |
| Temporal encoding | Navigation vs. Video +Verbal Memory | None | None |

*Appendix 2—table 16 Continued on next page*

*Appendix 2—table 16 Continued*

**Pre vs. Post**

| | | | |
|---|---|---|---|
| | Video | None | None |
| | Verbal Memory | None | None |
| | Navigation | None | None |
| | Verbal Memory vs. Video | None | None |
| | Navigation vs. Video | None | None |
| | Verbal Memory vs. Navigation | None | None |
| | Verbal Memory vs. Video +Navigation | None | None |
| Retrieval | Navigation vs. Video +Verbal Memory | None | None |
| | Video | None | None |
| | Verbal Memory | None | None |
| | Navigation | None | None |
| | Verbal Memory vs. Video | None | None |
| | Navigation vs. Video | None | None |
| | Verbal Memory vs. Navigation | None | None |
| | Verbal Memory vs. Video +Navigation | None | None |
| Spatial Retrieval | Navigation vs. Video +Verbal Memory | None | None |
| | Video | None | None |
| | Verbal Memory | None | None |
| | Navigation | None | None |
| | Verbal Memory vs. Video | None | None |
| | Navigation vs. Video | None | None |
| | Verbal Memory vs. Navigation | None | None |
| | Verbal Memory vs. Video +Navigation | None | None |
| Temporal Retrieval | Navigation vs. Video +Verbal Memory | None | None |

## Appendix 3

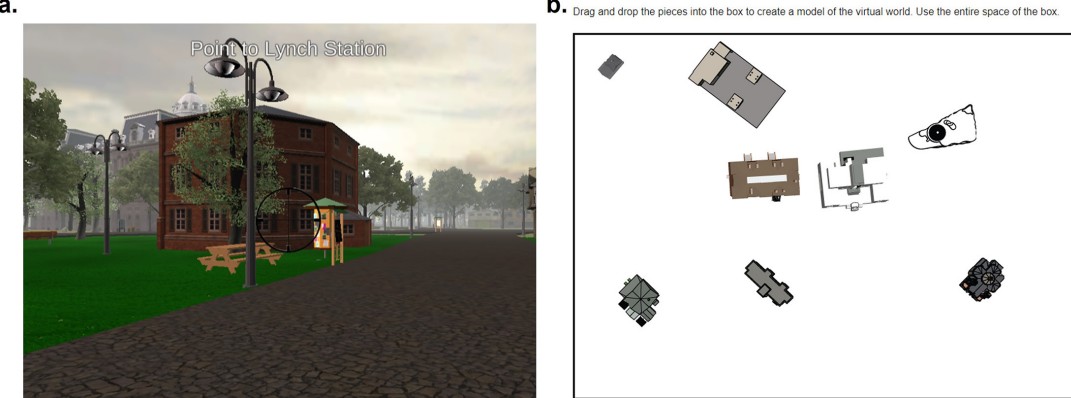

**Appendix 3—figure 1.** Schematic representations of additional formats of the Navigation Transfer Task. (**a**) Navigation Pointing Task. (**b**) Navigation Model Building Task.

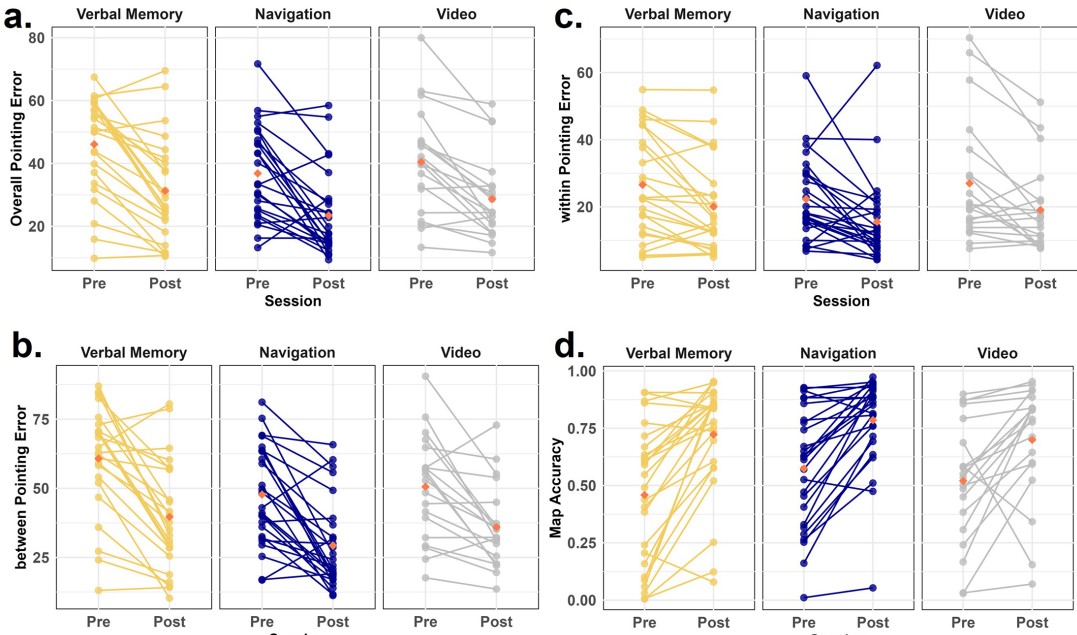

**Appendix 3—figure 2.** Performance on the Navigation Pointing task and Navigation Model Building task. (**a**) Paired-sample t-tests indicated that all three groups demonstrated a decrease in overall pointing error from pre-test to post-test, with the Verbal Memory group showing the largest effect (Navigation: t(26) = 4.76, p<0.001, Cohen's d=0.925; Verbal Memory: t(24) = 5.97, p<0.001, Cohen's d=0.983; Video Control: t(20) = 5.55, p<0.001, Cohen's d=0.758). (**b**) Paired-sample t-tests indicated that all three groups demonstrated a decrease in between-environment pointing error from pre to post-test (Navigation: t(26) = 5.16, p<0.001, Cohen's d=1.07; Verbal Memory: t(24) = 5.88, p<0.001, Cohen's d=1.13; Video Control: t(20) = 4.22, p<0.001, Cohen's d=0.871). (**c**) Paired-sample t-tests indicated that all three groups demonstrated a decrease in within-environment pointing error from pre to post (Navigation: t(26) = 2.36, p=0.03, Cohen's d=0.479; Verbal Memory: t(24) = 3.74, p=0.001, Cohen's d=0.5; Video Control: t(20) = 3.34, p=0.003, Cohen's d=0.435). (**d**) Paired-sample t-tests indicated that all three groups demonstrated an improvement in map accuracy from pre to post-test (Navigation: t(26) = 4.95, p<0.001, Cohen's d=0.81; Verbal Memory: t(26) = 5.72, p<0.001, Cohen's d=0.94; Video Control: t(20) = 2.97, p=0.008, Cohen's d=0.66). Each individual dot represents data from an individual subject. Red diamonds represent the mean value.

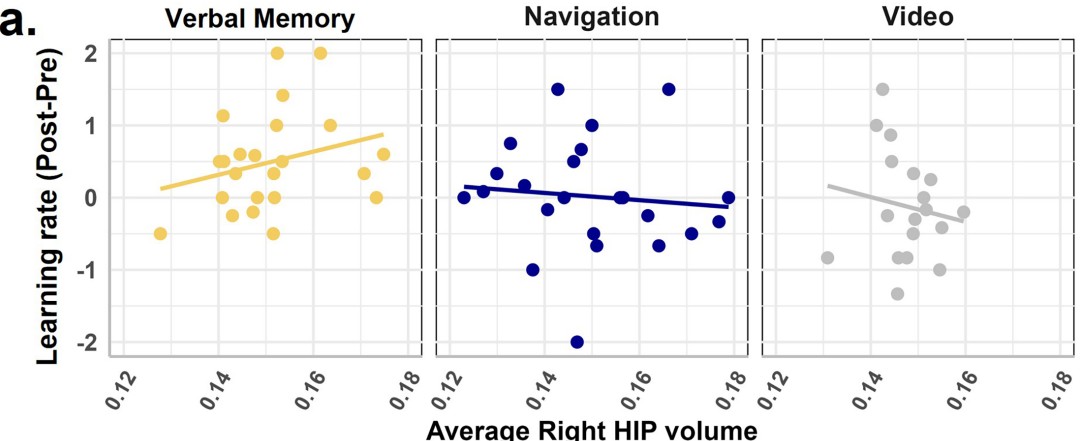

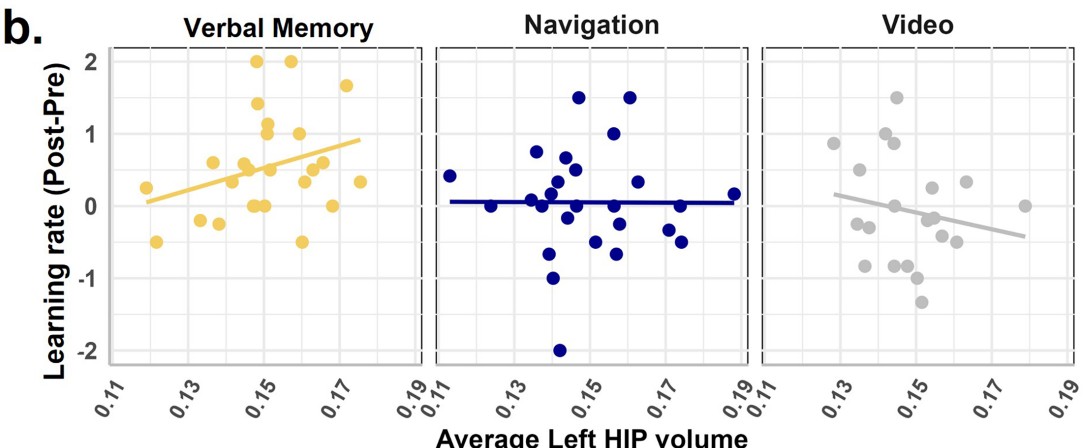

**Appendix 3—figure 3.** Left and right hippocampal volumes were correlated with changes in Verbal Memory Transfer task performance between pre-test and post-test. (**a**) A significant positive correlation was observed between the learning rate in the Verbal Memory group and right hippocampal volume (r(23) = 0.601, p-FDR=0.007), while no significant correlations were found for the Navigation (r(25) = 0.008, p-FDR=0.970) or Video Control group (r(19) = –0.163, p=0.758), after controlling for sex and site as covariates. (**b**) A significant positive correlation was observed between the learning rate in the Verbal Memory group and left hippocampal volume (r(23) = 0.568, p-FDR=0.014), while no significant correlations were found for the Navigation (r(25) = 0.038, p-FDR=0.858) or Video Control group (left: r(19) = –0.123, p-FDR=0.858), after controlling for sex and site as covariates.

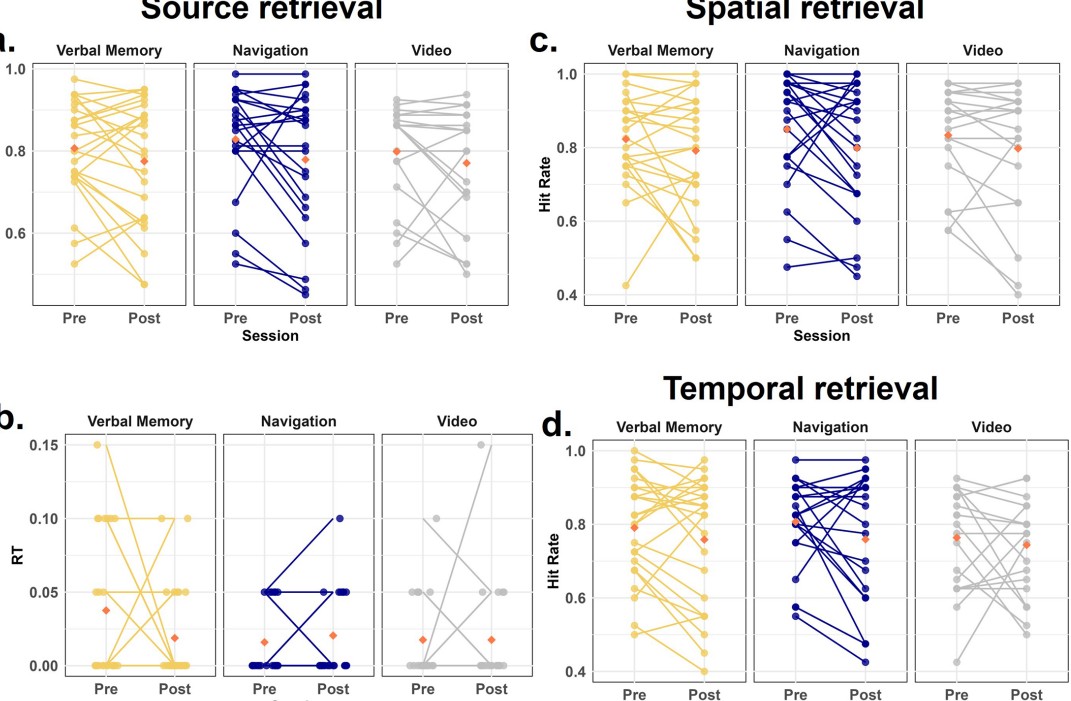

**Appendix 3—figure 4.** Memory performance (hit rate and false alarm rate) during the Source Memory task in the scanner. (**a**) Paired-sample t-tests indicated that none of the three conditions demonstrated a significant increase in hit rate from pre to post-test: Verbal Memory: t(23) = 1.429, p=0.166, Cohen's d=0.223; Navigation: t(21) = 1.847, p=0.079, Cohen's d=0.316; Control: t(19) = 1.082, p=0.293, Cohen's d=0.208. (**b**) Paired-sample t-tests showed no significant changes in FA across sessions for any condition: Verbal Memory: t(23) = 1.477, p=0.153, Cohen's d=0.451; Navigation: t(21) = –0.699, p=0.492, Cohen's d=–0.168; Control: t(19) = –0.0005, p=0.999, Cohen's d=–0.0001. (**c**) Paired-sample t-tests indicated no significant increase in hit rate from pre-test to post-test in any condition: Verbal Memory: t(23) = 1.102, p=0.282, Cohen's d=0.217; Navigation: t(21) = 1.690, p=0.106, Cohen's d=0.304; Control: t(19) = 1.433, p=0.168, Cohen's d=0.214. (**d**) Paired-sample t-tests revealed no significant changes in hit rate across sessions: Verbal Memory: t(23) = 1.288, p=0.210, Cohen's d=0.196; Navigation: t(21) = 1.513, p=0.145, Cohen's d=0.288; Control: t(19) = 0.564, p=0.580, Cohen's d=0.147.

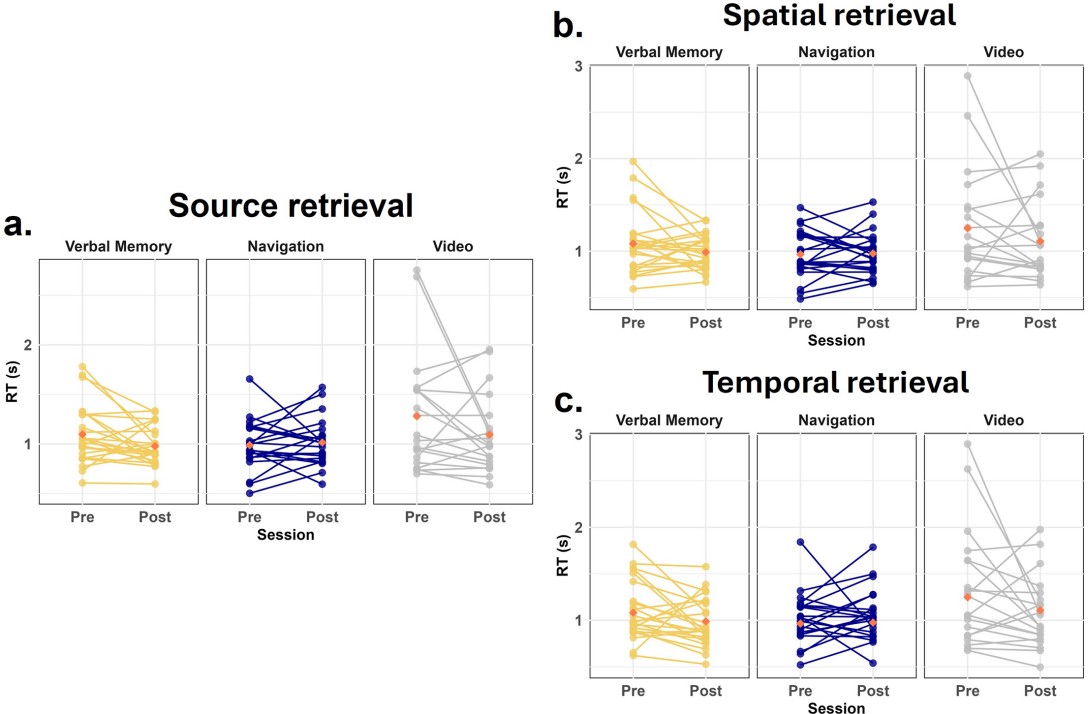

**Appendix 3—figure 5.** Memory performance (reaction time) during the Source Memory task in the scanner. (**a**) Paired-sample t-tests showed no significant changes in RT from pre-test to post-test for the Navigation condition (t(21) = –0.453, p=0.655, Cohen's d=–0.107) or the Video Control condition (t(19) = 1.454, p=0.162, Cohen's d=0.363). However, the Verbal Memory condition demonstrated a marginally significant decrease in RT from pre to post-test (t(23) = 2.042, p=0.053, uncorrected, Cohen's d=0.447). (**b**) For spatial source memory, paired-sample t-tests showed no statistically significant RT changes across conditions: Verbal Memory (t(23) = 1.444, p=0.162, Cohen's d=0.321), Navigation (t(21) = –0.189, p=0.852, Cohen's d=–0.041), and Video Control (t(19) = 1.073, p=0.297, Cohen's d=0.264). (**c**) For temporal source memory, paired-sample t-tests showed that the Verbal Memory group demonstrated a trend toward decreased RT in temporal source memory from pre to post-test (t(23) = 1.910, p=0.069, uncorrected, Cohen's d=0.434), while no such trends were observed in the Navigation (t(21) = –0.622, p=0.541, Cohen's d=–0.163) or Video Control conditions (t(19) = 1.708, p=0.103, Cohen's d=0.445).

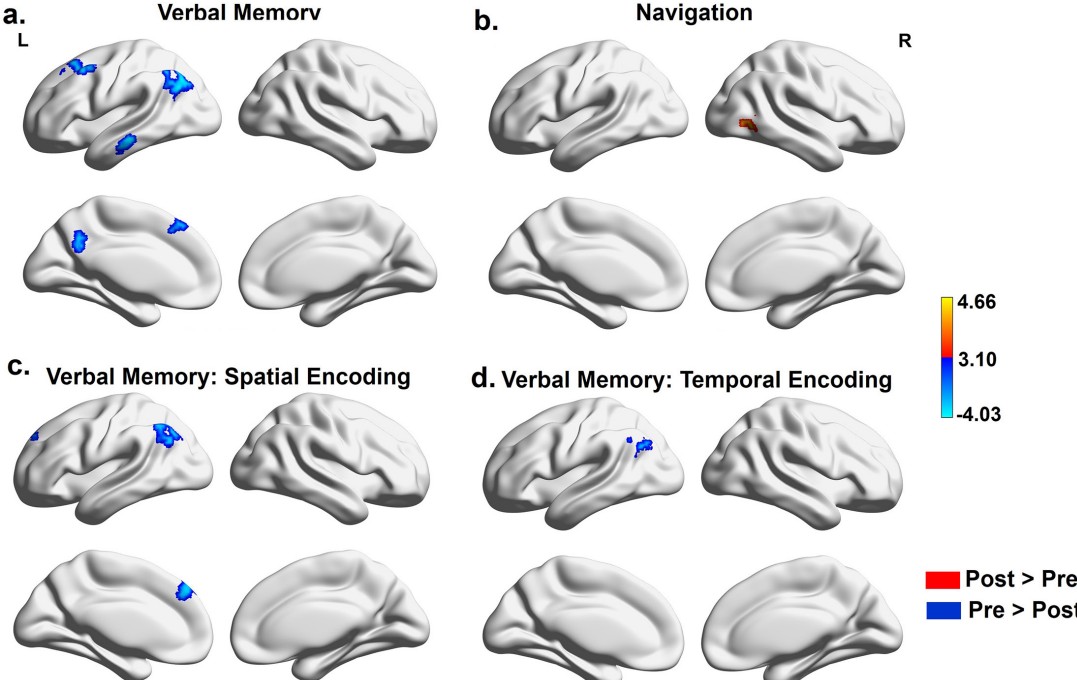

**Appendix 3—figure 6.** Univariate activation changes from pre-test to post-test during Source Memory task encoding. (**a**) In the Verbal Memory condition (n=25), a significant decrease in activation was observed in several regions during the post-test compared to the pre-test (post <pre). These regions included the left dorsolateral occipital cortex (dLOC; Z=4.66, MNI: −46,−62, 30), the left middle frontal gyrus (MFG; Z=4.32, MNI: −42, 12, 50), the left frontal pole (Z=4.20, MNI: −24, 40, 46), the left middle temporal gyrus (MTG; Z=4.28, MNI: −60,−24, −12), and the left precuneus (Z=3.80, MNI: −10,−52, 38). (**b**) In the Navigation condition (n=26), post-test activation was significantly increased in the right middle temporal gyrus (MTG; Z=4.03, MNI: 42,−54, 6) compared to the pre-test (post >pre). (**c**) In the Verbal Memory condition (n=25), a significant decrease in spatial encoding activation was observed during the post-test compared to the pre-test (LOC; Z=3.97, MNI: −44,−62, 32) (**d**) In the Verbal Memory group (n=25), a decrease in temporal encoding activation was observed in the left LOC (Z=3.97, MNI: −40,−62, 32) during the post-training stage relative to the pre-test. All results were derived from whole-brain voxel-wise paired t-tests, and statistical maps were thresholded using cluster-based detection methods with a voxel-wise threshold of z>3.1 and a cluster-level significance threshold of p<0.05, corrected for multiple comparisons across the whole brain using Gaussian Random Field Theory.

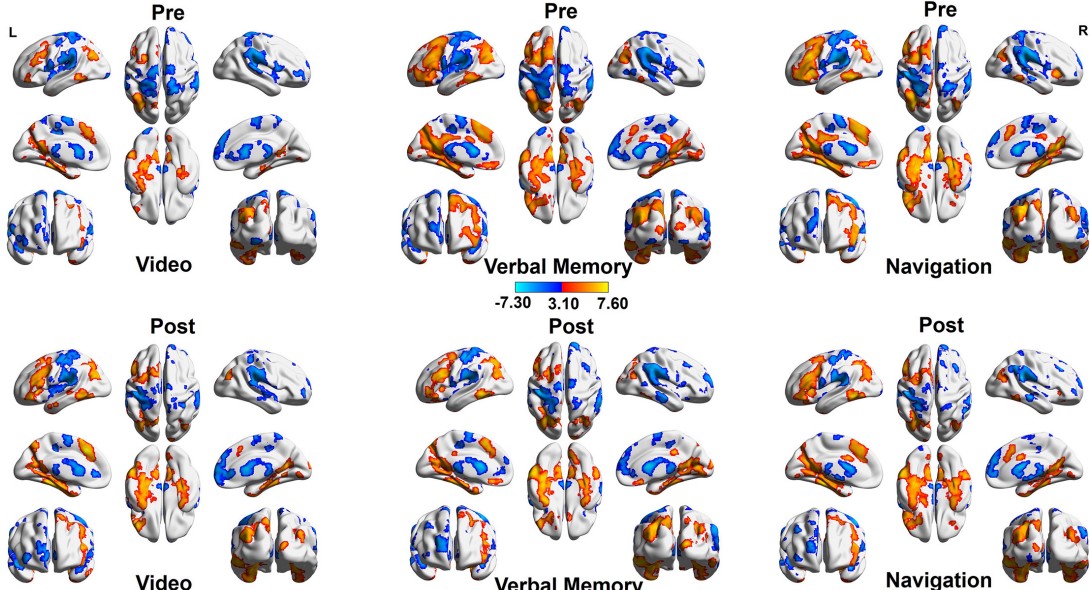

**Appendix 3—figure 7.** Univariate activation during the encoding stage of the Source Memory task, presented separately for pre-test and post-test sessions. Whole-brain mixed-effects analyses were conducted, and group-level statistical maps were thresholded using cluster-based detection methods with a voxel-wise height threshold of $z>3.1$ and a cluster-level significance threshold of $p<0.05$, corrected for multiple comparisons across the whole brain using Gaussian Random Field Theory. Results are shown for the Verbal Memory (n=25), Navigation (n=26), and Video Control (n=20) groups.

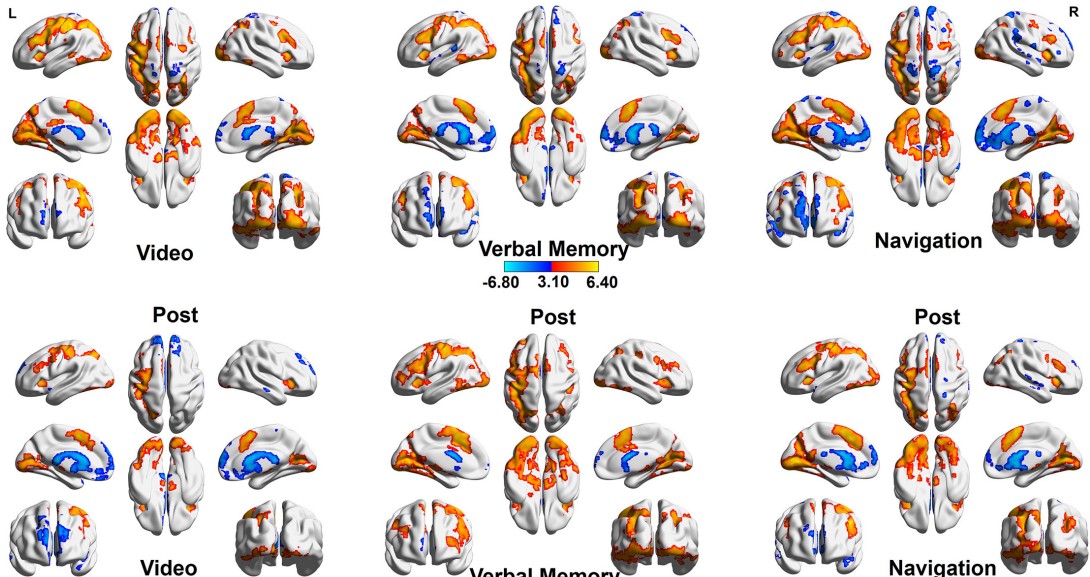

**Appendix 3—figure 8.** Univariate activation during the retrieval stage of the Source Memory task, presented separately for pre-test and post-test sessions. Whole-brain mixed-effects analyses were conducted, and group-level statistical maps were thresholded using cluster-based detection methods with a voxel-wise height threshold of $z>3.1$ and a cluster-level significance threshold of $p<0.05$, corrected for multiple comparisons across the whole brain using Gaussian Random Field Theory. Results are shown for the Verbal Memory (n=24), Navigation (n=22), and Video Control (n=20) groups.

