## [Editor Report · eLife Assessment]

This work presents a **useful** investigation of functional and structural brain changes following navigation and verbal memory training. The analyses of whole-brain volumetric changes are **convincing** and support the study's main conclusion regarding the lack of a volumetric whole-brain plasticity effects. Some analyses are **compelling** in demonstrating the presence of longitudinal behavioural effects, the presence of functional activation changes, and the lack of hippocampal volume changes.

---

## [Referee Report · Reviewer #1 (Public review)]

Summary:

This study investigates plasticity effects in brain function and structure from training in navigation and verbal memory.

The authors used a longitudinal design with a total of 75 participants across two sites. Participants were randomised to one of three conditions: verbal memory training, navigation training, or a video control condition. The results show behavioural effects in relevant tasks following the training interventions. The central claim of the paper is that network-based measures of task-based activation are affected by the training interventions, but structural brain metrics (T2w-derived volume and diffusion-weighted imaging microstructure) are not impacted by any of the training protocols tested.

Strengths:

(1) This is a well-designed study which uses two training conditions, an active control, and randomisation, as appropriate. It is also notable that the authors combined data acquisition across two sites to reach the needed sample size and accounted for it in their statistical analyses quite thoroughly. In addition, I commend the authors on using pre-registration of the analysis to enhance the reproducibility of their work.

(2) Some analyses in the paper are exhaustive and compelling in showcasing the presence of longitudinal behavioural effects, functional activation changes, and lack of hippocampal volume changes. The breadth of analysis on hippocampal volume (including hippocampal subfields) is convincing in supporting the claim regarding a lack of volumetric effect in the hippocampus.

Comments on revisions:

All my comments have been addressed. The evidence regarding lack of a volumetric effect at the whole-brain level now seems more robust. Many details are now clearer, particularly regarding the the volumetric analyses methods and the rationale and timeline of preregistration.

Minor comment:

I appreciate that there are limited possibilities with the available Diffusion-Weighted Imaging data. However, I would recommend the authors remove mentions of "white matter connectivity" in the Abstract and elsewhere, which are misleading if no tractography or voxel-wise analyses are performed.

---

## [Author Response]

The following is the authors’ response to the original reviews.

**Joint Public Review:**
Summary:This study investigates plasticity effects in brain function and structure from training in navigation and verbal memory.The authors used a longitudinal design with a total of 75 participants across two sites. Participants were randomised to one of three conditions: verbal memory training, navigation training, or a video control condition. The results show behavioural effects in relevant tasks following the training interventions. The central claim of the paper is that network-based measures of task-based activation are affected by the training interventions, but structural brain metrics (T2w-derived volume and diffusion-weighted imaging microstructure) are not impacted by any of the training protocols tested.Strengths:(1) This is a well-designed study which uses two training conditions, an active control, and randomisation, as appropriate. It is also notable that the authors combined data acquisition across two sites to reach the needed sample size and accounted for it in their statistical analyses quite thoroughly. In addition, I commend the authors on using pre-registration of the analysis to enhance the reproducibility of their work.(2) Some analyses in the paper are exhaustive and compelling in showcasing the presence of longitudinal behavioural effects, functional activation changes, and lack of hippocampal volume changes. The breadth of analysis on hippocampal volume (including hippocampal subfields) is convincing in supporting the claim regarding a lack of volumetric effect in the hippocampus.Weaknesses:(1) The rationale for the study and its relationship with previous literature is not fully clear from the paper. In particular, there is a very large literature that has already explored the longitudinal effects of different types of training on functional and structural neuroimaging. However, this literature is barely acknowledged in the Introduction, which focuses on cross-sectional studies. Studies like the one by Draganski et al. 2004 are cited but not discussed, and are clumped together with cross-sectional studies, which is confusing. As a reader, it is difficult to understand whether the study was meant to be confirmatory based on previous literature, or whether it fills a specific gap in the literature on longitudinal neuroimaging effects of training interventions.

We thank the reviewer for these comments and feedback.

We want to clarify that through our pre-registered analysis plan, our approach was confirmatory, rather than exploratory (or rather than post-hoc justified.) This confirmatory approach allowed us to critically evaluate the theoretically novel and important hypotheses which tested what no other study like our longitudinal/intervention study proposed or performed previously. We have now clarified this in the introduction.

**“**This allowed us to address the following novel theoretical questions: (1) what neural changes, if any, result from an intensive within-participant intervention that improves memory or navigation skills in healthy young adults (2) if such changes occur, what is the degree of neural overlap between the acquisition of these cognitive skills.”

“We pre-registered three novel and specific hypotheses, which are described in more detail here (https://osf.io/etxvj) ”

We have also attempted to better separate cross-section and longitudinal studies. Due to space limitations, we have focused on interventional studies that involved gray matter changes that could relevance to either navigation, episodic memory, or the hypothesized time frame we chose for the training. We also note that some of these relevant studies are discussed in more depth in the discussion.

“Successful cognitive interventions suggest that targeted within-participant cognitive training, even for as little as 1-2 weeks, can result in improvements to specific cognitive functions, including changes in focal gray matter [4,23-27]; but see[28].”

We have also added some additional citations to relevant cognitive intervention work, although we agree that this is an extensive literature, only a subset of which we are able to capture here:

“In some instances, interventions may even generalize to areas not explicitly trained but closely related to the training (termed “near transfer”)[29-33].”

(2.1) The main claim regarding the lack of changes in brain structure seems only partially supported by the analyses provided. The limited whole-brain evidence from structural neuroimaging makes it difficult to confirm whether there is indeed no effect of training. Beyond hippocampal analyses, many whole-brain analyses of both volumetric and diffusion-weighted imaging metrics are only based on coarse ROIs (for example, 34 cortical parcellations for grey matter analyses).Although vertex-wise analyses in FreeSurfer are reported, it is unclear what metrics were examined (cortical thickness? area? volume?).

We appreciate the reviewer’s thoughtful feedback. We apologize for the lack of clarity in the original manuscript regarding the type of metric used in the vertex-wise analysis. We confirm that these analyses were based on cortical volume, not thickness or area. To clarify this, we have explicitly stated in the revised Methods that the vertex-wise analyses were conducted on cortical volume using FreeSurfer’s mri_glmfit.

In addition, in response to the concern regarding the coarse nature of the ROI-based analyses, we have re-analyzed the volumetric data using the more fine-grained Destrieux atlas, which contains 148 cortical ROIs (74 per hemisphere), instead of the original, coarser 34-region atlas. These more detailed analyses still revealed no significant volume changes from pre- to post-training in any of the three groups. We believe this provides stronger support for the lack of training-induced volumetric changes outside the medial temporal lobe.

Relevant revisions have been made to the Results and Methods sections. Below is the updated content added to the manuscript:

In Results:

“We also analyzed gray matter volume changes outside of the medial temporal lobe using FreeSurfer (see Methods) to determine if any cortical or other relevant brain areas might have been affected by the training. We applied a vertex-wise analysis of cortical volume, again finding no significant differences across the entire cortex (see Methods). This finding was further validated using the Destrieux atlas, which includes 74 cortical parcellations per hemisphere (148 ROIs in total). Paired-sample t-tests revealed that none of the ROIs exhibited significant volume changes from pre- to post-test in any of the three groups (all ps > 0.542, FDR-corrected). These findings suggest that training did not result in any measurable cortical volumetric changes.”

In Methods:

“Whole-brain structural analyses were conducted using FreeSurfer (version 7.4.1; https://surfer.nmr.mgh.harvard.edu). T1-weighted anatomical images were processed using the longitudinal processing pipeline. Vertex-wise analyses of cortical volume were performed using FreeSurfer’s general linear modeling tool, mri_glmfit. Group-level comparisons were corrected for multiple comparisons using mri_glmfit-sim, which implements cluster-wise correction based on Monte Carlo simulations. A vertex-wise threshold of Z > 3.0 (corresponding to p < 0.001, two-sided) was applied to detect both positive and negative effects. Clusters were retained if they survived a cluster-wise corrected p < 0.05.

In addition to vertex-wise analysis, cortical parcellation was performed using the Destrieux atlas (aparc.a2009s), which includes 74 cortical regions per hemisphere, yielding 148 ROIs in total. To account for variability in brain size, each ROI volume was normalized by estimated intracranial volume (ICV) and scaled by a factor of 100. Longitudinal comparisons were conducted using paired-sample t-tests. To correct for multiple comparisons, we applied FDR correction (q < 0.05).”

(2.2) Diffusion-weighted imaging seems to focus on whole-tract atlas ROIs, which can be less accurate/sensitive than tractography-defined ROIs or voxel-wise approaches.

We appreciate the reviewer’s important point regarding diffusion-weighted imaging (DWI) analysis. We focused primarily on atlas-defined tract-level ROIs derived from a standard white matter tract atlas as we did not feel that we had the resolution for more fine-grained analyses with our sequences. While this approach has the advantage of robust anatomical correspondence and improved interpretability, we agree that it may be less sensitive than tractography-defined or voxel-wise methods for detecting more subtle, localized training-related changes. Because of limitations in our DWI sequence, which was optimized to be shorter and identical between different scanners, we are not able to provide more fine-grained analysis of the DWI data.

(3) Quality control of images is only mentioned for FA images in subject space. Given that most analyses are based on atlas ROIs, visual checks following registration are fundamental and should be described in further detail.

Thank you for your thoughtful comment. We agree that visual quality control is critical when using atlas-based ROI analyses. In our study, we implemented comprehensive quality control procedures across all structural and functional imaging analyses.

For hippocampal segmentation using ASHS, we performed manual visual inspections of each participant's subfield segmentation to verify the accuracy of the automated outputs. This is now clearly described in the revised Methods section:

“Each participant's subfield segmentations were manually inspected to ensure the accuracy and reliability of the segmentation protocol.”

For FreeSurfer-based hippocampal and cortical segmentation, we also conducted detailed visual inspections and manual edits following the standard FreeSurfer longitudinal pipeline. We have added the following description to the Methods section to clarify this process:

“Visual quality control was conducted by three trained raters who systematically inspected skull stripping, surface reconstruction, and segmentation accuracy at both the within-subject template and individual timepoints. Manual edits were primarily applied to the within-subject template to correct segmentation errors—particularly in challenging regions such as the hippocampus—since corrections to the template automatically propagate to all timepoints. Raters followed standardized FreeSurfer longitudinal editing guidelines to ensure consistent and reproducible corrections across subjects. Discrepancies were resolved via consensus discussion. This quality control approach enhanced the accuracy and consistency of segmentation across longitudinal scans, thereby improving the reliability of morphometric analyses and atlas-based ROI extractions.”

For functional MRI preprocessing, all registration steps—including transformations from individual functional runs to MNI space—were visually checked for each participant to ensure accurate alignment with the Schaefer atlas. We have clarified this point in the revised Methods section with the following statement:

“Prior to ROI extraction, all registration steps—from individual functional space to MNI space—were visually inspected for each participant to confirm accurate alignment between the functional images and the atlas parcellation.”

These additions now more clearly reflect the robust quality control procedures that were employed throughout our pipeline to ensure the validity of atlas-based analyses.

**Recommendations for the authors:**
(1) As a reader, I would have appreciated a short section in the methods regarding the preregistration and power analysis. Currently, it is not too straightforward to understand which analyses were included in the preregistration, and at what point in the project the pre-registration was written. Finding all the relevant information from OSF is feasible, but it would be more accessible if a summary of the information were available inside the text.

We thank the reviewer for this valuable suggestion. We agree that providing a concise summary within the manuscript's methods section will significantly improve accessibility for readers.

The full preregistration is now explicitly referenced in the Methods:

**“**Preregistration and Power Analysis

This study was preregistered on the Open Science Framework (OSF; https://osf.io/etxvj). The preregistration was completed on October 30, 2023, after approximately 80% of data collection had been completed, but prior to any analysis of the primary outcome variables. The preregistration outlines the study hypotheses, design, target sample size, and planned behavioral and neuroimaging analyses, including longitudinal ROI comparisons and statistical correction procedures.

A priori power analysis was conducted using G*Power 3.1 to estimate the required sample size for detecting a Group × Time interaction in a mixed-design ANOVA. Assuming a small-to-medium effect size (f = 0.35), we determined that 24 participants per group would provide 80% power to detect a significant effect at α = 0.05. To allow for potential attrition and data exclusion (e.g., due to excessive motion or incomplete datasets), we targeted recruitment of 30 participants per group across two study sites.

All primary hypotheses, analytic plans, and inference criteria are documented in the preregistration. Exploratory analyses are clearly delineated in both the preregistration and the present manuscript.”

(2) The relevance of the study for "disease" is mentioned in the Abstract but is absent in the Introduction. This may be worth removing?

Thank you for pointing this out. We agree that the reference to "disease" in the Abstract was not well-supported in the Introduction. To maintain consistency and avoid overstatement, we have removed the mention of "disease" from the Abstract in the revised manuscript.

In Abstract:

“Training cognitive skills, such as remembering a list of words or navigating a new city, has important implications for everyday life.”